# SELF-IMPROVEMENT IN LANGUAGE MODELS: THE SHARPENING MECHANISM

**Audrey Huang**[*]
UIUC
audreyh5@illinois.edu

**Adam Block**[*]
Microsoft Research
blockadam@microsoft.com

**Dylan J. Foster**[*]
Microsoft Research
dylanfoster@microsoft.com

**Dhruv Rohatgi**
MIT
drohatgi@mit.edu

**Cyril Zhang**
Microsoft Research
cyrilzhang@microsoft.com

**Max Simchowitz**
CMU
msimchow@andrew.cmu.edu

**Jordan T. Ash**
Microsoft Research
ash.jordan@microsoft.com

**Akshay Krishnamurthy**
Microsoft Research
akshaykr@microsoft.com

## ABSTRACT

Recent work in language modeling has raised the possibility of *self-improvement*, where a language models evaluates and refines its own generations to achieve higher performance without external feedback. It is impossible for this self-improvement to create information that is not already in the model, so why should we expect that this will lead to improved capabilities?

We offer a new perspective on the capabilities of self-improvement through a lens we refer to as *sharpening*. Motivated by the observation that language models are often better at verifying response quality than they are at generating correct responses, we formalize self-improvement as using the model itself as a verifier during post-training in order to "sharpen" the model to one placing large mass on high-quality sequences, thereby amortizing the expensive inference-time computation of generating good sequences. We begin by introducing a new statistical framework for sharpening in which the learner aims to sharpen a pre-trained base policy via sample access, and establish fundamental limits. Then, we analyze two natural families of self-improvement algorithms based on SFT and RLHF. We find that (i) the SFT-based approach is minimax optimal whenever the initial model has sufficient coverage, but (ii) the RLHF-based approach can improve over SFT-based self-improvement by leveraging online exploration, bypassing the need for coverage. Finally, we empirically validate the sharpening mechanism via inference-time and amortization experiments. We view these findings as a starting point toward a foundational understanding that can guide the design and evaluation of self-improvement algorithms.

## 1 INTRODUCTION

Contemporary language models are remarkably proficient on a wide range of natural language tasks (Brown et al., 2020; Ouyang et al., 2022; Touvron et al., 2023; OpenAI, 2023; Google, 2023), but inherit shortcomings of the data on which they were trained. A fundamental challenge is to achieve better performance than what is directly induced by the distribution of available, human-generated training data. To this end, recent work (Huang et al., 2022; Wang et al., 2022; Bai et al., 2022b; Pang et al., 2023; Yuan et al., 2024) has raised the possibility of "self-improvement," where a model—typically through forms of self-play or self-training in which the model critiques its own generations—learns to improve on its own, without external feedback. This phenomenon is somewhat counterintuitive; at first glance it would seem to disagree with the well-known data-processing inequality (Cover, 1999), which implies that no form of self-training should be able to create

---
[*]Equal contribution.

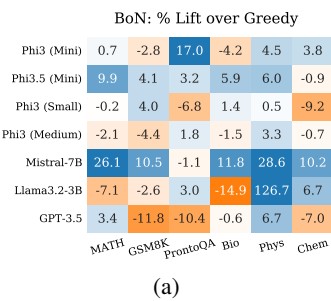 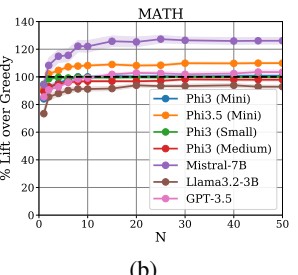 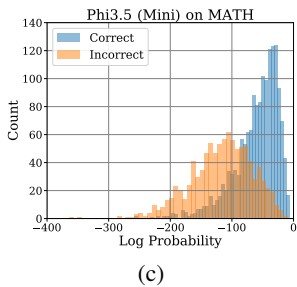

(a)                                    (b)                                    (c)

Figure 1: Validation of maximum-likelihood sharpening, via Best-of-$N$ (BoN) sampling, at inference time. (a) Percent accuracy improvement over greedy decoding for BoN sharpening with $N = 50$ on 6 tasks and 7 models, colored by performance. (b) Percent accuracy improvement over greedy for BoN sharpening as a function of $N$ for 7 different models on the MATH dataset. (c) Distribution of sequence-level log probabilities for responses sampled from `Phi3.5-Mini` ($N = 1$) on the MATH dataset, conditioned on correctness. Correct completions have noticeably higher likelihood than incorrect completions, demonstrating the utility of inference-time sharpening.

information not already in the model. This motivates the question of why we should expect such supervision-free interventions will lead to stronger reasoning and planning capabilities.

A dominant hypothesis for why improvement without external feedback might be possible is that models contain "hidden knowledge" (Hinton et al., 2015) that is difficult to access. Self-improvement, rather than creating knowledge from nothing, is a means of extracting and distilling this knowledge into a more accessible form, and thus is a computational phenomenon rather than a statistical one. While there is a growing body of empirical evidence for this hidden-knowledge hypothesis (Furlanello et al., 2018; Gotmare et al., 2019; Dong et al., 2019; Abnar et al., 2020; Allen-Zhu & Li, 2020), particularly in the context of self-distillation, a fundamental understanding of self-improvement remains missing. Concretely, where in the model is this hidden knowledge, and when and how can it be extracted?

## 1.1 OUR PERSPECTIVE: THE SHARPENING MECHANISM

In this paper we posit a source of hidden knowledge, and offer a formal perspective on how to extract it. Our starting point is the widely observed phenomenon that language models are often better at verifying whether responses are correct than they are at generating correct responses (Huang et al., 2022; Wang et al., 2022; Bai et al., 2022b; Pang et al., 2023; Yuan et al., 2024). This gap may be explained by the theory of computational complexity, which suggests that generating high-quality responses can be less computationally tractable than verification (Cook, 1971; Levin, 1973; Karp, 1972). In autoregressive language modeling, computing the most likely response for a given prompt is NP-hard in the worst case (Appendix E), whereas the model's likelihood for a given response can be easily evaluated.

We view self-improvement as any attempt to narrow this gap, i.e., use the model as its own verifier to improve generation and *sharpen* the model toward high-quality responses. Formally, consider a learner with access to a base model $\pi_{\text{base}} : \mathcal{X} \to \Delta(\mathcal{Y})$ representing a conditional distribution that maps a prompt $x \in \mathcal{X}$ to a distribution over responses (i.e., $\pi_{\text{base}}(y \mid x)$ is the probability that the model generates the response $y$ given the prompt $x$).[1] We posit that $\pi_{\text{base}}$ has already been trained in some manner (e.g., through next-token prediction or additional post-training steps such as SFT or RLHF), with the key feature being that $\pi_{\text{base}}$ is a good verifier, as measured by some *self-reward* function $r_{\text{self}}(y \mid x; \pi_{\text{base}})$ measuring model certainty. The self-reward function is derived purely from the base model $\pi_{\text{base}}$, without external supervision or feedback. Examples include normalized and/or regularized sequence likelihood (Meister et al., 2020), models-as-judges (Zheng et al., 2024; Yuan et al., 2024; Wu et al., 2024a; Wang et al., 2024), and model confidence (Wang & Zhou, 2024).

---

[1] Our general results are agnostic to the structure of $\mathcal{X}$, $\mathcal{Y}$, and $\pi_{\text{base}}$, but an important special case for language modeling is the autoregressive setting where $\mathcal{Y} = \mathcal{V}^H$ for a vocabulary space $\mathcal{V}$ and sequence length $H$, and where $\pi_{\text{base}}$ has the autoregressive structure $\pi_{\text{base}}(y_{1:H} \mid x) = \prod_{h=1}^{H} \pi_{\text{base},h}(y_h \mid y_{1:h-1}, x)$ for $y = y_{1:H} \in \mathcal{Y}$.

> **Sharpening**
>
> We refer to **sharpening** as any process that tilts $\pi_{\mathsf{base}}$ toward responses that are more certain in the sense that they enjoy greater self-reward $r_{\mathsf{self}}$. That is, a sharpened model $\widehat{\pi}$ is one that (approximately) maximizes the self-reward:
>
> $$\widehat{\pi}(x) \approx \arg\max_{y \in \mathcal{Y}} r_{\mathsf{self}}(y \mid x; \pi_{\mathsf{base}}). \tag{1}$$

An important special case for sharpening is in language/autoregressive modeling. Here, we have $\mathcal{Y} = \mathcal{V}^H$ for a vocabulary space $\mathcal{V}$ and sequence length $H$, and $\pi_{\mathsf{base}}$ has the autoregressive structure $\pi_{\mathsf{base}}(y_{1:H} \mid x) = \prod_{h=1}^{H} \pi_{\mathsf{base},h}(y_h \mid y_{1:h-1}, x)$ for $y = y_{1:H} \in \mathcal{Y}$. Sharpening in this setting pertains to entire responses, i.e., the optimization over responses in Eq. (1) is at the *sequence level*. In contrast, popular decoding strategies such as greedy, low-temperature sampling, and beam search operate at the token-level; nevertheless, they can be viewed as heuristics for *inference-time sharpening*.[2] The combinatorial response space can make sharpening computationally demanding and so, an appealing alternative to inference-time sharpening is *amortization via self-training* (Section 2). The latter captures many existing self-training schemes (Huang et al., 2022; Wang et al., 2022; Bai et al., 2022b; Pang et al., 2023; Yuan et al., 2024), and is the main focus of this paper; we use the term *sharpening* without further qualification to refer to the latter.

We refer to the **sharpening mechanism** as the phenomenon where responses from a model with the highest certainty (in the sense of large self-reward $r_{\mathsf{self}}$) exhibit the greatest performance on a task of interest. Though it is unclear a-priori whether there are self-rewards related to task performance, the successes of self-improvement in prior works (Huang et al., 2022; Wang et al., 2022; Bai et al., 2022b; Pang et al., 2023; Yuan et al., 2024) give strong positive evidence. These works suggest that, in many settings, models do have hidden knowledge: the model's own self-reward correlates with response quality, but it is computationally challenging to generate high self-rewarding—and thus high quality—responses. It is the role of (algorithmic) sharpening to leverage these verifications to improve the quality of generations, despite computational difficulty.

## 1.2 CONTRIBUTIONS

We initiate the theoretical study of self-improvement via the sharpening mechanism. We disentangle the choice of self-reward from the algorithms used to optimize it, and aim to understand: (i) When and how does self-training achieve sharpening? (ii) What are the fundamental limits for such algorithms?

**Algorithms for sharpening (Section 2).** The starting point for our work is to consider two natural families of self-improvement algorithms based on supervised fine-tuning (SFT) and reinforcement learning (RL/RLHF), respectively, `SFT-Sharpening` and `RLHF-Sharpening`. Both algorithms **amortize** the sharpening objective (1) into a dedicated post-training/fine-tuning phase:

- `SFT-Sharpening` filters responses where the self-reward $r_{\mathsf{self}}(y \mid x; \pi_{\mathsf{base}})$ is large and fine-tunes on the resulting dataset, invoking common SFT pipelines (Amini et al., 2024; Sessa et al., 2024).

- `RLHF-Sharpening` directly applies reinforcement learning techniques (e.g., PPO (Schulman et al., 2017) or DPO (Rafailov et al., 2023)) to optimize the self-reward function $r_{\mathsf{self}}(y \mid x; \pi_{\mathsf{base}})$.

In the remainder of the paper, we introduce a theoretical framework to analyze the performance of these algorithms. Our main contributions are as follows.

**Maximum-likelihood sharpening objective (Section 3.1).** As a concrete proposal for one source of hidden knowledge, we focus on self-rewards defined by the model's sequence-level log-probabilities:

$$r_{\mathsf{self}}(y \mid x; \pi_{\mathsf{base}}) := \log \pi_{\mathsf{base}}(y \mid x) \tag{2}$$

This is a stylized self-reward function, which offers perhaps the simplest objective for self-improvement in the absence of external feedback (i.e., purely supervision-free), yet also connects self-improvement to a rich body of theoretical computer science literature on computational trade-offs for optimization (inference) versus sampling (Appendix B). Despite its simplicity, maximum-likelihood sharpening is already sufficient to achieve non-trivial performance gains over

---

[2] More sophisticated decoding strategies like normalized/regularized sequence likelihood (Meister et al., 2020) or chain-of-thought decoding (Wang & Zhou, 2024) also admit an interpretation as sharpening; see Appendix B.

greedy decoding on a range of reasoning tasks with several language models; (Figure 1). We believe it can serve as a starting point toward understanding forms of self-improvement that use more sophisticated self-rewards (Huang et al., 2022; Wang et al., 2022; Pang et al., 2023; Yuan et al., 2024).

**A statistical framework for sharpening (Sections 3.2 and 3.3).** Though the goal of sharpening is computational in nature, we recast self-training according to the maximum-likelihood sharpening objective Eq. (2) as a **statistical** problem where we aim to produce a model approximating (1) using a polynomial number of (i) sample prompts $x \sim \mu$, (ii) sampling queries of the form $y \sim \pi_{\mathsf{base}}(x)$, and (iii) likelihood evaluations of the form $\pi_{\mathsf{base}}(y \mid x)$. Evaluating the efficiency of the algorithm through the number of such queries, this abstraction offers a natural way to evaluate the performance of self-improvement/sharpening algorithms and establish fundamental limits; we use our framework to prove new lower bounds that highlight the importance of the base model's coverage.

**Analysis of sharpening algorithms (Section 4).** Within our statistical framework for sharpening, we show that `SFT-Sharpening` and `RLHF-Sharpening` provably converge to sharpened models, establishing several results: **(i) `SFT-Sharpening` is minimax optimal**, and learns a sharpened model whenever $\pi_{\mathsf{base}}$ has sufficient coverage (we also show that a novel variant based on adaptive sampling can sidestep the minimax lower bound); **(ii) `RLHF-Sharpening` benefits from on-policy exploration**, and can bypass the need for coverage—improving over `SFT-Sharpening`.

**Empirical investigation (Appendix A).** We explore empirically the extent to which our theoretical framework and methods improve language model performance in a variety of tasks. We consider three choices of self-reward on an extensive list of model-dataset pairs and conclude that sharpening can often improve performance. We then implement one of our algorithms, SFT-Sharpening, on a subset of these model-dataset pairs and observe a significant positive effect on performance. A summary of our inference-time experiments can be found in Figure 1.

### 1.3 RELATED WORK

Our work is most directly related to a growing body of empirical research that studies self-training for language models in a supervision-free setting with no external feedback (Huang et al., 2022; Wang et al., 2022; Bai et al., 2022b; Pang et al., 2023; Yuan et al., 2024). The specific algorithms for self-improvement/sharpening we study can be viewed as applications of standard alignment algorithms (Amini et al., 2024; Sessa et al., 2024; Christiano et al., 2017; Bai et al., 2022a; Ouyang et al., 2022; Rafailov et al., 2023) with a specific choice of reward function. However, the maximum likelihood sharpening objective (2) used for our theoretical results has been relatively unexplored within the alignment and self-improvement literature.

Theoretical understanding of self-training is currently limited. One line of work analyzes the convergence of self-training for classification and regression with the *self-distillation objective*, but is limited to stylized setups such as linear models (Mobahi et al., 2020; Frei et al., 2022; Das & Sanghavi, 2023; Das et al., 2024; Pareek et al., 2024), feedforward neural networks (Allen-Zhu & Li, 2020), and a general PAC-style framework (Boix-Adsera, 2024). To the best of our knowledge, our work is the first to study self-training in a general framework that subsumes language modeling. See Appendix B for a more extensive discussion of related work.

## 2 SHARPENING ALGORITHMS FOR SELF-IMPROVEMENT

This section introduces the two families of self-improvement algorithms for sharpening that we study. Going forward, we omit the dependence of $r_{\mathsf{self}}$ on $\pi_{\mathsf{base}}$ when it is clear from context. We use the notation $\arg\max_{\pi \in \Pi}$ or $\arg\min_{\pi \in \Pi}$ to denote exact optimization over a user-specified model class $\Pi$ for theoretical results (Agarwal et al., 2019; Foster & Rakhlin, 2023); empirically, these operations can be implemented by training a neural network to low loss.

### 2.1 SELF-IMPROVEMENT THROUGH SFT: `SFT-Sharpening`

`SFT-Sharpening` filters responses for which the self-reward $r_{\mathsf{self}}(y \mid x)$ is large, and applies standard supervised fine-tuning on the resulting dataset (Amini et al., 2024; Sessa et al., 2024; Gui et al., 2024; Pace et al., 2024). This can be viewed as amortizing inference-time sharpening via the effective-but-costly best-of-$N$ sampling approach (Brown et al., 2024; Snell et al., 2024; Wu et al., 2024b). Concretely, suppose we have a collection of prompts $x_1, \ldots, x_n$. For each prompt, we sample $N$ responses $y_{i,1}, \ldots, y_{i,N} \sim \pi_{\mathsf{base}}(\cdot \mid x_i)$, then compute the best-of-$N$ response

$y_i^{\mathsf{BoN}} = \arg\max_{j \in [N]}\{r_{\mathsf{self}}(y_{i,j} \mid x_i)\}$, scoring via the model's self-reward function. We compute the sharpened model via supervised fine-tuning on the best-of-$N$ responses:

$$\widehat{\pi}^{\mathsf{BoN}} = \arg\max_{\pi \in \Pi} \sum_{i=1}^{n} \log \pi(y_i^{\mathsf{BoN}} \mid x_i).$$

This is a simple, flexible self-training scheme, and converges to a sharpened model as $n, N \to \infty$.

## 2.2 SELF-IMPROVEMENT THROUGH RLHF: RLHF-Sharpening

A drawback of the SFT-Sharpening algorithm is that it may ignore useful information contained in the self-reward function $r_{\mathsf{self}}(y \mid x)$. Fixing a regularization parameter $\beta > 0$ throughout, our second class of algorithms solve a KL-regularized reinforcement learning problem in the spirit of RLHF and other alignment methods (Christiano et al., 2017; Rafailov et al., 2023). Defining $\mathbb{E}_\pi[\cdot] = \mathbb{E}_{x \sim \mu, y \sim \pi(\cdot \mid x)}[\cdot]$ and $D_{\mathsf{KL}}(\pi \,\|\, \pi_{\mathsf{base}}) = \mathbb{E}_\pi\left[\log \frac{\pi(y \mid x)}{\pi_{\mathsf{base}}(y \mid x)}\right]$, we choose

$$\widehat{\pi} \approx \arg\max_{\pi \in \Pi}\{\mathbb{E}_\pi[r_{\mathsf{self}}(y \mid x)] - \beta D_{\mathsf{KL}}(\pi \,\|\, \pi_{\mathsf{base}})\}. \tag{3}$$

The exact optimizer $\pi_\beta^\star = \arg\max_{\pi \in \Pi}\{\mathbb{E}_\pi[r_{\mathsf{self}}(y \mid x)] - \beta D_{\mathsf{KL}}(\pi \,\|\, \pi_{\mathsf{base}})\}$ for this objective has the form $\pi_\beta^\star(y \mid x) \propto \pi_{\mathsf{base}}(y \mid x) \cdot \exp(\beta^{-1} r_{\mathsf{self}}(y \mid x))$, which converges to the solution to the sharpening objective in Eq. (1) as $\beta \to 0$. Thus, Eq. (3) can be seen to encourage sharpening.

There are many choices for what RLHF/alignment algorithm one might use to solve (3). For our theoretical results, we implement Eq. (3) using an approach inspired by DPO and its reward-based variants (Rafailov et al., 2023; Gao et al., 2024). Given a dataset $\mathcal{D} = \{(x, y, y')\}$ of $n$ examples sampled via $x \sim \mu$ and $y, y' \sim \pi_{\mathsf{base}}(y \mid x)$, we consider the algorithm that solves

$$\widehat{\pi} \in \arg\min_{\pi \in \Pi} \sum_{(x,y,y') \in \mathcal{D}} \left(\beta \log \frac{\pi(y \mid x)}{\pi_{\mathsf{base}}(y \mid x)} - \beta \log \frac{\pi(y' \mid x)}{\pi_{\mathsf{base}}(y' \mid x)} - \left(r_{\mathsf{self}}(y \mid x) - r_{\mathsf{self}}(y' \mid x)\right)\right)^2. \tag{4}$$

In Section 4, we show that this approach achieves guarantees similar to SFT-Sharpening, while a more sophisticated DPO variant with *online exploration* (Xie et al., 2024) provides provable benefits.

## 3 A STATISTICAL FRAMEWORK FOR SHARPENING

This section introduces the theoretical framework within which we will analyze the SFT-Sharpening and RLHF-Sharpening algorithms. We first introduce the maximum-likelihood sharpening objective as a stylized self-reward function, then introduce our statistical framework for sharpening. We write $f = \widetilde{O}(g)$ to denote $f = O(g \cdot \max\{1, \mathrm{polylog}(g)\})$ and $a \lesssim b$ as shorthand for $a = O(b)$.

### 3.1 MAXIMUM-LIKELIHOOD SHARPENING

Our theoretical results focus on the maximum-likelihood sharpening objective given by

$$r_{\mathsf{self}}(y \mid x) := \log \pi_{\mathsf{base}}(y \mid x),$$

which we aim to maximize using conditional samples $y \sim \pi_{\mathsf{base}}(\cdot \mid x)$ from the base model. This is a simple and stylized self-reward function, but we will show that it enjoys a rich theory. In particular, we can restate the problem of sharpening with this self-reward through the lens of *amortization*.

> *Can we efficiently **amortize maximum likelihood inference (optimization)** for a conditional distribution $\pi_{\mathsf{base}}(y \mid x)$ given access to a **sampling oracle** that can sample $y \sim \pi_{\mathsf{base}}(\cdot \mid x)$?*

The tacit assumption in this framing is that the maximum-likelihood response constitutes a useful form of hidden knowledge. Maximum-likelihood sharpening connects the study of self-improvement to a large body of research in theoretical computer science demonstrating computational reductions between optimization (inference) and sampling (generation) (Kirkpatrick et al., 1983; Lovász & Vempala, 2006; Singh & Vishnoi, 2014; Ma et al., 2019; Talwar, 2019). Our sharpening framework offers a new learning-theoretic perspective by focusing on the problem of amortizing this type of reduction.

We evaluate the quality of an approximately sharpened model as follows. Let

$$\boldsymbol{y}^\star(x) := \arg\max_{y \in \mathcal{Y}} \log \pi_{\mathsf{base}}(y \mid x);$$

we interpret $\boldsymbol{y}^\star(x) \subset \mathcal{Y}$ as a set to accommodate non-unique maximizers, and will write $y^\star(x)$ to indicate a unique maximizer when it exists (i.e., when $\boldsymbol{y}^\star(x) = \{y^\star(x)\}$).

**Definition 3.1** (Sharpened model). *We say that a model $\widehat{\pi}$ is $(\epsilon, \delta)$-sharpened relative to $\pi_{\mathsf{base}}$ if*
$$\mathbb{P}_{x \sim \mu}[\widehat{\pi}(\boldsymbol{y}^\star(x) \mid x) \geq 1 - \delta] \geq 1 - \epsilon.$$

That is, an $(\epsilon, \delta)$-sharpened model places at least $1 - \delta$ mass on arg-max responses on all but an $\epsilon$-fraction of prompts under $\mu$. For small $\delta$ and $\epsilon$, we are guaranteed that $\widehat{\pi}$ is a high-quality generator: sampling from the model will produce an arg-max response with high probability for most prompts.

**Maximum-likelihood sharpening for autoregressive models.** Though our most general results are agnostic to the structure of $\mathcal{X}$, $\mathcal{Y}$, and $\pi_{\mathsf{base}}$, our primary motivation is the autoregressive setting in which $\mathcal{Y} = \mathcal{V}^H$ for a *vocabulary space* $\mathcal{V}$ and sequence length $H$, and where $\pi_{\mathsf{base}}$ has the autoregressive structure $\pi_{\mathsf{base}}(y_{1:H} \mid x) = \prod_{h=1}^H \pi_{\mathsf{base},h}(y_h \mid y_{1:h-1}, x)$ for $y = y_{1:H} \in \mathcal{Y}$. We observe that when the response $y = (y_1, \ldots, y_H) \in \mathcal{Y} = \mathcal{V}^H$ is a sequence of tokens, the maximum-likelihood sharpening objective (2) sharpens toward the *sequence-level* arg-max response:
$$\arg\max_{y_{1:H}} \log \pi_{\mathsf{base}}(y_{1:H} \mid x). \tag{5}$$

Although somewhat stylized, Eq. (5) is a non-trivial (in general, computationally intractable; see Appendix E) solution concept. We view the sequence-level arg-max as a form of hidden knowledge that cannot necessarily be uncovered through naive sampling or greedy decoding.

**Role of $\delta$ for autoregressive models.** As can be verified through simple examples, beam-search and greedy tokenwise decoding do not return an exact (or even approximate) solution to (5) in general. There is one notable exception: If the model has already been sharpened to $\delta < 1/2$ and the arg-max sequence is unique, then greedy decoding will succeed.

**Proposition 3.1** (Greedy decoding succeeds for sharpened policies). *Let $\pi = \pi_{1:H}$ be an autoregressive model defined over response space $\mathcal{Y} = \mathcal{V}^H$. For a given prompt $x \in \mathcal{X}$, if $\boldsymbol{y}^\star(x) = \{y^\star(x)\}$ is a singleton and $\pi(y^\star(x) \mid x) > 1/2$, then the greedy decoding strategy that selects $\widehat{y}_h = \arg\max_{y_h \in \mathcal{V}} \pi_h(y_h \mid \widehat{y}_1, \ldots, \widehat{y}_{h-1}, x)$ guarantees that $\widehat{y} = y^\star(x)$. This result is tight, in the sense that there exist $\pi$ with $\pi(y^\star(x) \mid x) \leq 1/2$ for which greedy decoding fails to recover $y^\star(x)$.*

This means that if we start from an un-sharpened model, simply sharpening to $\delta < 1/2$ may suffice.

## 3.2 SAMPLE COMPLEXITY FRAMEWORK

Sharpening, as described in Definition 3.1, is a purely computational problem, which makes it difficult to evaluate the optimality of self-improvement algorithms. To address this, we introduce a novel statistical framework for sharpening, inspired by the oracle complexity in optimization (Nemirovski et al., 1983; Traub et al., 1988; Raginsky & Rakhlin, 2011; Agarwal et al., 2012) and statistical query complexity in computational learning theory (Blum et al., 1994; Kearns, 1998; Feldman, 2012; 2017).

**Definition 3.2** (Sample-and-evaluate framework). *In the **sample-and-evaluate** framework, the algorithm designer does not have explicit access to the base model $\pi_{\mathsf{base}}$. Instead, they access $\pi_{\mathsf{base}}$ only through* sample-and-evaluate queries: *The learner is allowed to sample $n$ prompts $x \sim \mu$. For each prompt $x$, they can sample $N$ responses $y_1, y_2, \ldots y_N \sim \pi_{\mathsf{base}}(\cdot \mid x)$ and observe the likelihood $\pi_{\mathsf{base}}(y_i \mid x)$ for each such response. The efficiency, or* sample complexity*, of the algorithm is measured through the total number of sample-and-evaluate queries $m := n \cdot N$.*

This framework can be seen to capture algorithms like SFT-Sharpening and RLHF-Sharpening (implemented with DPO), which only access the base model $\pi_{\mathsf{base}}$ through i) sampling responses via $y \sim \pi_{\mathsf{base}}(\cdot \mid x)$ (**generation**), and ii) evaluating the likelihood $\pi_{\mathsf{base}}(y \mid x)$ (**verification**) for these responses. We view the sample complexity $m = n \cdot N$ as a natural statistical abstraction for the computational complexity of self-improvement (a clear parallel to oracle complexity for optimization algorithms), one which is amenable to information-theoretic lower bounds.[3] We will aim to show that, under appropriate assumptions, SFT-Sharpening and RLHF-Sharpening can learn an $(\epsilon, \delta)$-sharpened model with sample complexity
$$m = \mathrm{poly}(\epsilon^{-1}, \delta^{-1}, C_{\mathsf{prob}})$$
where $C_{\mathsf{prob}}$ is a potentially problem-dependent constant.

---

[3]Concretely, the sample complexity $m = n \cdot N$ is a lower bound on the running time of any algorithm that operates in the sample-and-evaluate framework.

### 3.3 FUNDAMENTAL LIMITS

Before diving into our analysis of SFT-Sharpening and RLHF-Sharpening in the sample-and-evaluate framework, let us take a brief detour to give a sense for how sample complexity guarantees for sharpening should scale. To this end, we will prove a lower bound or fundamental limit on the sample complexity of any algorithm in the sample-and-evaluate framework.

Intuitively, the performance of any sampling-based sharpening algorithm should depend on well the base model $\pi_{\text{base}}$ covers the arg-max response $y^\star(x)$. To capture this, we use the *coverage coefficient*[4]

$$C_{\text{cov}} = \mathbb{E}_{x\sim\mu}\left[\frac{1}{\pi_{\text{base}}(\boldsymbol{y}^\star(x)\mid x)}\right], \tag{6}$$

and, for a model $\pi$, we define $\boldsymbol{y}^\pi(x) = \arg\max_{y\in\mathcal{Y}}\pi(y\mid x)$ and $C_{\text{cov}}(\pi) = \mathbb{E}_{x\sim\mu}\left[\frac{1}{\pi(\boldsymbol{y}^\pi(x)\mid x)}\right]$.

Our main lower bound shows that for a worst-case choice of $\Pi$, the coverage coefficient serves as a lower bound on the sample complexity of any sharpening algorithm.

**Theorem 3.1** (Lower bound for sharpening)**.** *Fix an integer $d \geq 1$ and parameters $\epsilon \in (0,1)$ and $C \geq 1$. There exists a class of models $\Pi$ such that (i) $\log|\Pi| \asymp d(1 + \log(C\epsilon^{-1}))$, (ii) $\sup_{\pi\in\Pi} C_{\text{cov}}(\pi) \lesssim C$, and (iii) $\boldsymbol{y}^\pi(x)$ is a singleton for all $\pi \in \Pi$, $x \in \mathcal{X}$. Any sharpening algorithm $\widehat{\pi}$ that achieves $\mathbb{E}[\mathbb{P}_{x\sim\mu}[\widehat{\pi}(\boldsymbol{y}^{\pi_{\text{base}}}(x)\mid x) > 1/2]] \geq 1-\epsilon$ for all $\pi_{\text{base}} \in \Pi$ must collect a total number of samples $m = n \cdot N$ at least*

$$m \gtrsim \frac{C\log|\Pi|}{\epsilon^2 \cdot (1 + \log(C\epsilon^{-1}))}.$$

This result shows that the complexity of any $(\epsilon, 1/2 - \delta)$-sharpening algorithm (for $\delta > 0$) in the sample-and-evaluate framework must depend polynomially on the coverage coefficient $C_{\text{cov}}$, as well as the accuracy $\epsilon$. The lower bound also depends on the expressivity of $\pi_{\text{base}}$, as captured by the model class complexity term $\log|\Pi|$. We will show in the sequel that it is possible to match this lower bound. Note that this result also implies a lower bound for the general sharpening problem (i.e., general $r_{\text{self}}$), since maximum-likelihood sharpening is a special case.

**Remark 3.1** (Relaxed notions of sharpening and coverage)**.** *The notion of coverage in Eq. (6) is somewhat stringent, since it requires that $\pi_{\text{base}}$ place large mass on $\boldsymbol{y}^\star(x)$ on average. In Appendix F, we introduce a more general and permissive notion of* approximate sharpening *(Definition F.1) which leads to weaker coverage requirements, and use this to give generalized versions of our main results.*

We close this section by noting that numerous recent works—focusing on inference-time computation—show that standard language models exhibit favorable coverage with respect to desirable responses (Brown et al., 2024; Snell et al., 2024; Wu et al., 2024b). We replicate these findings in our experimental setup in Appendix A. These works suggest that, despite the exponentially large response space, the coverage coefficient $C_{\text{cov}}$ may be small in standard language modeling tasks.

## 4 ANALYSIS OF SHARPENING ALGORITHMS

Equipped with the sample complexity framework from Section 3, we now prove that the SFT-Sharpening and RLHF-Sharpening families of algorithms provably learn a sharpened model for the maximum likelihood sharpening objective. We treat the model class $\Pi$ as a fixed, user-specified input. In the tradition of statistical learning theory, our results allow for general classes $\Pi$ and are agnostic to its structure beyond standard generalization arguments.

### 4.1 ANALYSIS OF SFT-Sharpening

Recall that when we specialize to the maximum-likelihood sharpening self-reward, the SFT-Sharpening algorithm takes the form $\widehat{\pi}^{\text{BoN}} = \arg\max_{\pi\in\Pi}\sum_{i=1}^n \log\pi_{\text{base}}(y_i^{\text{BoN}}\mid x_i)$, where $y_i^{\text{BoN}} = \arg\max_{j\in[N]}\{\log\pi_{\text{base}}(y_{i,j}\mid x_i)\}$ for $y_{i,1},\ldots,y_{i,N}\sim\pi_{\text{base}}(\cdot\mid x_i)$.

To analyze SFT-Sharpening, we first make a realizability assumption. Let $\pi_N^{\text{BoN}}(x)$ be the distribution of the random variable $y_N^{\text{BoN}}(x) \sim \arg\max\{\log\pi_{\text{base}}(y_i\mid x)\mid y_1,\ldots,y_N\sim\pi_{\text{base}}(x)\}$.

---

[4]This quantity can be interpreted as a special case of the $L_1$-concentrability coefficient (Farahmand et al., 2010; Xie & Jiang, 2020; Zanette et al., 2021; Amortila et al., 2024) studied in the theory of offline reinforcement learning.

**Assumption 4.1.** *The model class $\Pi$ satisfies $\pi_N^{\mathsf{BoN}} \in \Pi$.*

Our main sample complexity guarantee for SFT-Sharpening is as follows.

**Theorem 4.1** (Sample complexity of SFT-Sharpening). *Let $\epsilon, \delta, \rho \in (0, 1)$ be given, and suppose we set $n = c \cdot \frac{\log(|\Pi|\rho^{-1})}{\delta\epsilon}$ and $N^\star = c \cdot \frac{C_{\mathsf{cov}} \log(2\delta^{-1})}{\epsilon}$ for an appropriate constant $c > 0$. Then with probability at least $1 - \rho$, SFT-Sharpening produces a model $\widehat{\pi}$ such that that $\mathbb{P}_{x\sim\mu}[\widehat{\pi}(\boldsymbol{y}^\star(x) \mid x) \leq 1 - \delta] \leq \epsilon$, and has total sample complexity[5]*

$$m = O\left(\frac{C_{\mathsf{cov}} \log(|\Pi|\rho^{-1}) \log(\delta^{-1})}{\delta\epsilon^2}\right). \tag{7}$$

This result shows that SFT-Sharpening is minimax optimal in the sample-and-evaluate framework when $\delta$ is constant. In particular, the bound in Eq. (7) matches the lower bound in Theorem 3.1 up to polynomial dependence on $\delta$ and logarithmic factors. Whether the $1/\delta$ factor in Eq. (7) can be removed is an interesting technical question, but may not be practically consequential because—as discussed in Section 3.2—the regime $\delta < 1/2$ is most meaningful for autoregressive language modeling.

**Remark 4.1** (On realizability and coverage). *Realizability assumptions such as Assumption 4.1 (which asserts that the class $\Pi$ is powerful enough to model the distribution of the best-of-$N$ responses) are standard in learning theory (Agarwal et al., 2019; Foster & Rakhlin, 2023), though certainly non-trivial (see Appendix E for a natural example where they may not hold). The coverage assumption, while also standard, when combined with the hypothesis that high-likelihood responses are desirable, suggests that $\pi_{\mathsf{base}}$ generates high-quality responses with reasonable probability. In general, doing so may require leveraging non-trivial* serial *computation at inference time via procedures such as Chain-of-Thought (Wei et al., 2022). Although recent work shows that such serial computation* cannot *be amortized (Li et al., 2024; Malach, 2023), SFT-Sharpening instead amortizes the* parallel *computation of best-of-$N$ sampling, and thus has different representational considerations.*

**Benefits of adaptive sampling.** SFT-Sharpening is optimal in the sample-and-evaluate framework, but we show in Appendix D that a variant which selects the number of responses adaptively based on the prompt $x$ can bypass this lower bound, improving the $\epsilon$-dependence in Eq. (7) from $\frac{1}{\epsilon^2}$ to $\frac{1}{\epsilon}$.

**Empirical validation.** In Appendix A, we empirically investigate the benefits of BoN on a variety of model-dataset pairs. Our results, summarized in Table 1 and Figs. 7 and 8, broadly show that the aforementioned benefits of inference-time sharpening, to an extent, amortized at training time.

## 4.2 ANALYSIS OF RLHF-Sharpening

We now turn our attention to theoretical guarantees for the RLHF-Sharpening algorithm family, which uses tools from reinforcement learning to optimize the self-reward function. When specialized to maximum-likelihood sharpening, the RL objective used by RLHF-Sharpening takes the form $\widehat{\pi} \approx \arg\max_{\pi\in\Pi}\{\mathbb{E}_\pi[\log\pi_{\mathsf{base}}(y \mid x)] - \beta D_{\mathsf{KL}}(\pi \| \pi_{\mathsf{base}})\}$ for $\beta > 0$. The exact optimizer $\pi_\beta^\star = \arg\max_{\pi\in\Pi}\{\mathbb{E}_\pi[\log\pi_{\mathsf{base}}(y \mid x)] - \beta D_{\mathsf{KL}}(\pi \| \pi_{\mathsf{base}})\}$ for this objective has the form $\pi_\beta^\star(y \mid x) \propto \pi_{\mathsf{base}}^{1+\beta^{-1}}(y \mid x)$, which converges to a sharpened model (per Definition 3.1) as $\beta \to 0$.

The key challenge we encounter in this section is the mismatch between the RL reward $\log\pi_{\mathsf{base}}(y \mid x)$ and the sharpening desideratum $\widehat{\pi}(\boldsymbol{y}^\star(x) \mid x)$. For example, suppose a unique argmax—say, $y^\star(x)$—and second-to-argmax—say, $y'(x)$—are nearly as likely under $\pi_{\mathsf{base}}$. Then the RL reward $\mathbb{E}_{\widehat{\pi}}[\log\pi_{\mathsf{base}}(y \mid x)]$ must be optimized to extremely high precision before $\widehat{\pi}$ can be guaranteed to distinguish the two. To quantify this effect, we introduce a *margin condition*.

**Assumption 4.2** (Margin). *For a margin parameter $\gamma_{\mathsf{margin}} > 0$, the base model $\pi_{\mathsf{base}}$ satisfies*

$$\max_{y\in\mathcal{Y}} \pi_{\mathsf{base}}(y \mid x) \geq (1 + \gamma_{\mathsf{margin}}) \cdot \pi_{\mathsf{base}}(y' \mid x) \quad \forall y' \notin \boldsymbol{y}^\star(x), \quad \forall x \in \mathsf{supp}(\mu).$$

SFT-Sharpening does not suffer from the pathology in the example above, because once $y^\star(x)$ and $y'(x)$ are drawn in a batch of $N$ responses, we have $y_i^{\mathsf{BoN}} = y^\star(x_i)$ regardless of margin. However, as we shall show in Section 4.2.2, the RLHF-Sharpening algorithm is amenable to online exploration, which may improve dependence on other problem parameters.

---

[5]We focus on finite classes for simplicity, following a convention in reinforcement learning theory (Agarwal et al., 2019; Foster & Rakhlin, 2023), but our results extend to infinite classes through standard arguments.

### 4.2.1 GUARANTEES FOR RLHF-Sharpening WITH DIRECT PREFERENCE OPTIMIZATION

The first of our theoretical results for RLHF-Sharpening takes an offline reinforcement learning approach, whereby we implement Eq. (3) using a reward-based variant of Direct Preference Optimization (DPO) (Rafailov et al., 2023; Gao et al., 2024). Let $\mathcal{D}_{\text{pref}} = \{(x, y, y')\}$ be a dataset of $n$ examples sampled via $x \sim \mu$, $y, y' \sim \pi_{\text{base}}(y \mid x)$. For a parameter $\beta > 0$, we solve $\widehat{\pi} \in \arg\min_{\pi \in \Pi}$

$$\sum_{(x,y,y') \in \mathcal{D}_{\text{pref}}} \left( \beta \log \frac{\pi(y \mid x)}{\pi_{\text{base}}(y \mid x)} - \beta \log \frac{\pi(y' \mid x)}{\pi_{\text{base}}(y' \mid x)} - \left( \log \pi_{\text{base}}(y \mid x) - \log \pi_{\text{base}}(y' \mid x) \right) \right)^2. \quad (8)$$

**Assumptions.** Per Rafailov et al. (2023), the solution to Eq. (8) coincides with that of Eq. (2) asymptotically. To provide finite-sample guarantees, we make a number of statistical assumptions. First, we make a natural realizability assumption (e.g., Zhu et al. (2023); Xie et al. (2024)).

**Assumption 4.3** (Realizability). *The model class $\Pi$ satisfies $\pi_\beta^\star \in \Pi$.*[6]

Next, we define two concentrability coefficients for a model $\pi$:

$$\mathcal{C}_\pi = \mathbb{E}_\pi \left[ \frac{\pi(y \mid x)}{\pi_{\text{base}}(y \mid x)} \right], \quad \text{and} \quad \mathcal{C}_{\pi/\pi';\beta} := \mathbb{E}_\pi \left[ \left( \frac{\pi(y \mid x)}{\pi'(y \mid x)} \right)^\beta \right]. \quad (9)$$

The following result shows that both coefficients are bounded for the KL-regularized model $\pi_\beta^\star$.

**Lemma 4.1.** *The model $\pi_\beta^\star$ satisfies $\mathcal{C}_{\pi_\beta^\star} \leq C_{\text{cov}}$ and $\mathcal{C}_{\pi_{\text{base}}/\pi_\beta^\star;\beta} \leq |\mathcal{Y}|$.*

Motivated by this result, we assume the coefficients in Eq. (9) are bounded for all $\pi \in \Pi$.

**Assumption 4.4** (Concentrability). *All $\pi \in \Pi$ satisfy $\mathcal{C}_\pi \leq C_{\text{conc}}$ for a parameter $C_{\text{conc}} \geq C_{\text{cov}}$, and $\mathcal{C}_{\pi_{\text{base}}/\pi;\beta} \leq C_{\text{loss}}$ for a parameter $C_{\text{loss}} \geq |\mathcal{Y}|$.*

By Lemma 4.1, this assumption is consistent with Assumption 4.3 for reasonable bounds on $C_{\text{conc}}$ and $C_{\text{loss}}$; note that our sample complexity bounds will only incur logarithmic dependence on $C_{\text{loss}}$.

**Main result.** Our sample complexity guarantee for RLHF-Sharpening (via Eq. (8)) is as follows.

**Theorem 4.2.** *Let $\epsilon, \delta, \rho \in (0, 1)$ be given. Set $\beta \lesssim \gamma_{\text{margin}} \delta\epsilon$, and suppose that Assumptions 4.2 to 4.4 hold with parameters $C_{\text{conc}}$, $C_{\text{loss}}$, and $\gamma_{\text{margin}} > 0$. For an appropriate choice for $n$, the DPO algorithm (Eq. (8)) ensures that with probability at least $1 - \rho$, $\mathbb{P}_{x \sim \mu}[\widehat{\pi}(\boldsymbol{y}^\star(x) \mid x) \leq 1 - \delta] \leq \epsilon$, and has sample complexity*

$$m = \widetilde{O} \left( \frac{C_{\text{conc}} \log^3(C_{\text{loss}}|\Pi|\rho^{-1})}{\gamma_{\text{margin}}^2 \delta^2 \epsilon^2} \right).$$

Compared to the guarantee for SFT-Sharpening, RLHF-Sharpening learns a sharpened model with the same dependence on the accuracy $\epsilon$, but a worse dependence on $\delta$; as we primarily consider $\delta$ constant (cf. Proposition 3.1), we view this as relatively unimportant. We further remark that RLHF-Sharpening uses $N = 2$ responses per prompt, while SFT-Sharpening uses many ($N \approx C_{\text{cov}}/\epsilon$) responses but fewer prompts. Other differences include:

- RLHF-Sharpening requires the margin condition in Assumption 4.2, and has sample complexity scaling with $\gamma_{\text{margin}}^{-1}$. We believe this dependence is natural for algorithms based on reinforcement learning, as it relates suboptimality with respect to the reward function $r_{\text{self}}(y \mid x) = \log \pi_{\text{base}}(y \mid x)$ (i.e., $\mathbb{E}_{x \sim \mu} \left[ \max_{y \in \mathcal{Y}} \log \pi_{\text{base}}(y \mid x) - \mathbb{E}_{y \sim \widehat{\pi}(x)}[\log \pi_{\text{base}}(y \mid x)] \right] \leq \epsilon$, the objective minimized by reinforcement learning) to approximate sharpening error $\mathbb{P}_{x \sim \mu}[\widehat{\pi}(\boldsymbol{y}^\star(x) \mid x) \leq 1 - \delta]$. However, it is not clear if the precise dependence we pay is necessary.

- RLHF-Sharpening requires a bound on the uniform coverage parameter $C_{\text{conc}}$, which is generally larger than the parameter $C_{\text{cov}}$ required by SFT-Sharpening. We expect that this assumption can be removed by incorporating pessimism (Liu et al., 2024; Huang et al., 2024). Also, RLHF-Sharpening requires a bound on the parameter $C_{\text{loss}}$, which grants control over the (otherwise unbounded) range of the reward function $\log \pi_{\text{base}}(y \mid x)$. Since the dependence on $C_{\text{loss}}$ is only logarithmic, we view this as fairly mild. Overall, the guarantee in Theorem 4.2 may be somewhat pessimistic; it would be interesting if the result can be improved to match the sample complexity of SFT-Sharpening.

---

[6]See Remark 4.1 for a discussion of this assumption.

### 4.2.2 BENEFITS OF EXPLORATION

The guarantee in Theorem 4.2 scales with the coverage parameter $C_{\mathsf{cov}} = \mathbb{E}[1/\pi_{\mathsf{base}}(\boldsymbol{y}^{\star}(x)|x)]$, which in general is unavoidable in the sample-and-evaluate framework via our lower bound, Theorem 3.1. Although $C_{\mathsf{cov}}$ is a problem-dependent parameter, in the worst case it can be as large as $|\mathcal{Y}|$ (which is exponential in sequence length for autoregressive models). Fortunately, unlike SFT-Sharpening, the RLHF-Sharpening objective (3) is amenable to RL algorithms employing active exploration, leading to improved sample complexity when the class $\Pi$ has additional structure.

Our below guarantees for RLHF-Sharpening replace the assumption of bounded coverage with boundedness of a structural parameter for the model class $\Pi$ known as the "sequential extrapolation coefficient" (SEC) (Xie et al., 2023; 2024), which we denote by $\mathsf{SEC}(\Pi)$. The formal definition is deferred to Appendix J.2. Conceptually, $\mathsf{SEC}(\Pi)$ may thought of as a generalization of the eluder dimension (Russo & Van Roy, 2013; Jin et al., 2021). It can always be bounded by the coverability coefficient of the model class (Xie et al., 2024) and can be as large as $C_{\mathsf{conc}}$ in the worst case, so that bounds based on the SEC reflect improvements that are possible in favorable instances.

Beyond boundedness of the SEC, we require a bound on the range of the log-probabilities of $\pi_{\mathsf{base}}$.

**Assumption 4.5** (Bounded log-probabilities). *For all $\pi \in \Pi$, $(x, y) \in \mathcal{X} \times \mathcal{Y}$, $\left|\log \frac{1}{\pi_{\mathsf{base}}(y|x)}\right| \leq R_{\mathsf{max}}$.*

We expect that the dependence on $R_{\mathsf{max}}$ in our result can be replaced with $\log(C_{\mathsf{loss}})$ (Assumption 4.4), but we omit this extension to simplify presentation.

We appeal to (a slight modification of) XPO, an iterative language model alignment algorithm due to Xie et al. (2024). XPO is based on the objective in Eq. (8), but unlike DPO, incorporates a bonus term to encourage exploration to leverage **online** interaction. See Appendix J.2 for a detailed overview.

**Theorem 4.3** (Informal version of Theorem J.2). *Suppose that Assumptions 4.2 and 4.5 hold with parameters $\gamma_{\mathsf{margin}}, R_{\mathsf{max}} > 0$, and that Assumption 4.3 holds with $\beta = \gamma_{\mathsf{margin}}/(2\log(2|\mathcal{Y}|/\delta))$. For any $m \in \mathbb{N}$ and $\rho \in (0, 1)$, XPO (Algorithm 1), when configured appropriately, produces an $(\epsilon, \delta)$-sharpened model $\widehat{\pi} \in \Pi$ with probability at least $1 - \rho$, and uses sample complexity[7]*

$$m = \widetilde{O}\left(\frac{\mathsf{SEC}(\Pi) \cdot \log(|\Pi|\rho^{-1})}{\gamma_{\mathsf{margin}}^2 \delta^2 \epsilon^2}\right).$$

The takeaway from Theorem 4.3 is that there is no dependence on the coverage coefficient for $\pi_{\mathsf{base}}$. Instead, the rate depends on the complexity of exploration, as governed by the sequential extrapolation coefficient $\mathsf{SEC}(\Pi)$. We expect similar guarantees can derived for other active exploration algorithms and complexity measures (Jiang et al., 2017; Foster et al., 2021; Jin et al., 2021; Xie et al., 2023).

## 5 CONCLUSION

We view our theoretical framework for sharpening as a starting point toward a foundational understanding of self-improvement that can guide the design and evaluation of algorithms. To this end, we raise several directions for future research.

- *Representation learning.* A conceptually appealing feature of our framework is that it is agnostic to the structure of the model under consideration, but an important direction for future work is to study the dynamics of self-improvement for specific models/architectures and understand the representations that these models learn under self-training.

- *Richer forms of self-reward.* Our theoretical results study the dynamics of self-training in a stylized framework where the model uses its own log-probabilities as a self-reward. Empirical research on self-improvement leverages more sophisticated approaches (e.g., specific prompting techniques) (Huang et al., 2022; Wang et al., 2022; Bai et al., 2022b; Pang et al., 2023; Yuan et al., 2024) and it is important to understand when and how these forms of self-improvement are beneficial.

---

[7]Technically, Algorithm 1 operates in a slight generalization of the sample-and-evaluate framework (Definition 3.2), where the algorithm is allowed to query $\pi_{\mathsf{base}}(y \mid x)$ for arbitrary $x, y$. We expect that our lower bound (Theorem 3.1) can be extended to this more general framework, in which case Algorithm 1 is fundamentally using additional structure of $\Pi$ (via the SEC) to avoid dependence on $C_{\mathsf{cov}}$.

## ACKNOWLEDGMENTS

We thank Sivaraman Balakrishnan, Miro Dudík, Susan Dumais, John Langford, Qinghua Liu, and Yuda Song for helpful discussions.

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

CONTENTS OF APPENDIX

# Part I

# Additional Discussion and Results

## A    ADDITIONAL EXPERIMENTS AND DETAILS

In this section we detail the precise setup required to replicate our empirical results. All of our experiments were run either on 40G NVIDIA A100 GPUs, 192G AMD MI300X GPUs, or through the OpenAI API. We considered the following models. All models, except for `gpt-3.5-turbo-instruct`, are available on `https://huggingface.co` and we provide HuggingFace model identifiers below.

1. Phi models: We experiment with several models from the Phi family of models (Abdin et al., 2024), specifically `Phi3-Mini` ("microsoft/Phi-3-mini-4k-instruct"), `Phi3-Small` ("microsoft/Phi-3-small-8k-instruct"), `Phi3-Medium` ("microsoft/Phi-3-medium-4k-instruct"), and `Phi3.5-Mini` ("microsoft/Phi-3.5-mini-instruct").

2. `Llama3.2-3B-Instruct` ("meta-llama/Llama-3.2-3B-Instruct") (Dubey et al., 2024)

3. `Mistral-7B-Instruct-v0.3` ("mistralai/Mistral-7B-Instruct-v0.3") (Jiang et al., 2023)

4. `gpt-3.5-turbo-instruct` (Brown et al., 2020): We access this model via the OpenAI API.

5. `llama2-7b-game24-policy-hf` ("OhCherryFire/llama2-7b-game24-policy-hf"): We use the model of Wan et al. (2024), which is a Llama-2 model finetuned on the `GameOf24` task (Yao et al., 2024). We use this model only the `GameOf24` task.

We consider the following tasks:

1. `MATH`: We use the above models to generate responses to prompts from the `MATH` (Hendrycks et al., 2021), which consists of more difficult math questions. We consider "all" subsets and take the first 256 examples of the test set where the solution matches the regular expression (\d*).[8]

2. `GSM8k`: We use the above models to generate responses to prompts from the GSM-8k dataset (Cobbe et al., 2021) where the goal is to generate a correct answer to an elementary school math question. We take the first 256 examples from the test set in the main subset.[9]

3. `ProntoQA`: We use the above models to generate responses to prompts from the `ProntoQA` dataset (Saparov & He, 2023), which consists of chain-of-thought-style reasoning questions with boolean answers. We take the first 256 examples from the training set.[10]

4. `MMLU`: We use the above models to generate responses to prompts from three subsets of the `MMLU` dataset (Hendrycks et al., 2020), specifically `college_biology` (Bio),`college_physics` (Phys), and `college_chemistry` (Chem) all of which consist of multiple choice questions[11]. We take the first 256 examples of the test set.

5. `GameOf24`: We use only the model of Wan et al. (2024) (i.e., `llama2-7b-game24-policy-hf`), on the `GameOf24` task (Yao et al., 2024). The prompts are four numbers and the goal is to combine the numbers with standard arithmetic operations to reach the number '24.' Here we use both the train and test splits of the dataset.[12]

### A.1    INFERENCE-TIME VALIDATION EXPERIMENTS

To form the plots in Figure 1 and in Figures 3 and 4, for each (model, task) pair, we sampled $N$ generations per prompt with temperature 1 and returned the best of the $N$ generations according to the maximum-likelihood sharpening self-reward function $r_{\mathsf{self}}(y \mid x) = \log \pi_{\mathsf{base}}(y \mid x)$; we compare against greedy decoding as a baseline, whose accuracy is displayed in Figure 2(d).

---

[8]`https://huggingface.co/datasets/lighteval/MATH`.
[9]`https://huggingface.co/datasets/openai/gsm8k`.
[10]`https://huggingface.co/datasets/longface/prontoqa-train`.
[11]`https://huggingface.co/datasets/cais/mmlu`.
[12]`https://github.com/princeton-nlp/tree-of-thought-llm/tree/master/src/tot/data/24`

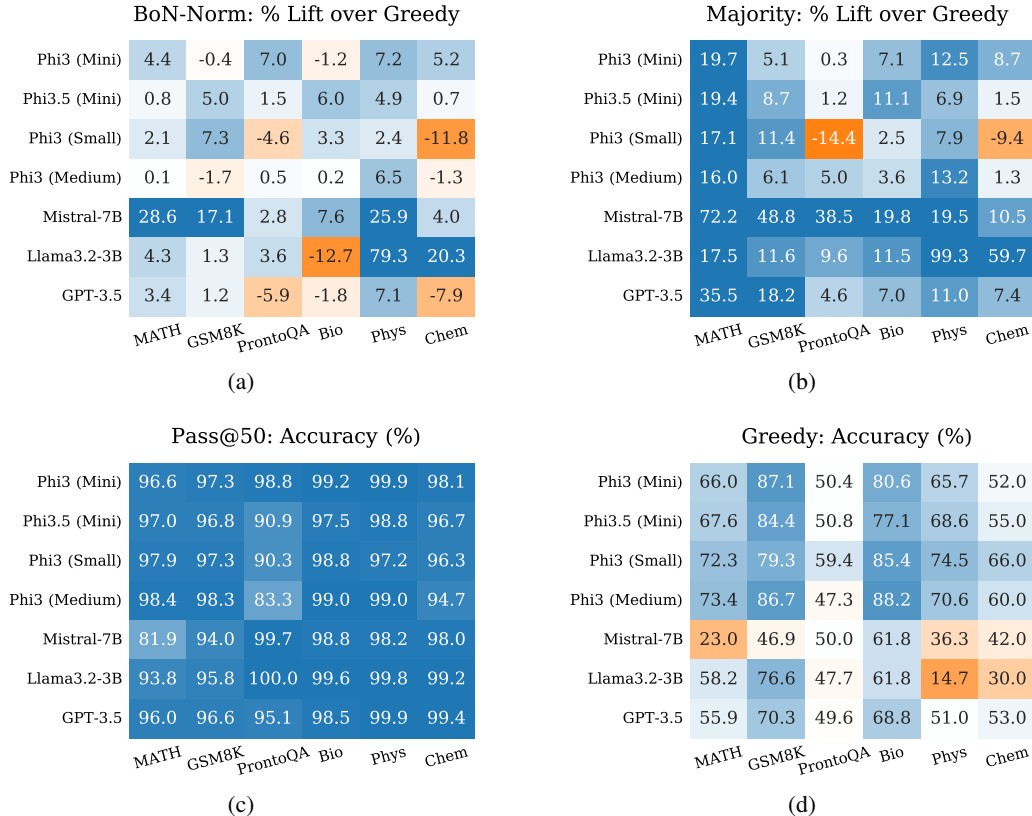

Figure 2: Performance of alternative decoding schemes beyond BoN. Percent accuracy improvement over greedy decoding for self-improvement with length-normalized log probability (a) and majority voting (b), with both demonstrating efficacy on a range of model-task pairs. (c) Measure of coverage of correct answer, demonstrating that most model-task pairs produce the correct answer most of the time with at least one completion out of 50. (d) Accuracy of greedy decoding baseline on each model-task pair.

**Implementation details.** For all models and datasets except for `GameOf24`, we used 1-shot prompting to ensure that models conform to the desired output format and to elicit chain of thought reasoning (for `GameOf24` we do not provide a demonstration in the prompt). We set the maximum length of decoding to be 512 tokens. We used 10 seeds for all (model, task) pairs with a maximum value of $N = 50$ in Best-of-$N$ sampling. We simulated $N$ responses for $N < 50$ by subsamplng the 50 generated samples. For Best-of-$N$ sampling, we always use temperature 1.0. Since greedy decoding is a deterministic strategy, we only use 1 seed for each (model, task) pair. In all experiments, we collect both the responses and their log-likelihoods under the *reference model* (i.e., the original model from which samples were generated).

**Results.** Results for most datasets are presented in Figures 3 and 4. Because we only consider a single model for `GameOf24`, we separate this task into Figure 5 For all datasets, we visualize both performance—measured as normalized improvement in accuracy over greedy decoding—and log-likelihoods—under $\pi_{\mathsf{base}}$—of the selected responses.

In all cases, Best-of-$N$ sampling (using $r_{\mathsf{self}}(y \mid x) = \log \pi_{\mathsf{base}}(y \mid x)$) improves over the naïve sampling strategy, wherein we simply sample a single generation with temperature 1.0. In all datasets, we also see improvements over the standard *greedy decoding* strategy, at least for some models. Analogously, for every model, there is at least one dataset for which Best-of-$N$ sampling improves over greedy decoding.

We further explore the relationship between sequence level log probabilities and generation quality in Figure 6, where we plot the empirical distributions of responses sampled with temperature 1 from

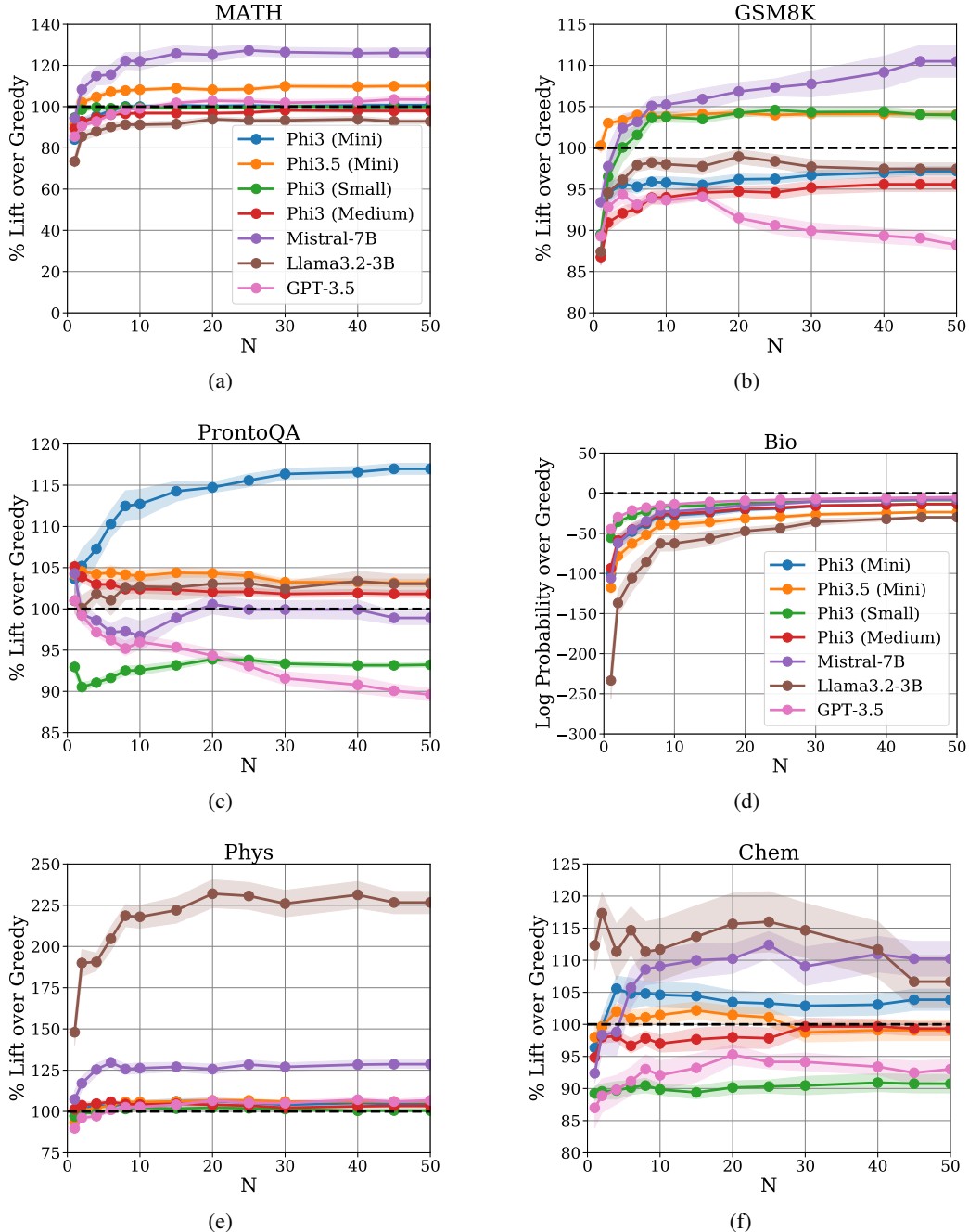

Figure 3: Percent lift in accuracy of inference-time BoN-sharpening over greedy decoding in each task as $N$ is varied. For many task-model pairs, the accuracy improves as $N$ increases, demonstrating the efficacy of maximum likelihood sharpening.

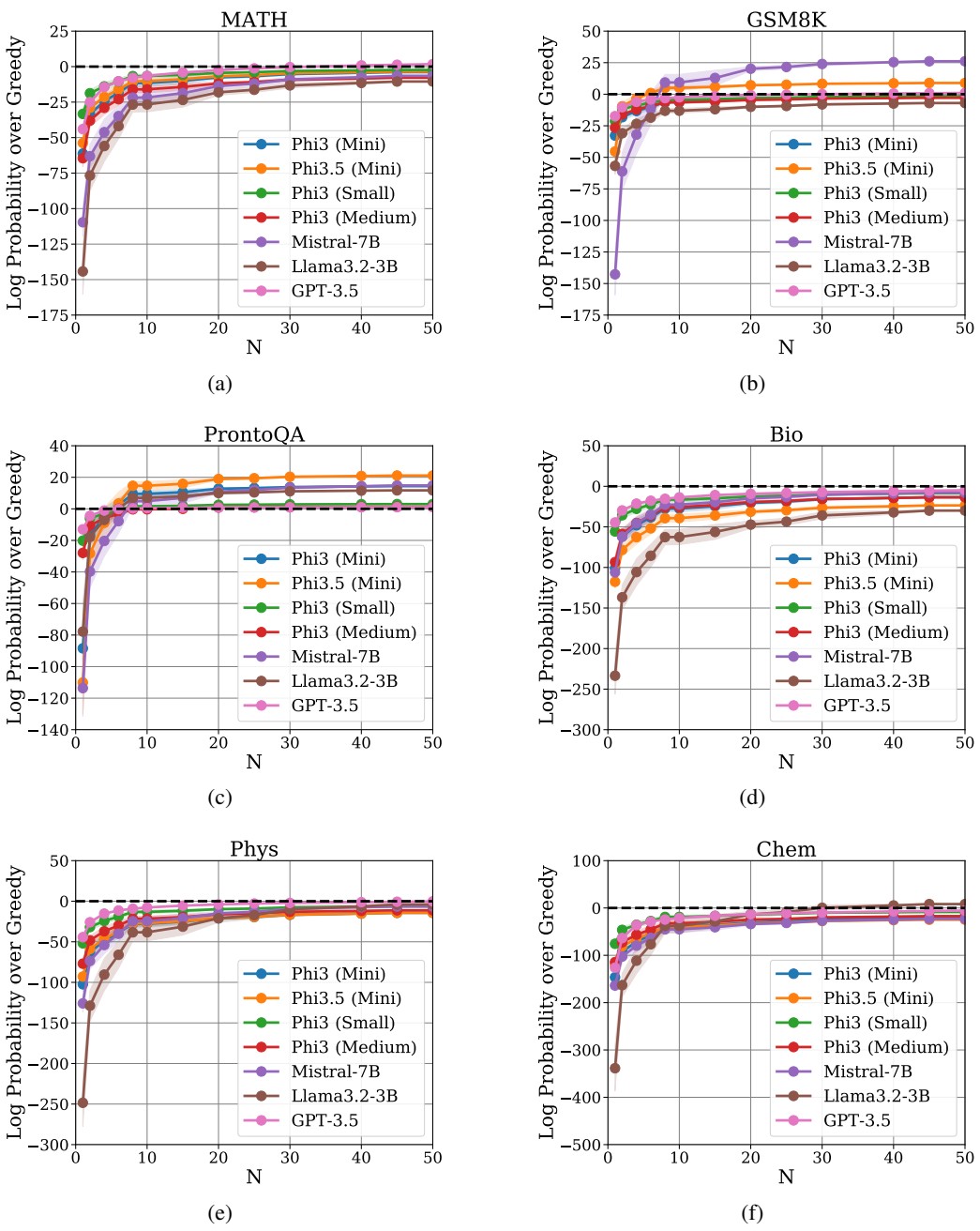

Figure 4: Effect of $N$ on average sequence level log-probabilities for inference-time BoN-sharpening on various model-task pairs, compared to greedy decoding baseline. As predicted by theory, the likelihood of sequences sampled with BoN-sharpening increases with $N$.

the base model for a variety of model-dataset pairs, conditioned on whether or not the response is correct. It is clear from the figures that the distribution of log probabilities conditioned on correctness stochastically dominates that conditioned on incorrectness in each case, which provides yet more evidence that log likelihoods represent a reasonable sel-improvement target.

We mention several other observations from the experiments. First, in most cases, performance and log-likelihood saturate at relatively small values of $N$, typically around 10 or 20. This suggests that significant improvements can be obtained with relatively low computational overhead. Second, in some cases, performance can degrade as $N$ increases. We found that this happens for two reasons: (1) the performance of the reference model is quite low and so $r_{\text{self}}$ provides a poor signal (e.g., with `Llama3.2-3B-Instruct`) and (2) the Best-of-$N$ criteria selects for short responses, which have higher log-likelihood but cannot leverage the computational/representational benefits of chain-of-thought, and thus yield worse performance (e.g., with `gpt-3.5-turbo-instruct` on GSM8k).

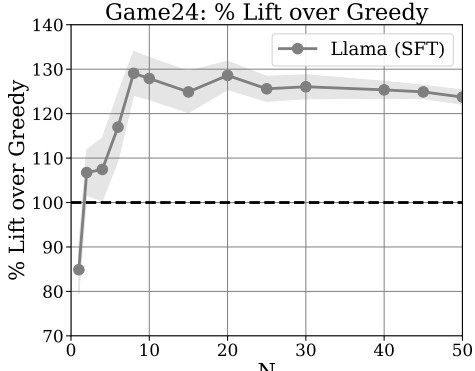
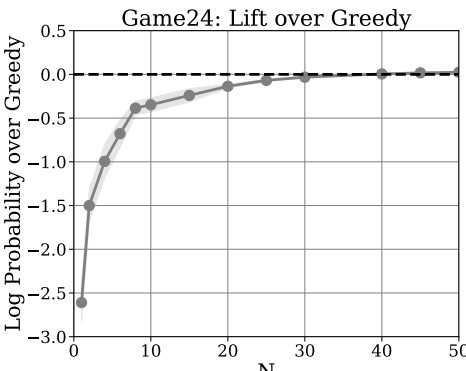

Figure 5: Effect of inference-time BoN-sharpening on `GameOf24` with the finetuned `llama2-7b-game24-policy-hf` from Wan et al. (2024).

## A.2  EXPERIMENTS WITH OTHER SELF REWARD FUNCTIONS

Although we focus on $r_{\text{self}}(y \mid x) = \log \pi_{\text{base}}(y \mid x)$ throughout the paper, the sharpening framework is significantly more general. As such, we also ran experiments with other choices for $r_{\text{self}}$, specifically:

1. Length-normalized log-likelihood: $r_{\text{self}}(y \mid x) = \log \pi_{\text{base}}(y \mid x)/|y|$ where $|y|$ is the length, in tokens, of the response.

2. Majority (self-consistency): All datasets except `GameOf24` have multiple-choice, boolean, or numerical answers. Although we allow responses to contain chain-of-thought tokens, we can extract the answer from each response and use the most-frequently-occuring answer. This can be seen as a sample-based approximation to the following self-reward function: $r_{\text{self}}(y \mid x) = \sum_{y':y'_{\text{ans}}=y_{\text{ans}}} \pi_{\text{base}}(y' \mid x)$, where $y_{\text{ans}}$ are the "answer" tokens in the full response $y$.

Finally, as a skyline we consider the *coverage* criterion (Brown et al., 2024), where we simply check if any of the sampled responses corresponds to the correct answer. This criterion is a skyline and does not fit into the self-improvement framework due to the fact that it uses knowledge of the ground truth (external) task reward function.

Results are displayed in Figure 2. For length-normalized log-likelihood and majority, we see qualitatively similar behavior to (unnormalized) log-likelihood in the sense that inference-time sharpening via these self-reward functions offers improvements over both vanilla (temperature 1.0) sampling and greedy decoding. In both cases, the improvements are generally much larger than those obtained with log-likelihood. Finally, examining the coverage criteria, we see that with $N = 50$ samples, these models almost always produce a correct answer on these tasks, raising the possibility of other self-reward functions that further improve performance.

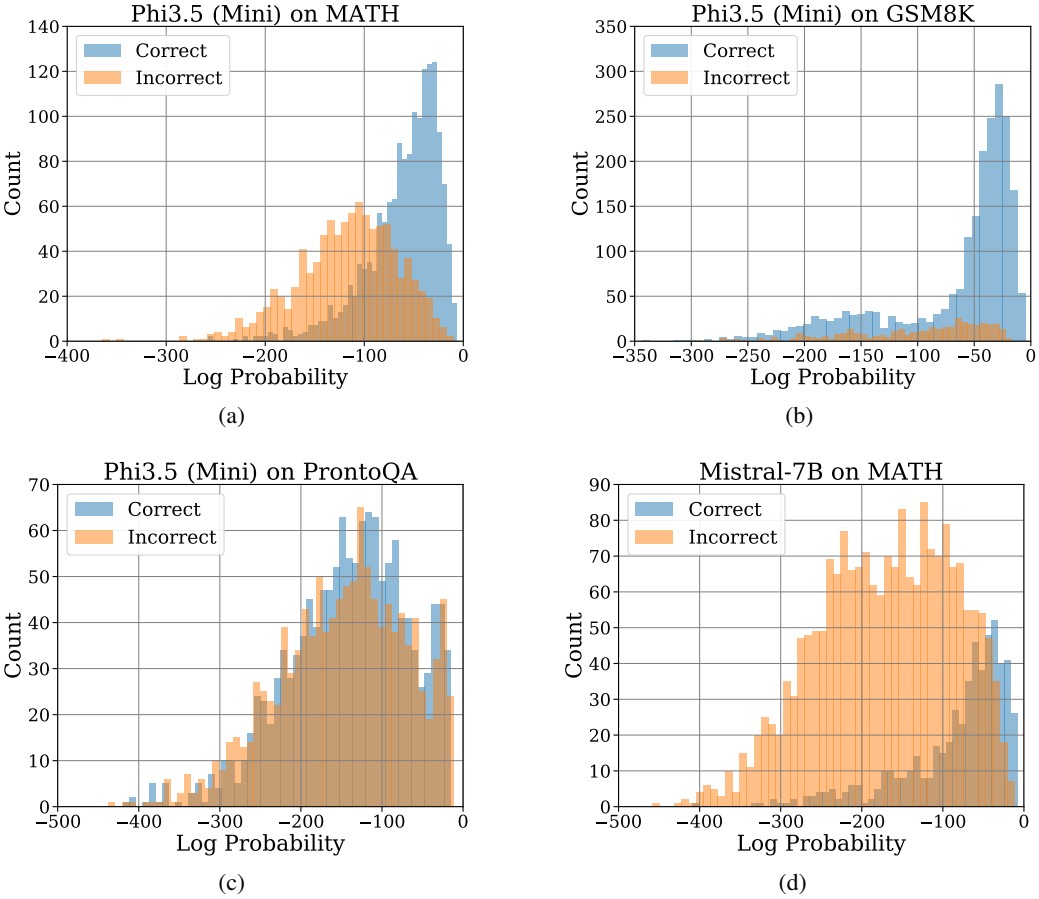

Figure 6: Distribution of sequence-level log-probabilities for responses sampled with temperature 1, conditioned on whether or not the response is correct. We consider four model-dataset pairs: (a) (Phi3.5-Mini, MATH); (b) (Phi3.5-Mini, GSM8k); (c) (Phi3.5-Mini, ProntoQA); (d) (Mistral-7B-Instruct-v0.3, MATH). In all cases except perhaps (c), conditioning on correctness of the response leads to a noticeable increase in log-probabilities, further justifying the use of sequence-level log-probabilities as a self-reward for self-improvement.

| Model | Dataset | % Lift over Greedy (Accuracy) | Lift over Greedy (Likelihood) |
|---|---|---|---|
| Phi3.5-Mini | MATH | $19.24 \pm 2.41$ | $48.33 \pm 0.17$ |
| Phi3.5-Mini | GSM8k | $1.82 \pm 0.64$ | $1.49 \pm 0.55$ |
| Phi3.5-Mini | ProntoQA | $12.46 \pm 1.08$ | $5.64 \pm 0.01$ |
| Mistral-7B | MATH | $8.88 \pm 5.55$ | $5.71 \pm 3.00$ |

Table 1: Experimental results for SFT-Sharpening

| Model | Dataset | Weight Decay | LoRA Rank |
|---|---|---|---|
| Phi3.5-Mini | MATH | 0.1 | 16 |
| Phi3.5-Mini | GSM8k | 0.5 | 16 |
| Phi3.5-Mini | ProntoQA | 0.0 | 16 |
| Mistral-7B-Instruct-v0.3 | MATH | 1.0 | 8 |

Table 2: Hyperparameters for SFT-Sharpening

## A.3 EFFECT OF SFT-Sharpening

In addition to inference-time experiments demonstrating the validity of the amortization objective considered in our theory, we also demonstrate empirically that amortization can be effected with SFT-Sharpening. Due to the realities of limited computational resources, we choose a strict subset of the model-task pairs considered in Appendix A.1 that have particularly promising inference-time BoN performance and apply SFT-Sharpening to amortize the inference time cost of multiple generations.

For each of the chosen model-dataset pairs (cf. Table 1), we sample $N = 50$ responses with temperature 1 for each prompt in the dataset and select the most likely (according to the relevant reference model). We then combine these likely responses with the prompts in order to form a training corpus and train a Low Rank Adaptation (Hu et al., 2021) to the model, sweeping over LoRA rank, learning rate scheduler, and weight decay in order to return the best optimized model.[13] We report the specific hyperparamters chosen in Table 2. On all models, we used a learning rate of $3 \times 10^{-4}$ with linear decay to zero and gradient clamping at 0.1.

**Results.** In Table 1 we report our results for the best model during training of each model-dataset pair, averaged across 3 random seeds, where responses are sampled with temperature 1 from the fine-tuned model. We report both the percent lift in accuracy on the dataset with respect to the greedy generation of the reference model and the increase in average sequence level log likelihood with respect to the same. In all cases, we see improvement on both metrics, demonstrating that some amortization is possible with SFT-Sharpening. In Figures 7 and 8, we display the evolution throughout training of these same metrics for each of the model-dataset pairs. While Phi3.5-Mini is quite well-behaved on MATH and ProntoQA, there appears to be a fair amount of noise in the training on GSM8k, with the log probability being a significantly less useful proxy for accuracy on this dataset than the others, as has been previously found in Block et al. (2023). In the case of Mistral-7B-Instruct-v0.3 on MATH, while we do see some improvement after sufficient training, the optimization suffers an initial substantial drop and then spends $\sim 90\%$ of the gradient steps recovering; we speculate that this is a function of insufficient hyper-parameter tuning of the optimization itself, rather than a fundamental barrier.

Finally, in Figure 9, we investigate the effect that the choice of $N$ has on SFT-Sharpening for Phi3.5-Mini on MATH. In particular, in forming our training set, we choose $N \in \{10, 25, 50\}$ and repeat the procedure described above, averaging our results over three seeds. We find that increasing $N$ leads to a modest increase in the sequence-level log-likelihood and a consequent increment in the accuracy of the fine-tuned model, in accordance with our theory.

---

[13]In all experiments involving Phi3.5-Mini we use a batch size of 4; unfortunately, due to a known numerical issue with LoRA on Mistral-7B-Instruct-v0.3 involving batch size $> 1$, we use a batch of 1 in this case. Because of this choice, instead of the 30 epochs we use to train our other models, for Mistral-7B-Instruct-v0.3, we run only 10 epochs.

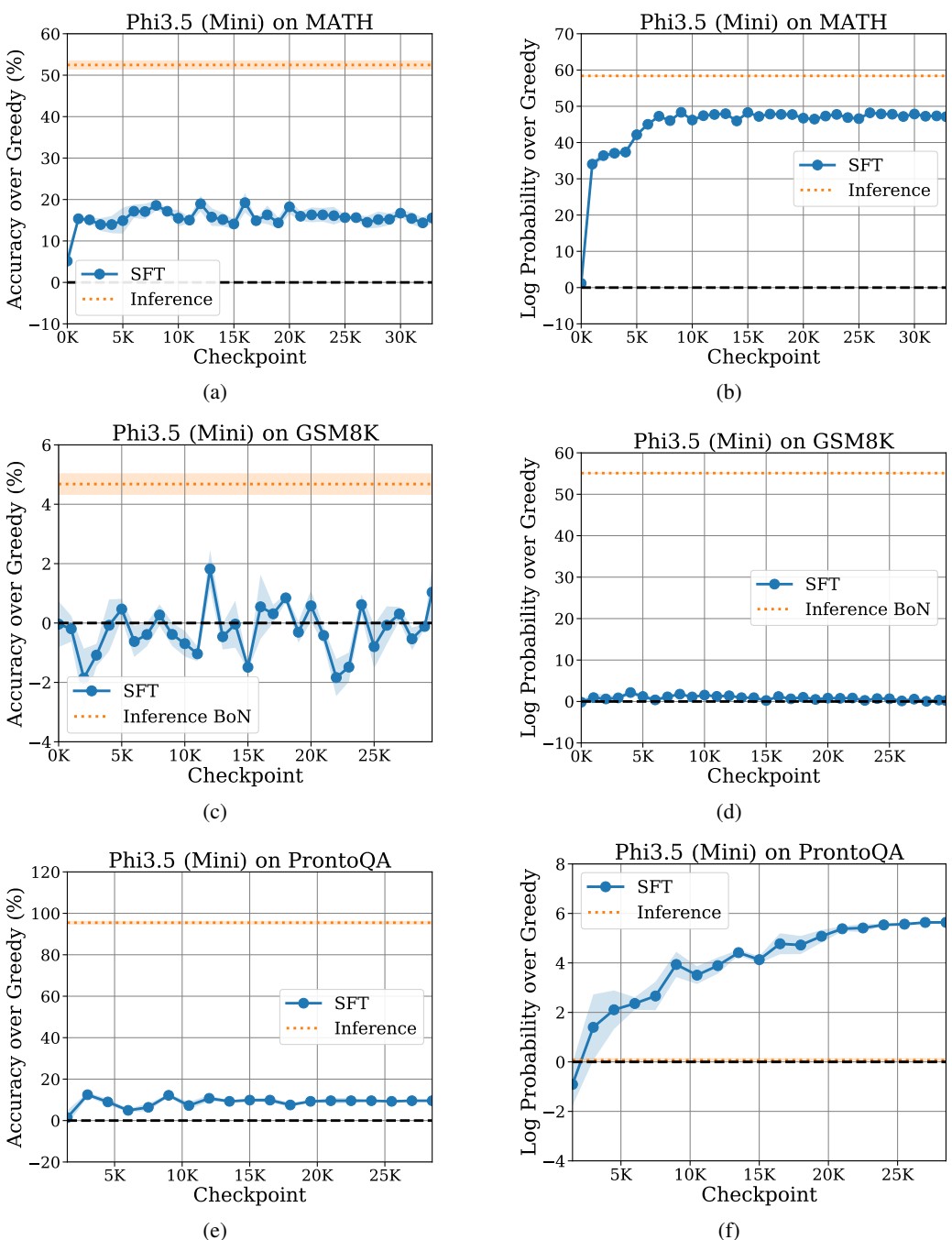

Figure 7: Evolution of Phi3.5-Mini under SFT-Sharpening ($N = 50$) on different datasets, as measured by (i) % lift over Greedy in accuracy; and (ii) difference in average sequence-level log-probability of generated responses under the reference model. The fine-tuned model learns to produce generations with high probability under the reference model, and consequently enjoys an increase in accuracy compared to the base model. However, the model does not fully reach the performance of inference-time BoN sharpening.

## B  DETAILED DISCUSSION OF RELATED WORK

In this section, we discuss related work in greater detail, including relevant works not already covered.

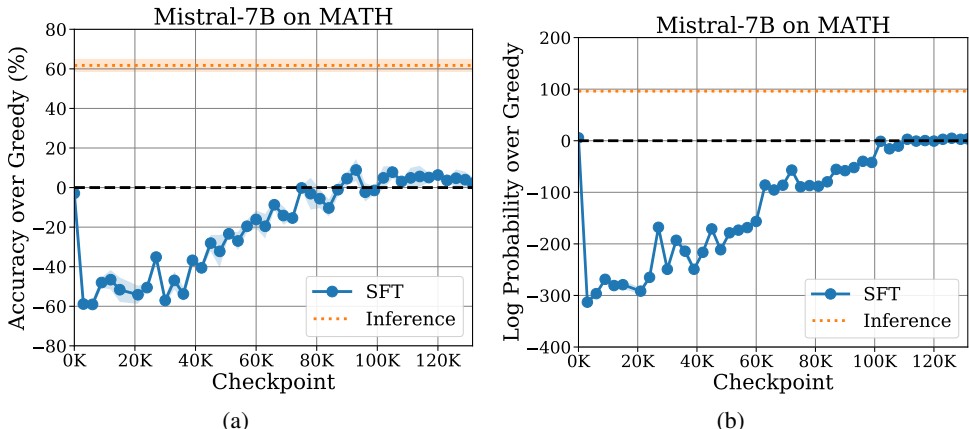

Figure 8: Evolution of `Mistral-7B-Instruct-v0.3` under `SFT-Sharpening` ($N = 50$) on `MATH`, as measured by (i) % lift over Greedy in accuracy; and (ii) difference in average sequence-level log-probability of generated responses under the reference model.

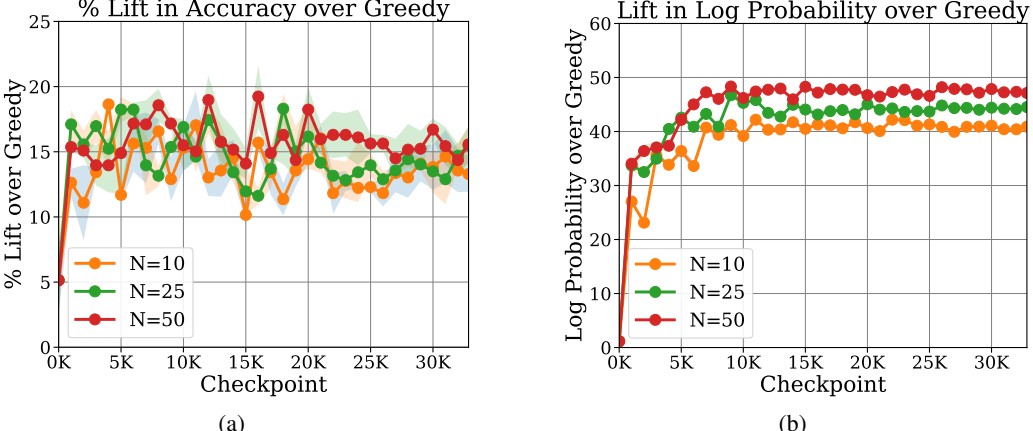

Figure 9: Effect of $N$ on `SFT-Sharpening` for `Phi3.5-Mini` on `MATH`. We report (a) % lift in accuracy over greedy; and (b) lift in sequence-level log-likelihood (averaged over the dataset). In both cases, we see that increasing $N$ leads to greater lift, in accordance with theory.

**Self-improvement and self-training.** Our work is most directly related to a growing body of empirical research that studies self-improvement/self-training for language models in a supervision-free setting in which there is no external feedback (Huang et al., 2022; Wang et al., 2022; Bai et al., 2022b; Pang et al., 2023), and takes a first step toward providing a theoretical understanding for these methods. There is also a closely related body of research on "LLM-as-a-Judge" techniques, which investigates approaches to designing self-reward functions $r_{\text{self}}$, often based on specific prompting techniques (Zheng et al., 2024; Yuan et al., 2024; Wu et al., 2024a; Wang et al., 2024).

A somewhat complementary line of research develops algorithms based on self-training and self-play (Zelikman et al., 2022; Chen et al., 2024; Wu et al., 2024c; Qu et al., 2024), but leverages various forms of external feedback (e.g., positive examples for SFT or explicit reward signal). These methods typically outperform feedback-free self-improvement methods (Zelikman et al., 2022). However, in many scenarios, obtaining external feedback can be costly or laborious; it may require collecting high-quality labeled/annotated data, rewriting examples in a formal language, etc. Thus, these two approaches are not directly comparable.

We also mention that the self-improvement problem we study is related to a classical line of research on *self-distillation* (Buciluǎ et al., 2006; Hinton et al., 2015; Devlin, 2018; Pham et al., 2021; Rizve et al., 2021), but this specific form of self-training has received limited investigation in the context of language modeling.

**Entropy minimization.** Sharpening is also closely related to a line of work on *entropy minimization* or *minimum entropy regularization*, where we seek models that have high predictive accuracy and low entropy/uncertainty. This line of work originated in the semi-supervised learning literature (Grandvalet & Bengio, 2004) and was popularized as a test-time adaptation method in computer vision (c.f., Wang et al., 2020; Press et al., 2024). Maximum-likelihood sharpening, especially via RL, is closely related in that Eq. (3) with $\beta \to 0$ and $r_{\text{self}} = \log \pi_{\text{base}}$ maximizes $\mathbb{E}_\pi[\log \pi_{\text{base}}(y \mid x)]$ rather than $-H(\pi) = \mathbb{E}_\pi[\log \pi(y \mid x)]$. (It is important that the latter is optimized continuously with $\pi_{\text{base}}$ as an initialization, but when this is done it can be seen to sharpen $\pi_{\text{base}}$, at least heuristically.) Prior work in this direction is largely empirical, focused on computer vision domains with small output spaces $\mathcal{Y}$, and hence studies statistical benefits of entropy minimization. In contrast, we initiate a theoretical study of sharpening, are primarily motivated by applications to language modeling with exponentially large output spaces, and view sharpening primarily as a computational phenomena. However, it would be interesting to understand whether statistical benefits observed in computer vision translate to the language modeling setting.

**Alignment and RLHF.** The specific algorithms for self-improvement/sharpening we study can be viewed as special cases of standard alignment algorithms, including classical RLHF methods (Christiano et al., 2017; Bai et al., 2022a; Ouyang et al., 2022), direct alignment (Rafailov et al., 2023), and (inference-time or training-time) best-of-$N$ methods (Amini et al., 2024; Sessa et al., 2024; Gui et al., 2024; Pace et al., 2024). However, the maximum likelihood sharpening objective (2) used for our theoretical results has been relatively unexplored within the alignment literature.

**Inference-time decoding.** Many inference-time decoding strategies such as greedy/low-temperature decoding, beam-search (Meister et al., 2020), and chain-of-thought decoding (Wang & Zhou, 2024) can be viewed as instances of inference-time sharpening for specific choices of the self-reward function $r_{\text{self}}$. More sophisticated inference-time search strategies such tree search and MCTS (Yao et al., 2024; Wan et al., 2024; Mudgal et al., 2023; Zhao et al., 2024) are also related, though this line of work frequently makes use of external reward signals or verification, which is somewhat complementary to our work.

**Theoretical guarantees for self-training.** On the theoretical side, current understanding of self-training is limited. One line of work, focusing on the *self-distillation* objective (Hinton et al., 2015) for binary classification and regression, aims to provide convergence guarantees for self-training in stylized setups such as linear models (Mobahi et al., 2020; Das & Sanghavi, 2023; Das et al., 2024; Pareek et al., 2024), with Allen-Zhu & Li (2020) giving guarantees for feedforward neural networks. Perhaps most closely related to our work is Frei et al. (2022), who show that self-training on a model's pseudo-labels can amplify the margin for linear logistic regression. However, to the best of our knowledge, our work is the first to study self-training in a general framework that subsumes language modeling.

Our results for `RLHF-Sharpening` are related to a body of work that provides sample complexity guarantees for alignment methods (Zhu et al., 2023; Xiong et al., 2023; Ye et al., 2024; Huang et al., 2024; Liu et al., 2024; Song et al., 2024; Xie et al., 2024), but our results leverage the structure of the maximum-likelihood sharpening self-reward function $r_{\text{self}}(y \mid x) = \log \pi_{\text{base}}(y \mid x)$, and provide guarantees for the sharpening objective in Definition 3.1 instead of the usual notion of reward suboptimality used in reinforcement learning theory.

Lastly, we mention that our results—particularly our *amortization* perspective on self-improvement—are related to work that studies representational advantages afforded by additional inference time (Malach, 2023; Li et al., 2024). These work focus on truly sequential tasks, while our work focuses on the complementary question of amortizing *parallel* computation. Thus the representational implications are quite different.

**Optimization versus sampling.** The maximum-likelihood sharpening objective we introduce in Section 3 connects the study of *self-improvement* to a large body of research in theoretical computer science on computational tradeoffs (e.g., separations and equivalences) between optimization and sampling (Barahona, 1982; Kirkpatrick et al., 1983; Lovász & Vempala, 2006; Singh & Vishnoi, 2014; Ma et al., 2019; Talwar, 2019; Eldan et al., 2022). On the one hand, this line of research highlights that there exist natural classes of distributions for which sampling is tractable, yet maximum likelihood optimization is intractable, and vice-versa. On the other hand, various works in this line of research also demonstrate *computational reductions* between optimization and sampling, whereby optimization can be reduced to sampling and vice-versa.

Our setting indeed includes natural model classes where one should not expect there to be a computational reduction from optimization ($\arg\max_{y \in \mathcal{Y}} \pi_{\text{base}}(y \mid x)$) to sampling ($y \sim \pi_{\text{base}}(\cdot \mid x)$), and hence inference-time sharpening is computationally intractable (Proposition E.1). Of course, coverage assumptions eliminate this intractability. For training-time sharpening (where the goal is to *amortize* across prompts by training a sharpened model, as formulated in Section 3) the obstacle in natural, concrete model classes is not just computational but in fact *representational* (Proposition E.2). Regarding the latter point, we note that while amortized Bayesian inference has received extensive investigation empirically (Beal, 2003; Gershman & Goodman, 2014; Swersky et al., 2020; Bengio et al., 2021; Hu et al., 2023), we are unaware of theoretical guarantees outside of this work.

## C  GUARANTEES FOR INFERENCE-TIME SHARPENING

In this section, we give theoretical guarantees for the inference-time best-of-$N$ sampling algorithm for sharpening described in Section 3.1, under the maximum-likelihood sharpening self-reward function

$$r_{\text{self}}(y \mid x; \pi_{\text{base}}) = \log \pi_{\text{base}}(y \mid x).$$

Recall that given a prompt $x \in \mathcal{X}$, the inference-time best-of-$N$ sampling algorithm draws $N$ responses $y_1, \ldots, y_n \sim \pi_{\text{base}}(\cdot \mid x)$, then return the response $\widehat{y} = \arg\max_{y_i} \log \pi_{\text{base}}(y_i \mid x)$. We show that this algorithm returns an approximate maximizer for the maximum-likelihood sharpening objective whenever the base policy $\pi_{\text{base}}$ has sufficient coverage. For a parameter $\gamma \in [0, 1)$ we define

$$\boldsymbol{y}_\gamma^\star(x) := \left\{ y \mid \pi_{\text{base}}(y \mid x) \geq (1 - \gamma) \cdot \max_{y \in \mathcal{Y}} \pi_{\text{base}}(y \mid x) \right\}$$

as the set of $(1 - \gamma)$-approximate maximizers for $\log \pi_{\text{base}}(y \mid x)$ (see Appendix F.1 for background on $\boldsymbol{y}_\gamma^\star(x)$).

**Proposition C.1.** *Let a prompt $x \in \mathcal{X}$ be given. For any $\rho \in (0, 1)$ and $\gamma \in [0, 1)$, as long as*

$$N \geq \frac{\log(\rho^{-1})}{\pi_{\text{base}}(\boldsymbol{y}_\gamma^\star(x) \mid x)},$$

*inference-time best-of-$N$ sampling produces a response $\widehat{y} \in \boldsymbol{y}_\gamma^\star(x)$ with probability at least $1 - \rho$.*

**Proof of Proposition C.1.** Fix a prompt $x \in \mathcal{X}$, failure probability $\rho \in (0, 1)$, and parameter $\gamma \in (0, 1)$. By definition of the set $\boldsymbol{y}_\gamma^\star(x)$, $\widehat{y} \in \boldsymbol{y}_\gamma^\star(x)$ if and only if there exists $i \in [N]$ such that $y_i \in \boldsymbol{y}_\gamma^\star(x)$. The complement of this event, i.e., that $y_i \notin \boldsymbol{y}_\gamma^\star(x)$ for all $i \in [N]$, has probability

$$\mathbb{P}\big(y_i \notin \boldsymbol{y}_\gamma^\star(x), \forall i \in [N]\big) = \big(1 - \pi_{\text{base}}(\boldsymbol{y}_\gamma^\star(x) \mid x)\big)^N.$$

Rearranging the right-hand side, we have

$$\left(1 - \pi_{\mathsf{base}}(\boldsymbol{y}_\gamma^\star \mid x)\right)^N = \exp\left(-N \log\left(\frac{1}{1 - \pi_{\mathsf{base}}(\boldsymbol{y}_\gamma^\star \mid x)}\right)\right) \leq \exp\left(-N \cdot \pi_{\mathsf{base}}(\boldsymbol{y}_\gamma^\star \mid x)\right),$$

since $\log(x) \geq 1 - \frac{1}{x}$ for $x > 0$, which implies that $\log\left(\frac{1}{1-\pi_{\mathsf{base}}(\boldsymbol{y}_\gamma^\star|x)}\right) \geq \pi_{\mathsf{base}}(\boldsymbol{y}_\gamma^\star \mid x)$. Thus, as long as $N \geq \frac{\log(\rho^{-1})}{\pi_{\mathsf{base}}(\boldsymbol{y}_\gamma^\star|x)}$, we have

$$\mathbb{P}\left(y_i \notin \boldsymbol{y}_\gamma^\star(x), \forall i \in [N]\right) \leq \exp\left(-N \cdot \pi_{\mathsf{base}}(\boldsymbol{y}_\gamma^\star \mid x)\right) \leq \exp(-\log(\rho^{-1})) = \rho.$$

We conclude that with probability at least $1 - \rho$, there exists $i \in [N]$ such that $y_i \in \boldsymbol{y}_\gamma^\star(x)$, and $\widehat{y} \in \boldsymbol{y}_\gamma^\star(x)$ as a result.

$\square$

## D  GUARANTEES FOR SFT-Sharpening WITH ADAPTIVE SAMPLING

SFT-Sharpening is a simple and natural self-training scheme, and converges to a sharpened policy as $n, N \to \infty$. However, using a fixed response sample size $N$ may be wasteful for prompts where the model is confident. To this end, in this section we introduce and analyze, a variant of SFT-Sharpening based on *adaptive sampling*, which adjusts the number of sampled responses adaptively.

**Algorithm.**   We present the adaptive SFT-Sharpening algorithm only for the special case of the maximum likelihood sharpening self-reward. Let a *stopping parameter* $\mu > 0$ be given. For $x_i \in \mathcal{X}$, and $y_{i,1}, y_{i,2} \ldots \sim \pi_{\mathsf{base}}(\cdot \mid x_i)$, define a stopping time (e.g., Benjamini & Hochberg (1995)) via:

$$N_\mu(x_i) := \inf\left\{k : \frac{1}{\max_{1 \leq j \leq k} \pi_{\mathsf{base}}(y_{i,j} \mid x_i)} \leq \frac{k}{\mu}\right\}. \tag{10}$$

The adaptive SFT-Sharpening algorithm computes adaptively sampled responses $y_i^{\mathsf{AdaBoN}}$ via

$$y_i^{\mathsf{AdaBoN}} \sim \arg\max\left\{\log \pi_{\mathsf{base}}(y_{i,j} \mid x_i) \mid y_{i,1}, \ldots, y_{i,N_\mu(x_i)}\right\},$$

then trains the sharpened model through SFT:

$$\widehat{\pi}^{\mathsf{AdaBoN}} = \arg\max_{\pi \in \Pi} \sum_{i=1}^n \log \pi(y_i^{\mathsf{AdaBoN}} \mid x_i).$$

Critically, by using scheme in Eq. (10), this algorithm can stop sampling responses for the prompt $x_i$ if it becomes clear that the confidence is large.

**Theoretical guarantee.**   We now show that adaptive SFT-Sharpening enjoys provable benefits over its non-adaptive counterpart through the dependence on the accuracy parameter $\epsilon > 0$.

Given $x \in \mathcal{X}$, and $y_1, y_2 \ldots \sim \pi_{\mathsf{base}}(x)$, let $N_\mu(x) := \inf\{k : \frac{1}{\max_{1 \leq i \leq k} \pi_{\mathsf{base}}(y_i|x)} \leq k/\mu\}$, and define a random variable $y^{\mathsf{AdaBoN}}(x) \sim \arg\max\{\log \pi_{\mathsf{base}}(y_i \mid x) \mid y_1, \ldots, y_{N_\mu} \sim \pi_{\mathsf{base}}(x)\}$. Let $\pi_\mu^{\mathsf{AdaBoN}}(x)$ denote the distribution over $y^{\mathsf{AdaBoN}}(x)$. We make the following realizability assumption.

**Assumption D.1.** *The model class* $\Pi$ *satisfies* $\pi_\mu^{\mathsf{AdaBoN}} \in \Pi$.

Compared to SFT-Sharpening, we require a somewhat stronger coverage coefficient given by

$$\overline{C}_{\mathsf{cov}} = \mathbb{E}_{x \sim \mu}\left[\frac{1}{\max_{y \in \mathcal{Y}} \pi_{\mathsf{base}}(y \mid x)}\right].$$

This definition coincides with Eq. (6) when the arg-max response is unique, but is larger in general.

Our main theoretical guarantee for adaptive SFT-Sharpening is as follows.

**Theorem D.1.** *Let $\delta, \rho \in (0,1)$ be given. Set $\mu = \ln(2\delta^{-1})$, and assume [Assumption D.1] holds. Then with probability at least $1 - \rho$, the adaptive* SFT-Sharpening *algorithm has*

$$\mathbb{P}_{x \sim \mu}[\widehat{\pi}(\boldsymbol{y}^\star(x) \mid x) \leq 1 - \delta] \lesssim \frac{\log(|\Pi|\rho^{-1})}{\delta n},$$

*and has sample complexity $\mathbb{E}[m] = n \cdot \overline{C}_{\mathsf{cov}} \log(\delta^{-1})$. Taking $n \gtrsim \frac{\log(|\Pi|\rho^{-1})}{\delta\epsilon}$ ensures that with probability at least $1 - \rho$, $\mathbb{P}_{x \sim \mu}[\widehat{\pi}(\boldsymbol{y}^\star(x) \mid x) \leq 1 - \delta] \leq \epsilon$, and gives total sample complexity*

$$\mathbb{E}[m] = O\left(\frac{\overline{C}_{\mathsf{cov}} \log(|\Pi|\rho^{-1}) \log(\delta^{-1})}{\delta\epsilon}\right).$$

Compared to the result for SFT-Sharpening in [Theorem 4.1], this shows that adaptive SFT-Sharpening achieves sample complexity scaling with $\frac{1}{\epsilon}$ instead of $\frac{1}{\epsilon^2}$. We believe the dependence on $\overline{C}_{\mathsf{cov}}$ for this algorithm is tight, as the adaptive stopping rule used in the algorithm can be overly conservative when $|\boldsymbol{y}^\star(x)|$ is large.

**A matching lower bound.** We now prove a complementary lower bound, which shows that the $\epsilon$-dependence in [Theorem D.1] is tight. To do so, we consider the following adaptive variant of the sample-and-evaluate framework.

**Definition D.1** (Adaptive sample-and-evaluate framework)**.** *In the **Adaptive Sample-and-Evaluate** framework, the learner is allowed to sample $n$ prompts $x \sim \mu$, and sample an arbitrary, adaptively chosen number of samples $y_1, y_2, \cdots \sim \pi_{\mathsf{base}}(\cdot \mid x)$ before sampling a new prompt $x' \sim \mu$. In this framework we define sample complexity $m$ as the total number of pairs $(x, y)$ sampled by the algorithm, which is a random variable.*

Our main lower bound is as follows.

**Theorem D.2** (Lower bound for sharpening under adaptive sampling)**.** *Fix an integer $d \geq 1$ and parameters $\epsilon \in (0,1)$ and $C \geq 1$. There exists a class of models $\Pi$ such that (i) $\log|\Pi| \approx d(1 + \log(C\epsilon^{-1}))$, (ii) $\sup_{\pi \in \Pi} C_{\mathsf{cov}}(\pi) \lesssim C$, and (iii) $\boldsymbol{y}^\pi(x)$ is a singleton for all $\pi \in \Pi$, for which any sharpening algorithm $\widehat{\pi}$ in the adaptive sample-and-evaluate framework that achieves $\mathbb{E}[\mathbb{P}_{x \sim \mu}[\widehat{\pi}(\boldsymbol{y}^{\pi_{\mathsf{base}}}(x) \mid x) > 1/2]] \geq 1 - \epsilon$ for all $\pi_{\mathsf{base}} \in \Pi$ must collect a total number of samples $m = n \cdot N$ at least*

$$\mathbb{E}[m] \gtrsim \frac{C \log|\Pi|}{\epsilon \cdot (1 + \log(C\epsilon^{-1}))}.$$

[Theorem D.2] is a special case of a more general theorem, [Theorem 3.1′], which is stated and proven in [Appendix H].

## E   COMPUTATIONAL AND REPRESENTATIONAL CHALLENGES IN SHARPENING

In this section, we make several basic observations about the inherent computational and representational challenges of maximum-likelihood sharpening. First, in [Appendix E.1], we focus on computational challenges, and show that computing a sharpened response for a given prompt $x$ can be computationally intractable in general, even when sampling $y \sim \pi_{\mathsf{base}}(\cdot \mid x)$ can be performed efficiently. Then, in [Appendix E.2], we shift our focus to representational challenges, and show that even if $\pi_{\mathsf{base}}$ is an autoregressive model, the "sharpened" version of $\pi_{\mathsf{base}}$ may not be representable as an autoregressive model with the same architecture. These results motivate the statistical assumptions (coverage and realizability) made in our analysis of SFT-Sharpening and RLHF-Sharpening in [Section 4].

To make the results in this section precise, we work in perhaps the simplest special case of autoregressive language modelling, where the model class consists of *multi-layer linear softmax models*. Formally, let $\mathcal{X}$ be the space of prompts, and let $\mathcal{Y} := \mathcal{V}^H$ be the space of responses, where $\mathcal{V}$ is the vocabulary space and $H$ is the horizon. For a collection of fixed/known $d$-dimensional feature mappings $\phi_h : \mathcal{X} \times \mathcal{V}^h \to \mathbb{R}^d$ and a norm parameter $B$, we define the model class $\Pi_{\phi, B, H}$ as the set of models

$$\pi_\theta(y_{1:H} \mid x) = \prod_{h=1}^{H} \pi_{\theta_h}(y_h \mid x, y_{1:h-1}) \tag{11}$$

where

$$\pi_\theta(y_h \mid x, y_{1:h-1}) \propto \exp(\langle \phi(x, y_{1:h}), \theta_h \rangle)$$

and $\theta = (\theta_1, \ldots, \theta_H) \in (\mathbb{R}^d)^H$ is any tuple with $\|\theta_h\|_2 \leq B$ for all $h \in [H]$.

## E.1 COMPUTATIONAL CHALLENGES

Given query access to $\phi$, for any given parameter vector $\theta$ and prompt $x$, *sampling* from a linear softmax model $\pi_\theta$ (Eq. (11)) is computationally tractable, since it only requires time $\mathrm{poly}(H, |\mathcal{V}|, d)$. Similarly, *evaluating* $\pi_\theta(y_{1:H} \mid x)$ for given prompt $x$ and response $y_{1:H}$ is computationally tractable. However, the following proposition shows that computing the sharpened response $\arg\max_{y_{1:H} \in \mathcal{V}^H} \pi_\theta(y_{1:H} \mid x)$ for a given parameter $\theta$ and response $x$ is NP-hard. Hence, even inference-time sharpening is computationally intractable in the worst case.

**Proposition E.1.** *Set $\mathcal{X} = \{\bot\}$ and $\mathcal{V} = \{-1, 1\}$. Set $d = d(H) := H + H^2 + H^3$. Identifying $[d]$ with $[H] \sqcup [H]^2 \sqcup [H]^3$, we define $\phi_h : \mathcal{X} \times \mathcal{V}^h \to \mathbb{R}^d$ by $\phi_h(\bot, y_{1:h})_i = y_i$ and $\phi_h(\bot, y_{1:h})_{(i,j)} = y_i y_j$ and $\phi_h(\bot, y_{1:h})_{(i,j,k)} = y_i y_j y_k$. There is a function $B(H) \leq \mathrm{poly}(H)$ such that the following problem is NP-hard: given $\theta = (\theta_1, \ldots, \theta_H)$ with $\max_{h \in [H]} \|\theta_h\|_2 \leq B(H)$, compute any element of $\arg\max_{y_{1:H} \in \mathcal{V}^H} \pi_\theta(y_{1:H} \mid x)$.*

Note that our results in Section 4 and Appendix C bypass this hardness through the assumption that the coverage parameter $C_{\mathsf{cov}}$ is bounded.

**Proof of Proposition E.1.** Fix $H$ and recall that $d(H) = H + H^2 + H^3$. We define three collection of basis vectors: $\{e_h\}_{h \in [H]}$ cover the first $H$ coordinates, $\{e_{(h,h')}\}_{h,h' \in [H]^2}$ cover the next $H^2$ coordinates, and $\{e_{(h,h',h'')}\}_{h,h',h'' \in [H]^3}$ cover the last $H^3$ coordinates. Suppose we define $\theta_1, \ldots, \theta_{H-2} = 0$, so that $\pi_\theta(y_h|x, y_{1:h-1}) = 1/2$ for all $1 \leq h \leq H - 2$. Define $\theta_{H-1} = \sum_{1 \leq i,j \leq H-2} J_{ij} e_{(i,j,H-1)}$ for a matrix $J \in \mathbb{R}^{(H-2) \times (H-2)}$ to be specified later, and define $\theta_H = \frac{B}{2}(e_{(H-1,H)} + e_H)$. Then $2^{H-2} \cdot \pi_\theta(y_{1:H} \mid \bot) \leq 1/2$ for any $y_{1:H}$ with $y_{H-1} = -1$ or $y_H = -1$, since this implies that $\pi_{\theta_H}(y_H \mid \bot, y_{1:H-1}) \leq 1/2$. Meanwhile, for any $y_{1:H}$ with $y_{H-1} = y_H = 1$, we have

$$2^{H-2} \cdot \pi_\theta(y_{1:H} \mid \bot) = \frac{\exp\left(\sum_{i,j \leq H-2} J_{ij} y_i y_j\right)}{\exp\left(\sum_{i,j \leq H-2} J_{ij} y_i y_j\right) + \exp\left(-\sum_{i,j \leq H-2} J_{ij} y_i y_j\right)} \cdot \frac{\exp(B)}{\exp(B) + \exp(-B)}.$$

Let $G$ be any graph on vertex set $[H - 2]$ and let $J = -A(G)$ where $A(G)$ is the adjacency matrix of $G$. Then among $y_{1:H}$ with $y_{H-1} = y_H = 1$, $2^{H-2} \cdot \pi_\theta(y_{1:H} \mid \bot)$ is maximized when $y_{1:H-2}$ corresponds to a max-cut in $G$. If $G$ has an odd number of edges, then some max-cut removes strictly more than half of the edges, and for the corresponding sequence $y_{1:H}$ we have $2^{H-2} \cdot \pi_\theta(y_{1:H} \mid \bot) \geq (1/2 + \Omega(1)) \cdot (1 - \exp(-\Omega(B)))$, which is greater than $1/2$ when we take $B := H$ and $H$ is sufficiently large. Thus, computing $\arg\max_{y_{1:H} \in \mathcal{V}^H} \pi_\theta(y_{1:H} \mid \bot)$ yields a max-cut of $G$. It is well-known that computing a max-cut in a graph is NP-hard, and the assumption that $G$ has an odd number of edges is without loss of generality. $\square$

## E.2 REPRESENTATIONAL CHALLENGES

To give provable guarantees for our sharpening algorithms, we required certain *realizability* assumptions, which in particular posited that the model class actually contains a "sharpened" version of $\pi_{\mathsf{base}}$ (Assumptions 4.1 and 4.3). In the simple example of a *single-layer* linear softmax model classes (corresponding to $H = 1$ in the above definition), Assumption 4.3 is in fact satisfied, and the sharpened model can be obtained by increasing the temperature of $\pi_{\mathsf{base}}$. However, multi-layer linear softmax models with $H \gg 1$ are more realistic. The following proposition shows that as soon as $H \geq 2$, multi-layer linear softmax model classes may not be closed under sharpening. This illustrates a potential drawback of training-time sharpening compared to inference-time sharpening, which requires no realizability assumptions. It also provides a simple example where greedy decoding does not yield a sequence-level arg-max response (since increasing temperature in a multi-layer softmax model class exactly converges to the greedy decoding).

**Proposition E.2.** *Let $\mathcal{X} = \{\bot\}$, $\mathcal{V} = [n]$, and $H = d = 2$. For any $n$ sufficiently large, there is a multi-layer linear softmax policy class $\Pi_{\phi,B,H}$ and a policy $\pi_{\mathsf{base}} \in \Pi_{\phi,B,H}$ such that $y^\star_{1:H} :=$*

$\arg\max_{y_{1:H} \in \mathcal{V}^H} \pi_\theta(y_{1:H} \mid \perp)$ *is unique, but for all $B' > B$ and $\pi \in \Pi_{\phi,B',H}$, it holds that*
$\pi(y_{1:H}^\star \mid \perp) \le 1/2$.

**Proof of Proposition E.2.** Throughout, we omit the dependence on the prompt $\perp$ for notational clarity. Since $H = 2$, the model class consists of models $\pi_\theta$ of the form

$$\pi_\theta(a) = \pi_{\theta_1}(y_1)\pi_{\theta_2}(y_2 \mid y_1) = \frac{\exp(\langle \phi_1(y_1), \theta_1 \rangle)}{Z_{\theta_1}} \frac{\exp(\langle \phi_2(y_{1:2}), \theta_2 \rangle)}{Z_{\theta_2}(y_1)} \tag{12}$$

for $Z_{\theta_1} := \sum_{y_1 \in \mathcal{V}} \exp(\langle \phi_1(y_1), \theta_1 \rangle)$ and $Z_{\theta_2}(y_1) := \sum_{y_2 \in \mathcal{V}} \exp(\langle \phi_2(y_{1:2}), \theta_2 \rangle)$.

Define $\phi_1$ by:

$$\phi_1(i) = \begin{cases} e_1 & \text{if } i = 1 \\ e_1 & \text{if } i = 2 \\ e_2 & \text{if } i \ge 3 \end{cases}.$$

Define $\phi_2$ by:

$$\phi_2(i, j) = \begin{cases} e_1 & \text{if } i = 2, j = 1 \\ e_2 & \text{if } i = 2, j \ne 1 \\ 0 & \text{if } i \ne 2 \end{cases}.$$

Define $\pi_{\mathsf{base}} := \pi_{\theta^\star}$ where $\theta_1^\star := \theta_2^\star := B \cdot e_1$ for a parameter $B \ge \log(n)$. Then $\pi_{\mathsf{base}}(1) = \pi_{\mathsf{base}}(2)$ and $\pi_{\mathsf{base}}(i) \le e^{-B}\pi_{\mathsf{base}}(2)$ for all $i \in \{3, \ldots, n\}$. Moreover, $\pi_{\mathsf{base}}(\cdot \mid i) = \mathsf{Unif}([n])$ for all $i \ne 2$, and $\pi_{\mathsf{base}}(j \mid 2) \le e^{-B}\pi_{\mathsf{base}}(1 \mid 2)$ for all $j \ne 1$. Thus,

$$\pi_{\mathsf{base}}(2, 1) = \pi_{\mathsf{base}}(2)\pi_{\mathsf{base}}(1 \mid 2) \ge \frac{1}{2 + (n-2)e^{-B}} \cdot \frac{1}{1 + (n-1)e^{-B}} \ge \Omega(1)$$

whereas $\pi_{\mathsf{base}}(i, j) = O(1/n)$ for all $(i, j) \ne (2, 1)$. Thus, $(2, 1)$ is the sequence-level argmax for sufficiently large $n$. However, for any $\pi_\theta$ of the form described in Eq. (12), we have

$$\pi_\theta(2, 1) \le \pi_\theta(2) \le \frac{\pi_\theta(2)}{\pi_\theta(1) + \pi_\theta(2)} = \frac{1}{2}$$

since $\phi(1) = \phi(2)$. This means that there is no $B'$ for which $\Pi_{\phi,B',H}$ contains an $(\epsilon, \delta)$-sharpened policy for $\pi_{\mathsf{base}}$ for any $\delta > 1/2$. $\qquad\square$

# Part II

# Proofs

## F PRELIMINARIES

### F.1 GUARANTEES FOR APPROXIMATE MAXIMIZERS

Recall that the theoretical guarantees for sharpening algorithms in Section 4 provide convergence to the set $\boldsymbol{y}^\star(x) := \arg\max_{y \in \mathcal{Y}} \pi_{\mathsf{base}}(y \mid x)$ of (potentially non-unique) maximizers for the maximum-likelihood sharpening self-reward function $\log \pi_{\mathsf{base}}(y \mid x)$. These guarantees require that the base model $\pi_{\mathsf{base}}$ places sufficient provability mass on $\boldsymbol{y}^\star(x)$, which may not always be realistic. To address this, throughout this appendix we state and prove more general versions of our theoretical results that allow for approximate maximizers, and consequently enjoy weaker coverage assumptions

For a parameter $\gamma \in [0, 1)$ we define

$$\boldsymbol{y}^\star_\gamma(x) := \left\{ y \mid \pi_{\mathsf{base}}(y \mid x) \geq (1 - \gamma) \cdot \max_{y \in \mathcal{Y}} \pi_{\mathsf{base}}(y \mid x) \right\}$$

as the set of $(1 - \gamma)$-approximate maximizers for $\log \pi_{\mathsf{base}}(y \mid x)$. We quantify the quality of a sharpened model as follows.

**Definition F.1** (Sharpened model). *We say that a model $\widehat{\pi}$ is $(\epsilon, \delta, \gamma)$-sharpened relative to $\pi_{\mathsf{base}}$ if*

$$\mathbb{P}_{x \sim \mu}\left[\widehat{\pi}\left(\boldsymbol{y}^\star_\gamma(x) \mid x\right) \geq 1 - \delta\right] \geq 1 - \epsilon.$$

That is, an $(\epsilon, \delta, \gamma)$-sharpened policy places at least $1 - \delta$ mass on $(1 - \gamma)$-approximate arg-max responses on all but an $\epsilon$-fraction of prompts under $\mu$.

Lastly, we will make use of the following generalized coverage coefficient

$$C_{\mathsf{cov}, \gamma} = \mathbb{E}_{x \sim \mu}\left[\frac{1}{\pi_{\mathsf{base}}(\boldsymbol{y}^\star_\gamma(x) \mid x)}\right],$$

which has $C_{\mathsf{cov}, \gamma} \leq C_{\mathsf{cov}}$.

### F.2 TECHNICAL TOOLS

For a pair of probability measures $\mathbb{P}$ and $\mathbb{Q}$ with a common dominating measure $\omega$, Hellinger distance is defined via

$$D_{\mathsf{H}}^2(\mathbb{P}, \mathbb{Q}) = \int \left(\sqrt{\frac{\mathrm{d}\mathbb{P}}{\mathrm{d}\omega}} - \sqrt{\frac{\mathrm{d}\mathbb{Q}}{\mathrm{d}\omega}}\right)^2 \mathrm{d}\omega.$$

**Lemma F.1** (MLE for conditional density estimation (e.g., Wong & Shen (1995); van de Geer (2000); Zhang (2006))). *Consider a conditional density $\pi^\star : \mathcal{X} \to \Delta(\mathcal{Y})$. Let $\mathcal{D} = \{(x_i, y_i)\}_{i=1}^n$ be a dataset in which $(x_i, y_i)$ are drawn i.i.d. as $x_i \sim \mu \in \Delta(\mathcal{X})$ and $y_i \sim \pi^\star(\cdot \mid x)$. Suppose we have a finite function class $\Pi \subset (\mathcal{X} \to \Delta(\mathcal{Y}))$ such that $\pi^\star \in \Pi$. Define the maximum likelihood estimator*

$$\widehat{\pi} := \arg\max_{\pi \in \Pi} \sum_{(x,y) \in \mathcal{D}} \log \pi(y \mid x).$$

*Then with probability at least $1 - \rho$,*

$$\mathbb{E}_{x \sim \mu}\left[D_{\mathsf{H}}^2(\widehat{\pi}(\cdot \mid x), \pi^\star(\cdot \mid x))\right] \leq \frac{2 \log(|\Pi|\rho^{-1})}{n}.$$

**Lemma F.2** (Elliptic potential lemma). *Let $\lambda, K > 0$, and let $A_1, \ldots, A_T \in \mathbb{R}^{d \times d}$ be positive semi-definite matrices with $\mathrm{Tr}(A_t) \leq K$ for all $t \in [T]$. Fix $\Gamma_0 = \lambda I_d$ and $\Gamma_t = \lambda I_d + \sum_{i=1}^t A_i$ for $t \in [T]$. Then*

$$\sum_{t=1}^T \mathrm{Tr}(\Gamma_{t-1}^{-1} A_t) \leq \frac{dK \log \frac{(T+1)K}{\lambda}}{\lambda \log(1 + K/\lambda)}.$$

**Proof of Lemma F.2.** Fix $t \in [T]$. Since $\text{Tr}(A_t) \leq 1$, there is some $p_t \in \Delta(\mathbb{R}^d)$ such that $A_t = \mathbb{E}_{a \sim p_t}[aa^\top]$ and $\mathbb{P}[\|a\|_2 \leq 1] = 1$. Now observe that

$$\log \det(\Gamma_t) = \log \det(\Gamma_{t-1} + A_t)$$
$$= \log \det(\Gamma_{t-1}) + \log \det(I_d + \Gamma_{t-1}^{-1/2} A_t \Gamma_{t-1}^{-1/2})$$
$$= \log \det(\Gamma_{t-1}) + \log \det \left( \mathbb{E}_{a \sim p_t} \left[ I_d + \Gamma_{t-1}^{-1/2} aa^\top \Gamma_{t-1}^{-1/2} \right] \right)$$
$$\geq \log \det(\Gamma_{t-1}) + \mathbb{E}_{a \sim p_t} \log \det(I_d + \Gamma_{t-1}^{-1/2} aa^\top \Gamma_{t-1}^{-1/2})$$
$$= \log \det(\Gamma_{t-1}) + \mathbb{E}_{a \sim p_t} \log(1 + a^\top \Gamma_{t-1}^{-1} a).$$

Now $a^\top \Gamma_{t-1}^{-1} a \leq 1/\lambda$ with probability 1, where $\lambda = \lambda_{\min}(\Gamma_0)$. We know that $\lambda x \log(1 + 1/\lambda) \leq \log(1 + x)$ for all $x \in [0, 1/\lambda]$. Thus,

$$\log \det(\Gamma_t) \geq \log \det(\Gamma_{t-1}) + \lambda \log(1 + 1/\lambda) \mathbb{E}_{a \sim p_t} a^\top \Gamma_{t-1}^{-1} a.$$

Summing over $t \in [T]$, we get

$$\log \det(\Gamma_T) \geq \log \det(\Gamma_0) + \lambda \log(1 + 1/\lambda) \sum_{t=1}^{T} \text{Tr}(\Gamma_{t-1}^{-1} A_t).$$

Finally note that $\lambda_{\max}(\Gamma_T) \leq T + 1$ so $\log \det(\Gamma_T) \leq d \log T$, whereas $\log \det(\Gamma_0) \geq d \log \lambda$. Thus,

$$\sum_{t=1}^{T} \text{Tr}(\Gamma_{t-1}^{-1} A_t) \leq \frac{d \log \frac{T+1}{\lambda}}{\lambda \log(1 + 1/\lambda)}$$

as claimed. $\qquad \square$

**Lemma F.3** (Freedman's inequality, e.g. Agarwal et al. (2014))**.** *Let $(Z_t)_{t=1}^{T}$ be a martingale difference sequence adapted to filtration $(\mathcal{F}_t)_{t=0}^{T-1}$. Suppose that $|Z_t| \leq R$ holds almost surely for all $t$. For any $\delta \in (0, 1)$ and $\eta \in (0, 1/R)$, it holds with probability at least $1 - \delta$ that*

$$\sum_{t=1}^{T} Z_t \leq \eta \sum_{t=1}^{T} \mathbb{E}[Z_t^2 | \mathcal{F}_{t-1}] + \frac{\log(1/\delta)}{\eta}.$$

**Corollary F.1.** *Let $(Z_t)_{t=1}^{T}$ be a sequence of random variables adapted to filtration $(\mathcal{F}_t)_{t=0}^{T-1}$. Suppose that $Z_t \in [0, R]$ holds almost surely for all $t$. For any $\delta \in (0, 1)$, it holds with probability at least $1 - \delta$ that*

$$\sum_{t=1}^{T} \mathbb{E}[Z_t | \mathcal{F}_{t-1}] \leq 2 \sum_{t=1}^{T} Z_t + 4R \log(1/\delta).$$

**Proof of Corollary F.1.** Observe that for any $t \in [T]$,
$$\mathbb{E}[(Z_t - \mathbb{E}[Z_t | \mathcal{F}_{t-1}])^2 | \mathcal{F}_{t-1}] \leq \mathbb{E}[Z_t^2 | \mathcal{F}_{t-1}]$$
$$\leq R \cdot \mathbb{E}[Z_t | \mathcal{F}_{t-1}].$$

Applying Lemma F.3 to the sequence $(\mathbb{E}[Z_t | \mathcal{F}_{t-1}] - Z_t)_{t=1}^{T}$, which is a martingale difference sequence with elements supported almost surely on $[-R, R]$, we get for any $\eta \in (0, 1/R)$ that with probability at least $1 - \delta$,

$$\sum_{t=1}^{T} (\mathbb{E}[Z_t | \mathcal{F}_{t-1}] - Z_t) \leq \eta \sum_{t=1}^{T} \mathbb{E}[(Z_t - \mathbb{E}[Z_t | \mathcal{F}_{t-1}])^2 | \mathcal{F}_{t-1}] + \frac{\log(1/\delta)}{\eta}$$
$$\leq \eta R \sum_{t=1}^{T} \mathbb{E}[Z_t | \mathcal{F}_{t-1}] + \frac{\log(1/\delta)}{\eta}.$$

Set $\eta = 1/(2R)$. Simplifying gives

$$\sum_{t=1}^{T} \mathbb{E}[Z_t | \mathcal{F}_{t-1}] \leq 2 \sum_{t=1}^{T} Z_t + 4R \log(1/\delta).$$

as claimed. $\qquad \square$

## G  PROOFS FROM SECTION 3.1

**Proof of Proposition 3.1.**  We prove the result by induction. Fix $x \in \mathcal{X}$, and let $y_1^\star, \ldots, y_H^\star := y^\star(x)$. Fix $h \in [H]$, and assume by induction that $\widehat{y}_{h'} = y_{h'}^\star$ for all $h' < h$. We claim that in this case,

$$\pi_h(y_h^\star \mid \widehat{y}_1, \ldots, \widehat{y}_{h-1}, x) = \pi_h(y_h^\star \mid y_1^\star, \ldots, y_{h-1}^\star, x) > 1/2,$$

which implies that $\widehat{y}_h = y_h^\star$. To see this, we observe that by Bayes' rule,

$$\pi(y_1^\star, \ldots, y_H^\star \mid x) \leq \pi(y_1^\star, \ldots, y_h^\star \mid x)$$
$$= \prod_{h'=1}^{h} \pi_{h'}(y_{h'}^\star \mid y_1^\star, \ldots, y_{h'-1}^\star, x) \leq \pi_h(y_h^\star \mid y_1^\star, \ldots, y_{h-1}^\star, x).$$

If we were to have $\pi_h(y_h^\star \mid \widehat{y}_1, \ldots, \widehat{y}_{h-1}, x) = \pi_h(y_h^\star \mid y_1^\star, \ldots, y_{h-1}^\star, x) \leq 1/2$, it would contradict the assumption that $\pi(y_1^\star, \ldots, y_H^\star \mid x) > 1/2$. This proves the result. $\qquad\square$

## H  PROOFS FROM SECTION 3.3

Below, we state and prove a generalization of Theorems 3.1 and D.2 which allows for approximate maximizers in the sense of Definition F.1, as well as a more general coverage coefficient.

To state the result, for a model $\pi$, we define

$$\boldsymbol{y}_\gamma^\pi(x) = \left\{ y \mid \pi(y \mid x) \geq (1 - \gamma) \cdot \max_{y \in \mathcal{Y}} \pi(y \mid x) \right\}.$$

Next, for any integer $p \in \mathbb{N}$, we define

$$C_{\mathsf{cov},\gamma,p}(\pi) = \left( \mathbb{E}\left[ \frac{1}{(\pi(\boldsymbol{y}_\gamma^\pi(x) \mid x))^p} \right] \right)^{1/p},$$

with the convention that $C_{\mathsf{cov},\gamma,p} = C_{\mathsf{cov},\gamma,p}(\pi_{\mathsf{base}})$. Our most general lower bound, Theorem 3.1′, holds in the regime where $\gamma = 1/2$, and thus the best response $y$ has bounded margin away from suboptimal responses.

**Theorem 3.1′** (Lower bound for sharpening). *Fix integers $d \geq 1$ and $p \geq 1$ and parameters $\epsilon \in (0, 1)$ and $C \geq 1$, and set $\gamma = 1/2$. There exists a class of models $\Pi$ such that i) $\log |\Pi| \asymp d(1 + \log(C\epsilon^{-1/p}))$, ii) $\sup_{\pi \in \Pi} C_{\mathsf{cov},\gamma,p}(\pi) \lesssim C$, and iii) $\boldsymbol{y}_\gamma^\pi(x)$ is a singleton for all $\pi \in \Pi$, for which any sharpening algorithm $\widehat{\pi}$ that attains $\mathbb{E}\left[ \mathbb{P}_{x \sim \mu}[\widehat{\pi}(\boldsymbol{y}_\gamma^{\pi_{\mathsf{base}}}(x)) > 1/2] \right] \geq 1 - \epsilon$ for all $\pi_{\mathsf{base}} \in \Pi$ must collect a total number of samples $m = n \cdot N$ at least*

$$m \gtrsim \begin{cases} \frac{C \log |\Pi|}{\epsilon^{1+1/p}(1+\log(C\epsilon^{-1/p}))} & \text{sample-and-evaluate oracle,} \\ \frac{C \log |\Pi|}{\epsilon^{1/p}(1+\log(C\epsilon^{-1/p}))} & \text{adaptive sample-and-evaluate oracle.} \end{cases}$$

**Proof of Theorem 3.1′.**  Let parameters $d, p \in \mathbb{N}$ and $\epsilon > 0$ be given, and set $\gamma = 1/2$. Let $M \in \mathbb{N}$ and $\Delta > 0$ be parameters to be chosen later. Let $\mathcal{X} = \{x_0, x_1, \ldots, x_d\}$ and $\mathcal{Y} = \{y_0, y_1, \ldots, y_M\}$ be arbitrary discrete sets (with $|\mathcal{X}| = d + 1$ and $|\mathcal{Y}| = M + 1$).

**Construction of prompt distribution and model class.**  We use the same construction for the non-adaptive and adaptive lower bounds in the theorem statement. We define the prompt distribution $\mu$ via

$$\mu := (1 - \Delta)\delta_{x_0} + \frac{\Delta}{d} \sum_{i=1}^{d} \delta_{x_i},$$

where $\delta_x$ denotes the Dirac delta distribution on element $x$.

As the first step toward constructing the model class $\Pi$, we introduce a family of distributions $(P_0, P_1, \ldots, P_M)$ on $\mathcal{Y}$ as follows

$$P_0 = \delta_{y_0}, \quad \forall i \geq 1, \ P_i = \frac{1}{(1-\gamma)M} \delta_{y_i} + \sum_{j \in [M] \setminus \{i\}} \frac{1}{M}\left(1 - \frac{\gamma}{(M-1)(1-\gamma)}\right)\delta_{y_j}.$$

Next, for or any index $\mathcal{I} = (j_1, j_2, \ldots, j_d) \in [M]^d$, define a model

$$\pi^{\mathcal{I}}(x_i) = \begin{cases} P_0 & i = 0 \\ P_{j_i} & i > 0 \end{cases}.$$

We define the model class as

$$\Pi := \{\pi^{\mathcal{I}} : \mathcal{I} \in [M]^d\},$$

which we note has

$$\log |\Pi| = d \log M.$$

**Preliminary technical results.** Define

$$\boldsymbol{y}_\gamma^{\mathcal{I}}(x) := \{y : \pi^{\mathcal{I}}(y \mid x) \geq (1 - \gamma) \max_{y \in \mathcal{Y}} \pi^{\mathcal{I}}(y \mid x)\}.$$

The following property is immediate.

**Lemma H.1.** *Let $\mathcal{I} = (j_1, \ldots, j_d) \in [d]^M$. Then $\boldsymbol{y}_\gamma^{\mathcal{I}}(x_i) = \{y_{j_i}\}$ if $i > 0$, and $\boldsymbol{y}_\gamma^{\mathcal{I}}(x_0) = \{y_0\}$.*

In view of this result, we define $y^{\mathcal{I}}(x) = \arg\max_y \pi^{\mathcal{I}}(y \mid x)$ as the unique arg-max response for $x$.

Going forward, let us fix the algorithm under consideration. Let $\mathbb{P}^{\mathcal{I}}[\cdot]$ denote the law over the dataset used by the algorithm when the true instance is $\pi^{\mathcal{I}}$ (including possible randomness and adaptivity from the algorithm itself), and let $\mathbb{E}^{\mathcal{I}}[\cdot]$ denote the corresponding expectation. The following lemma is a basic technical result.

**Lemma H.2** (Reduction to classification). *Let $\widehat{\pi}$ be the model produced by an algorithm with access to a (adaptive) sample-and-evaluate oracle for $\pi^{\mathcal{I}}$. Suppose that for some $\epsilon \geq 0$,*

$$\mathbb{E}_{\mathcal{I} \sim \mathrm{Unif}} \mathbb{E}^{\mathcal{I}} \mathbb{P}_{x \sim \mu}[\widehat{\pi}(\boldsymbol{y}_\gamma^{\mathcal{I}}(x) \mid x) > 1/2] \geq 1 - \epsilon.$$

*Define $\widehat{\mathcal{I}} = (\widehat{j}_1, \ldots, \widehat{j}_d)$ via $\widehat{j}_i = \arg\max_j \widehat{\pi}(y_j \mid x_i)$, and write $\mathcal{I} = (j_1^\star, \ldots, j_d^\star)$. Then,*

$$\frac{1}{d} \sum_{i=1}^d \mathbb{E}_{\mathcal{I} \sim \mathrm{Unif}} \mathbb{E}^{\mathcal{I}} \left[ \mathbb{I}\{\widehat{j}_i \neq j_i^\star\} \right] \leq \epsilon/\Delta.$$

**Proof of Lemma H.2.** As established in Lemma H.1, under instance $\mathcal{I}$, $\boldsymbol{y}_\gamma^{\mathcal{I}}(x_i) = \{y_{j_i^\star}\}$ for any $i \in [d]$. Thus, whenever $\widehat{\pi}(\boldsymbol{y}_\gamma^{\mathcal{I}}(x_i)) > 1/2$, $j_i^\star = \arg\max_j \widehat{\pi}(y_j \mid x_i) =: \widehat{j}_i$. The result follows by noting that the event $\{\exists i \in [d] : x = x_i\}$ occurs with probability at least $\Delta$ under $x \sim \mu$. $\qquad\square$

**Lower bound under sample-and-evaluate oracle.** Recall that in the non-adaptive framework, the sample complexity $m$ is fixed. In light of Lemma H.2, it suffices to establishes the following claim.

**Lemma H.3.** *There exists a universal constant $c > 0$ such that for all $M \geq 8$, if $m \leq cdM/\Delta$, then $\mathbb{E}_{\mathcal{I} \sim \mathrm{Unif}} \mathbb{E}^{\mathcal{I}} \left[ \mathbb{I}\{\widehat{j}_i \neq j_i^\star\} \right] \geq 1/8$ for all $i$.*

With this, the result follows by selecting $\Delta = 16\epsilon$, with which Lemma H.2 implies that any algorithm with $\mathbb{E}_{\mathcal{I} \sim \mathrm{Unif}} \mathbb{E}^{\mathcal{I}} \mathbb{P}_{x \sim \mu}[\widehat{\pi}(\boldsymbol{y}_\gamma^{\mathcal{I}}(x) \mid x) > 1/2] \geq 1 - \epsilon$ must have $m \gtrsim dM/\Delta$. To conclude, we choose $M \asymp 1 + C\epsilon^{-1/p}$, which gives $m \asymp dM/\Delta \asymp dC\epsilon^{-(1+1/p)} \asymp \epsilon^{-(1+1/p)} \log \Pi / \log(1 + C\epsilon^{1/p})$. Finally, we check that with this choice, all $\pi \in \Pi$ satisfy

$$\begin{aligned} C_{\mathsf{cov},\gamma,p}(\pi) &= (\mathbb{P}_{x \sim \mu}[x = x_0] + (M(1 - \gamma))^p \mathbb{P}_{x \sim \mu}[x \neq x_0])^{1/p} \\ &= ((1 - \Delta) + (M(1 - \gamma))^p \Delta)^{1/p} \\ &\lesssim ((1 - \Delta) + (8C(1 - \gamma))^p)^{1/p} \lesssim C. \end{aligned}$$

**Proof of Lemma H.3.** Let $i \in [d]$ be fixed. Of the $m = n \cdot N$ tuples $(x, y, \log \pi_{\mathsf{base}}(y \mid x))$ that are observed by the algorithm, let $m_i$ denote the (random) number of such examples for which $x = x_i$. From Markov's inequality, we have

$$\mathbb{P}[m_i \leq 2\Delta m/d] \geq \frac{1}{2} \tag{13}$$

Going forward, let $\mathcal{D} = \{(x, y, \log \pi_{\mathsf{base}}(y \mid x))\}$ denote the dataset collected by the algorithm, which has $|\mathcal{D}| = m$. Let $\mathcal{E}_i$ denote the event that, for prompt $x = x_i$, (i) there are at least two distinct responses $y_j$ for which $(x_i, y_j) \notin \mathcal{D}$; and (ii) there are no pairs $(x_i, y) \in \mathcal{D}$ for which $\pi_{\mathsf{base}}(y \mid x_i) > \frac{1}{M}$. Since $\mathcal{E}_i$ is a measurable function of $\mathcal{D}$, we can write

$$\mathbb{E}_{\mathcal{I} \sim \mathsf{Unif}} \mathbb{E}^{\mathcal{I}} \left[ \mathbb{I}\{\widehat{j}_i \neq j_i^\star\} \right] \geq \mathbb{E}_{\mathcal{I} \sim \mathsf{Unif}} \mathbb{E}^{\mathcal{I}} \left[ \mathbb{I}\{\widehat{j}_i \neq j_i^\star\} \cdot \mathbb{I}\{\mathcal{E}_i\} \right]$$
$$= \mathbb{E}_{\mathcal{I} \sim \mathsf{Unif}} \mathbb{E}^{\mathcal{I}} \left[ \mathbb{I}\{\mathcal{E}_i\} \mathbb{E}_{\mathcal{I} \sim \mathbb{P}[\mathcal{I} = \cdot \mid \mathcal{D}]} \left[ \mathbb{I}\{\widehat{j}_i \neq j_i^\star\} \right] \right], \quad (14)$$

where $\mathcal{I} \sim \mathbb{P}[\mathcal{I} = \cdot \mid \mathcal{D}]$ is sampled from the posterior distribution over $\mathcal{I}$ conditioned on the dataset $\mathcal{D}$. Observe that conditioned on $\mathcal{E}_i$, the posterior distribution over $j_i^\star$ under $\mathcal{I} \sim \mathbb{P}[\mathcal{I} = \cdot \mid \mathcal{D}]$ is uniform over the set of indices $j \in [M]$ for which $(x_i, y_j) \notin \mathcal{D}$, and this set has size at least 2. Hence, $\mathbb{I}\{\mathcal{E}_i\} \mathbb{E}_{\mathcal{I} \sim \mathbb{P}[\mathcal{I} = \cdot \mid \mathcal{D}]} \left[ \mathbb{I}\{\widehat{j}_i \neq j_i^\star\} \right] \geq \frac{1}{2}$, and resuming from Eq. (14), we have

$$\mathbb{E}_{\mathcal{I} \sim \mathsf{Unif}} \mathbb{E}^{\mathcal{I}} \left[ \mathbb{I}\{\widehat{j}_i \neq j_i^\star\} \right] \geq \frac{1}{2} \mathbb{E}_{\mathcal{I} \sim \mathsf{Unif}} \mathbb{E}^{\mathcal{I}} \left[ \mathbb{I}\{\mathcal{E}_i\} \right] \geq \frac{1}{2} \mathbb{E}_{\mathcal{I} \sim \mathsf{Unif}} \mathbb{P}^{\mathcal{I}} \left[ \mathcal{E}_i \cap \{m_i \leq 2\Delta m/d\} \right]$$
$$\geq \frac{1}{4} \mathbb{E}_{\mathcal{I} \sim \mathsf{Unif}} \mathbb{P}^{\mathcal{I}} \left[ \mathcal{E}_i \mid m_i \leq 2\Delta m/d \right],$$

where the last inequality is from Eq. (13). Finally, we can check that under the law $\mathbb{P}^{\mathcal{I}}$, the probability of the event $\mathcal{E}_i$—conditioned on the value $m_i$—is at least the probability that $(x_i, y_{j_i^\star}), (x_i, y_{j'}) \notin \mathcal{D}$ for an arbitrary fixed index $j' \neq j_i^\star$, which on the event $\{m_i \leq 2\Delta m/d\}$ is at least

$$\left(1 - \frac{3}{M}\right)^{m_i} \geq \left(1 - \frac{3}{M}\right)^{2\Delta m/d},$$

where we have used that $\gamma = 1/2$. The value above is at least $\frac{1}{4}$ whenever $m \leq c \cdot dM/\Delta$ for a sufficiently small absolute constant $c > 0$. For this value of $m$, we conclude that $\mathbb{E}_{\mathcal{I} \sim \mathsf{Unif}} \mathbb{E}^{\mathcal{I}} \left[ \mathbb{I}\{\widehat{j}_i \neq j_i^\star\} \right] \geq \frac{1}{4} \mathbb{E}_{\mathcal{I} \sim \mathsf{Unif}} \mathbb{P}^{\mathcal{I}} \left[ \mathcal{E}_i \mid \{m_i \leq 2\Delta m/d\} \right] \geq \frac{1}{8}$. $\qquad\square$

**Lower bound under adaptive sample-and-evaluate oracle.** In the adaptive framework, we let $m_i$ denote the (potentially random) number of tuples $(x, y, \log \pi_{\mathsf{base}}(y \mid x))$ observed by the algorithm in which $x = x_i$. Note that unlike the non-adaptive framework, the distribution over $m_i$ depends on the underlying instance $\mathcal{I}$ with which the algorithm interacts.

To begin, from Lemma H.2 and Markov's inequality, if $\widehat{\pi}$ satisfies the guarantee $\mathbb{E}_{\mathcal{I} \sim \mathsf{Unif}} \mathbb{E}^{\mathcal{I}} \mathbb{P}_{x \sim \mu}[\widehat{\pi}(\boldsymbol{y}_\gamma^{\mathcal{I}}(x)) > 1/2] \geq 1 - \epsilon$, then there exists a set of indices $S_{\mathsf{good}} \subset [d]$ such that[14]

$$|S_{\mathsf{good}}| \geq \lfloor d/2 \rfloor, \quad \forall i \in S_{\mathsf{good}}, \ \mathbb{E}_{\mathcal{I} \sim \mathsf{Unif}} \mathbb{E}^{\mathcal{I}} \left[ \mathbb{I}\{\widehat{j}_i \neq j_i^\star\} \right] \leq \frac{2\epsilon}{\Delta}. \quad (15)$$

We now appeal to the following lemma.

**Lemma H.4.** *As long as $M \geq 6$, it holds that for all $i \in [d]$,*

$$\mathbb{E}_{\mathcal{I} \sim \mathsf{Unif}} \mathbb{E}^{\mathcal{I}} \left[ \mathbb{I}\{\widehat{j}_i \neq j_i^\star\} \right] \geq \frac{1}{4e} \mathbb{E}_{\mathcal{I} \sim \mathsf{Unif}} \mathbb{E}^{\mathcal{I}} \left[ \mathbb{I}\{m_i \leq M/3\} \right].$$

Combining Lemma H.4 with Eq. (15), it follows that there exist absolute constant $c_1, c_2, c_3 > 0$ such that if $\Delta = c_1 \cdot \epsilon$, then for all $i \in S_{\mathsf{good}}$,

$$\mathbb{E}_{\mathcal{I} \sim \mathsf{Unif}} \mathbb{P}^{\mathcal{I}}[m_i \geq c_2 M] \geq c_3.$$

Thus, with this choice for $\Delta$, we have that $i \in S_{\mathsf{good}}$,

$$\mathbb{E}_{\mathcal{I} \sim \mathsf{Unif}} \mathbb{E}^{\mathcal{I}} [m_i] \gtrsim M,$$

---

[14]We emphasize that the set $S_{\mathsf{good}}$ is not a random variable, and depends only on the algorithm itself.

and we can lower bound the algorithm's expected sample complexity by summing over $i \in S_{\text{good}}$:

$$\mathbb{E}_{\mathcal{I} \sim \text{Unif}} \, \mathbb{E}^{\mathcal{I}} \, [m] \geq \mathbb{E}_{\mathcal{I} \sim \text{Unif}} \, \mathbb{E}^{\mathcal{I}} \left[ \sum_{i \in S_{\text{good}}} m_i \right] \gtrsim |S_{\text{good}}| M \gtrsim dM.$$

The result now follows by tuning $M \asymp 1 + C\epsilon^{-1/p}$ as in the proof of the lower bound for non-adaptive sampling, which gives $\mathbb{E}[m] \gtrsim dM \asymp dC\epsilon^{-1/p} \asymp \epsilon^{-1/p} \log \Pi / \log(1 + C\epsilon^{1/p})$ and $C_{\text{cov},\gamma,p}(\pi) \lesssim C$ for all $\pi \in \Pi$.

**Proof of Lemma H.4.** Let $i \in [d]$ be fixed. Let $\mathcal{D} = \{(x, y, \log \pi_{\text{base}}(y \mid x))\}$ denote the dataset collected by the algorithm at termination, which has $|\mathcal{D}| = m$. Let $\mathcal{E}_i$ denote the event that, for prompt $x = x_i$, (i) there are at least two distinct responses $y_j$ for which $(x_i, y_j) \notin \mathcal{D}$; and (ii) there are no pairs $(x_i, y) \in \mathcal{D}$ for which $\pi_{\text{base}}(y \mid x_i) > \frac{1}{M}$. Since $\mathcal{E}_i$ is a measurable function of $\mathcal{D}$, we can write

$$\mathbb{E}_{\mathcal{I} \sim \text{Unif}} \, \mathbb{E}^{\mathcal{I}} \left[ \mathbb{I}\{\widehat{j}_i \neq j_i^\star\} \right] \geq \mathbb{E}_{\mathcal{I} \sim \text{Unif}} \, \mathbb{E}^{\mathcal{I}} \left[ \mathbb{I}\{\widehat{j}_i \neq j_i^\star\} \cdot \mathbb{I}\{\mathcal{E}_i\} \right]$$
$$= \mathbb{E}_{\mathcal{I} \sim \text{Unif}} \, \mathbb{E}^{\mathcal{I}} \left[ \mathbb{I}\{\mathcal{E}_i\} \, \mathbb{E}_{\mathcal{I} \sim \mathbb{P}[\mathcal{I}=\cdot|\mathcal{D}]} \left[ \mathbb{I}\{\widehat{j}_i \neq j_i^\star\} \right] \right], \quad (16)$$

where $\mathcal{I} \sim \mathbb{P}[\mathcal{I} = \cdot \mid \mathcal{D}]$ is sampled from the posterior distribution over $\mathcal{I}$ conditioned on the dataset $\mathcal{D}$. Observe that conditioned on $\mathcal{E}_i$, the posterior distribution over $j_i^\star$ under $\mathcal{I} \sim \mathbb{P}[\mathcal{I} = \cdot \mid \mathcal{D}]$ is uniform over the set of indices $j \in [M]$ for which $(x_i, y_j) \notin \mathcal{D}$, and this set has size at least 2. Hence, $\mathbb{I}\{\mathcal{E}_i\} \, \mathbb{E}_{\mathcal{I} \sim \mathbb{P}[\mathcal{I}=\cdot|\mathcal{D}]} \left[ \mathbb{I}\{\widehat{j}_i \neq j_i^\star\} \right] \geq \frac{1}{2}$, and resuming from Eq. (16), we have

$$\mathbb{E}_{\mathcal{I} \sim \text{Unif}} \, \mathbb{E}^{\mathcal{I}} \left[ \mathbb{I}\{\widehat{j}_i \neq j_i^\star\} \right] \geq \frac{1}{2} \mathbb{E}_{\mathcal{I} \sim \text{Unif}} \, \mathbb{E}^{\mathcal{I}} \left[ \mathbb{I}\{\mathcal{E}_i\} \right]$$
$$\geq \frac{1}{2} \mathbb{E}_{\mathcal{I} \sim \text{Unif}} \, \mathbb{P}^{\mathcal{I}} \left[ \mathcal{E}_i \cap \{m_i \leq M/3\} \right]$$
$$= \frac{1}{2} \mathbb{E}_{\mathcal{I} \sim \text{Unif}} \left[ \mathbb{P}^{\mathcal{I}} \left[ \mathcal{E}_i \mid m_i \leq M/3 \right] \cdot \mathbb{P}^{\mathcal{I}}[m_i \leq M/3] \right].$$

The event $\mathcal{E}_i$ is a superset of the event $\mathcal{E}_{i,j'}$ that $(x_i, y_{j_i^\star}), (x_i, y_{j'}) \notin \mathcal{D}$ for an arbitrary fixed index $j' \neq j_i^\star$. Thus,

$$\mathbb{P}^{\mathcal{I}} \left[ \mathcal{E}_i \mid m_i \leq M/3 \right] \geq \mathbb{P}^{\mathcal{I}} \left[ \mathcal{E}_{i,j'} \mid m_i \leq M/3 \right]$$

Moreover, we can realize the law of $\mathbb{P}^{\mathcal{I}}$ considering an infinite tape, associated to index $i$, of i.i.d. samples $y \sim \pi_{\text{base}}(\cdot \mid x_i)$, and taking the first $m_i$ elements on this tape to be the samples $(x, y, \log \pi_{\text{base}}(y \mid x)) \in \mathcal{D}$ with $x = x_i$ (see, e.g. Simchowitz et al. (2017) for an argument of this form). On the event $\{m_i \leq M/3\}$, the $m_i$ samples in $(x, y, \log \pi_{\text{base}}(y \mid x)) \in \mathcal{D}$ with $x = x_i$ are a subset of the first $M/3$ samples from the index-$i$ tape. Viewed in this way, we can lower bound the probability of $\mathcal{E}_{i,j}$ by the probability of the event $\tilde{\mathcal{E}}_{i,j'}$ that the first $M/3$ $y$'s on the index-$i$ tape contain neither $j_i^\star$, nor the designated index $j'$. As these first $M/3$ $y$'s are not chosen adaptively, the probability of $\tilde{\mathcal{E}}_{i,j'}$ is at least

$$\left( 1 - \frac{3}{M} \right)^{m_i} \geq \left( 1 - \frac{3}{M} \right)^{M/3} \geq \frac{1}{2e},$$

as long as $M \geq 6$ and $\gamma = 1/2$. We conclude that

$$\mathbb{E}_{\mathcal{I} \sim \text{Unif}} \, \mathbb{E}^{\mathcal{I}} \left[ \mathbb{I}\{\widehat{j}_i \neq j_i^\star\} \right] \geq \frac{1}{4e} \mathbb{E}_{\mathcal{I} \sim \text{Unif}} \, \mathbb{E}^{\mathcal{I}} \left[ \mathbb{I}\{m_i \leq M/3\} \right].$$

$\square$

# I  PROOFS FROM SECTION 4.1 AND APPENDIX D

The following theorem is a generalization of Theorem 4.1 which allows for approximate maximizers in the sense of Definition F.1.

**Theorem 4.1'.** *Let $\rho, \delta \in (0, 1)$ be given, and suppose we set $N = N^\star \log(2\delta^{-1})$ for a parameter $N^\star \in \mathbb{N}$. Then for any $n \in \mathbb{N}$,* SFT-Sharpening *ensures that with probability at least $1 - \rho$, for any $\gamma \in (0, 1)$, the output model $\widehat{\pi}$ satisfies*

$$\mathbb{P}_{x \sim \mu}\big[\widehat{\pi}(\boldsymbol{y}_\gamma^\star(x) \mid x) \leq 1 - 2\delta\big] \lesssim \frac{1}{\delta} \cdot \frac{\log(|\Pi|\rho^{-1})}{n} + \frac{C_{\mathsf{cov},\gamma}}{N^\star}.$$

*In particular, given $(\epsilon, \delta, \gamma)$, by setting $n = C_{4.1} \frac{\log|\Pi|}{\delta\epsilon}$ and $N^\star = C_{4.1} \frac{C_{\mathsf{cov},\gamma}}{\epsilon}$ for a sufficiently large absolute constant $C_{4.1} > 0$, we are guaranteed that*

$$\mathbb{P}_{x \sim \mu}\big[\widehat{\pi}(\boldsymbol{y}_\gamma^\star(x) \mid x) \leq 1 - \delta\big] \leq \epsilon.$$

*The total sample complexity is*

$$m = O\left(\frac{C_{\mathsf{cov},\gamma} \log(|\Pi|\rho^{-1}) \log(\delta^{-1})}{\delta\epsilon^2}\right).$$

**Proof of Theorem 4.1'.** Under realizability of $\pi_N^{\mathsf{BoN}}$ (Assumption 4.1), Lemma F.1 implies that the output of SFT-Sharpening satisfies, with probability at least $1 - \rho$,

$$\mathbb{E}_{x \sim \mu}\big[D_{\mathsf{H}}^2\big(\widehat{\pi}(\cdot \mid x), \pi_N^{\mathsf{BoN}}(\cdot \mid x)\big)\big] \leq \varepsilon_{\mathsf{stat}}^2 := \frac{2\log(|\Pi|/\rho)}{n}. \tag{17}$$

Henceforth we condition on the event that Eq. (17) holds. Let

$$\mathcal{X}_{\mathsf{good}} := \left\{x \in \mathcal{X} \mid N^\star \geq \frac{1}{\pi_{\mathsf{base}}(\boldsymbol{y}_\gamma^\star(x) \mid x)}\right\}$$

denote the set of prompts for which $\pi_{\mathsf{base}}$ places sufficiently high mass on $\boldsymbol{y}_\gamma^\star(x)$. We can bound

$$\mathbb{P}_{x \sim \mu}\big[\widehat{\pi}(\boldsymbol{y}_\gamma^\star(x) \mid x) \leq 1 - \delta\big]$$
$$\leq \mathbb{P}_{x \sim \mu}\big[\widehat{\pi}(\boldsymbol{y}_\gamma^\star(x) \mid x) \leq 1 - \delta, x \in \mathcal{X}_{\mathsf{good}}\big] + \mathbb{P}_{x \sim \mu}[x \notin \mathcal{X}_{\mathsf{good}}]. \tag{18}$$

To bound the first term in Eq. (18), note that if $x \in \mathcal{X}_{\mathsf{good}}$, then $\pi_N^{\mathsf{BoN}}(\boldsymbol{y}_\gamma^\star(x) \mid x) \geq 1 - \delta/2$. Indeed, observe that $y \sim \pi_N^{\mathsf{BoN}}(\cdot \mid x) \notin \boldsymbol{y}_\gamma^\star(x)$ if and only if $y_1, \ldots, y_N \sim \pi_{\mathsf{base}}(x)$ have $y_i \notin \boldsymbol{y}_\gamma^\star(x)$ for all $i$, which happens with probability $(1 - \pi_{\mathsf{base}}(\boldsymbol{y}_\gamma^\star(x) \mid x))^N \leq (1 - 1/N^\star)^N \leq \delta/2$ since $x \in \mathcal{X}_{\mathsf{good}}$. It follows that for any such $x$, we can lower bound (using the data processing inequality)

$$D_{\mathsf{H}}^2\big(\widehat{\pi}(\cdot \mid x), \pi_N^{\mathsf{BoN}}(\cdot \mid x)\big) \geq \left(\sqrt{1 - \widehat{\pi}(\boldsymbol{y}_\gamma^\star(x) \mid x)} - \sqrt{1 - \pi_N^{\mathsf{BoN}}(\boldsymbol{y}_\gamma^\star(x) \mid x)}\right)^2$$
$$\gtrsim \delta \cdot \mathbb{I}\big\{\widehat{\pi}(\boldsymbol{y}_\gamma^\star(x) \mid x) \leq 1 - \delta\big\}. \tag{19}$$

By Eqs. (17) and (19), it follows that

$$\mathbb{P}_{x \sim \mu}\big[\widehat{\pi}(\boldsymbol{y}_\gamma^\star(x) \mid x) \leq 1 - \delta, x \in \mathcal{X}_{\mathsf{good}}\big] \lesssim \frac{\varepsilon_{\mathsf{stat}}^2}{\delta}.$$

For the second term in Eq. (18), we bound

$$\mathbb{P}_{x \sim \mu}[x \notin \mathcal{X}_{\mathsf{good}}] = \mathbb{P}_{x \sim \mu}\left[N^\star < \frac{1}{\pi_{\mathsf{base}}(\boldsymbol{y}_\gamma^\star(x) \mid x)}\right]$$
$$= \mathbb{P}_{x \sim \mu}\left[\frac{1}{N^\star \pi_{\mathsf{base}}(\boldsymbol{y}_\gamma^\star(x) \mid x)} > 1\right]$$
$$\leq \frac{1}{N^\star} \mathbb{E}_{x \sim \mu}\left[\frac{1}{\pi_{\mathsf{base}}(\boldsymbol{y}_\gamma^\star(x) \mid x)}\right]$$
$$\leq \frac{C_{\mathsf{cov},\gamma}}{N^\star}$$

via Markov's inequality and the definition of $C_{\mathsf{cov},\gamma}$. Substituting both bounds into Eq. (18) completes the proof. $\qquad\square$

**Proof of Theorem D.1.** The proof begins similarly to Theorem 4.1. By realizability of $\pi_{N_\mu}$, Lemma F.1 implies that the output of SFT-Sharpening satisfies, with probability at least $1 - \rho$,

$$\mathbb{E}_{x \sim \mu}\big[D_{\mathsf{H}}^2\big(\widehat{\pi}(\cdot \mid x), \pi_{N_\mu}(\cdot \mid x)\big)\big] \leq \varepsilon_{\mathsf{stat}}^2 := \frac{2\log(|\Pi|/\rho)}{n}.$$

Condition on the event that this guarantee holds. We invoke the following lemma, proven in the sequel.

**Lemma I.1.** *Let $P$ be a distribution on a discrete space $\mathcal{Y}$. Let $\boldsymbol{y}^\star = \arg\max_{y \in \mathcal{Y}} P(y)$ and let $P^\star := \max_{y \in \mathcal{Y}} P(y)$. Let $y_1, y_2, \ldots \sim P$, and for any stopping time $\tau$, define*

$$\widehat{y}_\tau \in \arg\max \{P(y) : y \in \{y_1, \ldots, y_\tau\}\}.$$

*Next, for a parameter $\mu > 0$, define the stopping time*

$$N_\mu := \inf\left\{k : \frac{1}{\max_{1 \leq i \leq k} P(y_i)} \leq k/\mu\right\}.$$

*Then*

$$\mathbb{E}[N_\mu] \leq \frac{\mu + (1/|\boldsymbol{y}^\star|)}{P^\star}.$$

*In addition, for any stopping time $\tau \geq N_\mu$ (including $\tau = N_\mu$ itself), we have $\mathbb{P}[\widehat{y}_\tau \notin \boldsymbol{y}^\star] \leq e^{-|\boldsymbol{y}^\star|\mu}$.*

This lemma, with our choice of $\mu$, ensures that *for all $x \in \mathcal{X}$,*

$$\pi_{N_\mu}(\boldsymbol{y}^\star(x) \mid x) \geq 1 - e^{-\mu} = 1 - \delta/2.$$

Following the reasoning in Eq. (19), this implies that

$$D_{\mathsf{H}}^2\big(\widehat{\pi}(\cdot \mid x), \pi_{N_\mu}(\cdot \mid x)\big) \gtrsim \delta \cdot \mathbb{I}\{\widehat{\pi}(\boldsymbol{y}^\star(x) \mid x) \leq 1 - \delta\},$$

so that

$$\mathbb{P}_{x \sim \mu}[\widehat{\pi}(\boldsymbol{y}^\star(x) \mid x) \leq 1 - \delta] \lesssim \frac{\varepsilon_{\mathsf{stat}}^2}{\delta}$$

as desired.

To bound the expected sample complexity, we observe that

$$\mathbb{E}[m] = n \cdot \mathbb{E}[N_\mu(x)] \overset{(i)}{\leq} \mathbb{E}\left[\frac{1 + \mu}{\pi_{\mathsf{base}}(\boldsymbol{y}^\star(x) \mid x)}\right] = (1 + \mu)\overline{C}_{\mathsf{cov}},$$

where inequality $(i)$ invokes Lemma I.1 once more. $\qquad\square$

**Proof of Lemma I.1.** Define $N^\star := \mu/P^\star$. To bound the tails of $N_\mu$, define

$$\tau = \inf\{k \mid k \geq N^\star \text{ and } \boldsymbol{y}^\star \cap \{y_1, \ldots, y_k\} \neq \varnothing\}.$$

It follows from the definition that $N_\mu \leq \tau$, since for any $k \geq N^\star$, if there exists $i \leq k$ such that $y_i \in \boldsymbol{y}^\star$, then

$$\frac{1}{P(y_i)} = \frac{1}{P^\star} = \frac{N^\star}{\mu} \leq \frac{k}{\mu}.$$

Thus, for $k \geq N^\star$, we can bound

$$\mathbb{P}[N_\mu > k] \leq \mathbb{P}[\tau > k] = \mathbb{P}[\mathcal{Y}^\star \cap \{y_1, \ldots, y_k\} = \varnothing] \leq (1 - |\boldsymbol{y}^\star|P^\star)^k,$$

and consequently

$$\mathbb{E}[N_\mu] \leq \mathbb{E}[\tau] \leq \mathbb{E}[\tau\mathbb{I}\{\tau \leq N^\star\}] + \mathbb{E}[\tau\mathbb{I}\{\tau > N^\star\}]$$

$$\leq N^\star + \sum_{k > N^\star} (1 - |\boldsymbol{y}^\star|P^\star)^k$$

$$\leq N^\star + \frac{1}{|\boldsymbol{y}^\star|P(y^\star)} = \frac{\mu + 1/|\boldsymbol{y}^\star|}{P(y^\star)}.$$

To prove correctness, observe that $N_\mu \geq N^\star$, because for all $y \in \mathcal{Y}$, $\frac{1}{P(y)} \geq N^\star/\mu$. Hence, any stopping time $\tau \geq N_\mu$ also satisfies $\tau \geq N^\star$, and moreover has $\widehat{y}_\tau \in \boldsymbol{y}^\star$ whenever $\boldsymbol{y}^\star \cap \{y_1, y_2, \ldots, y_\tau\} \neq \varnothing$. This fails to occur with probability no more than

$$\left(1 - \frac{|\boldsymbol{y}^\star|}{P^\star}\right)^{N^\star} = \left(1 - \frac{|\boldsymbol{y}^\star|}{P^\star}\right)^{\mu/P^\star} \leq e^{-|\boldsymbol{y}^\star|\mu}.$$

$\square$

## J  PROOFS FROM SECTION 4.2

### J.1  PROOF OF THEOREM 4.2

We state and prove a generalized version of Theorem 4.2. In the assumptions below, we fix a parameter $\gamma \in [0, 1)$; the setting $\gamma = 0$ corresponds to Theorem 4.2.

**Assumption J.1** (Coverage). *All $\pi \in \Pi$ satisfy $\mathcal{C}_\pi \leq C_{\mathsf{conc}}$ for a parameter $C_{\mathsf{conc}} \geq (1-\gamma)^{-1}C_{\mathsf{cov},\gamma}$, and $\mathcal{C}_{\pi_{\mathsf{base}}/\pi;\beta} \leq C_{\mathsf{loss}}$ for a parameter $C_{\mathsf{loss}} \geq |\mathcal{Y}|$.*

By Lemma 4.1′, Assumption J.1 is consistent with the assumption that $\pi_\beta^\star \in \Pi$.

**Assumption J.2** (Margin). *For all $x \in \mathrm{supp}(\mu)$, the initial model $\pi_{\mathsf{base}}$ satisfies*

$$\pi_{\mathsf{base}}(\boldsymbol{y}_\gamma^\star(x) \mid x) \geq (1 + \gamma_{\mathsf{margin}}) \cdot \pi_{\mathsf{base}}(y \mid x) \quad \forall y \notin \boldsymbol{y}_\gamma^\star(x)$$

*for a parameter $\gamma_{\mathsf{margin}} > 0$.*

**Theorem 4.2′.** *Assume that $\pi_\beta^\star \in \Pi$ (Assumption 4.3), and that Assumption 4.4 and Assumption 4.2 hold with respect to some $\gamma \in [0, 1)$, with parameters $C_{\mathsf{conc}}$, $C_{\mathsf{loss}}$, and $\gamma_{\mathsf{margin}} > 0$. For any $\delta, \rho \in (0, 1)$, the DPO algorithm in Eq. (4) ensures that with probability at least $1 - \rho$,*

$$\mathbb{P}_{x \sim \mu}\left[\widehat{\pi}(\boldsymbol{y}_\gamma^\star(x) \mid x) \leq 1 - \delta\right] \lesssim \frac{1}{\gamma_{\mathsf{margin}}\delta} \cdot \widetilde{O}\left(\sqrt{\frac{C_{\mathsf{conc}} \log^3(C_{\mathsf{loss}}|\Pi|\rho^{-1})}{n}} + \beta \log(C_{\mathsf{conc}}) + \gamma\right)$$

*where $\widetilde{O}(\cdot)$ hides factors logarithmic in $n$ and $C_{\mathsf{conc}}$ and doubly logarithmic in $\Pi$, $C_{\mathsf{loss}}$, and $\rho^{-1}$.*

We first state and prove some supporting technical lemmas, then proceed to the proof of Theorem 4.2′.

#### J.1.1  TECHNICAL LEMMAS

The following result is a generalization of Lemma 4.1.

**Lemma 4.1′.** *For all $\gamma \in (0, 1)$, the model $\pi_\beta^\star$ satisfies $\mathcal{C}_{\pi_\beta^\star} \leq (1-\gamma)^{-1}C_{\mathsf{cov},\gamma}$ and $\mathcal{C}_{\pi_{\mathsf{base}}/\pi_\beta^\star;\beta} \leq |\mathcal{Y}|$.*

**Proof of Lemma 4.1′.** For any fixed $x \in \mathcal{X}$, we have

$$\mathbb{E}_{y \sim \pi_\beta^\star(\cdot|x)} \left[ \frac{\pi_\beta^\star(y \mid x)}{\pi_{\mathsf{base}}(y \mid x)} \right] = \mathbb{E}_{y \sim \pi_\beta^\star(\cdot|x)} \left[ \frac{\pi_{\mathsf{base}}^{1+\beta^{-1}}(y \mid x)}{\pi_{\mathsf{base}}(y \mid x)} \right] \cdot \left( \sum_{y' \in \mathcal{Y}} \pi_{\mathsf{base}}^{1+\beta^{-1}}(y' \mid x) \right)^{-1}$$

$$\leq \max_{y \in \mathcal{Y}} \pi_{\mathsf{base}}^{\beta^{-1}}(y \mid x) \cdot \left( \sum_{y' \in \mathcal{Y}} \pi_{\mathsf{base}}^{1+\beta^{-1}}(y' \mid x) \right)^{-1}$$

$$\leq (1-\gamma)^{-1} \pi_{\mathsf{base}}^{\beta^{-1}}(\boldsymbol{y}_\gamma^\star(x) \mid x) \cdot \left( \sum_{y' \in \mathcal{Y}} \pi_{\mathsf{base}}^{1+\beta^{-1}}(y' \mid x) \right)^{-1}$$

$$= (1-\gamma)^{-1} \frac{\pi_{\mathsf{base}}^{1+\beta^{-1}}(\boldsymbol{y}_\gamma^\star(x) \mid x)}{\pi_{\mathsf{base}}(\boldsymbol{y}_\gamma^\star(x) \mid x)} \cdot \left( \sum_{y' \in \mathcal{Y}} \pi_{\mathsf{base}}^{1+\beta^{-1}}(y' \mid x) \right)^{-1}$$

$$= (1-\gamma)^{-1} \frac{\sum_{y \in \boldsymbol{y}_\gamma^\star(x)} \pi_{\mathsf{base}}^{1+\beta^{-1}}(y \mid x)}{\pi_{\mathsf{base}}(\boldsymbol{y}_\gamma^\star(x) \mid x)} \cdot \left( \sum_{y' \in \mathcal{Y}} \pi_{\mathsf{base}}^{1+\beta^{-1}}(y' \mid x) \right)^{-1}$$

$$\leq (1-\gamma)^{-1} \frac{1}{\pi_{\mathsf{base}}(\boldsymbol{y}_\gamma^\star(x) \mid x)}.$$

It follows that $\mathcal{C}_{\pi_\beta^\star} \leq (1-\gamma)^{-1} C_{\mathsf{cov},\gamma}$ as claimed.

For the second result, we have

$$\mathcal{C}_{\pi_{\mathsf{base}}/\pi_\beta^\star;\beta} = \mathbb{E}_{\pi_{\mathsf{base}}} \left[ \frac{1}{\pi_{\mathsf{base}}(y \mid x)} \cdot \left( \sum_{y' \in \mathcal{Y}} \pi_{\mathsf{base}}^{1+\beta^{-1}}(y' \mid x) \right)^\beta \right] \leq \mathbb{E}_{\pi_{\mathsf{base}}} \left[ \frac{1}{\pi_{\mathsf{base}}(y \mid x)} \right] = |\mathcal{Y}|.$$

$\square$

The next lemmas provide bounds on the tails of the self-rewards used in the algorithm.

**Lemma J.1.** *Suppose $\beta \in [0,1]$. For any model $\pi$, with probability at least $1-\delta$ over the draw of $x \sim \mu$, $y, y' \sim \pi_{\mathsf{base}}(\cdot \mid x)$, we have that for all $s > 0$,*

$$\mathbb{P}\left[ \left| \beta \log\left( \frac{\pi(y \mid x)}{\pi_{\mathsf{base}}(y \mid x)} \right) - \beta \log\left( \frac{\pi(y' \mid x)}{\pi_{\mathsf{base}}(y' \mid x)} \right) \right| > \log(2\mathcal{C}_{\pi_{\mathsf{base}}/\pi;\beta}) + s \right] \leq \exp(-s).$$

**Proof of Lemma J.1.** Define

$$X := \left| \beta \log\left( \frac{\pi(y \mid x)}{\pi_{\mathsf{base}}(y \mid x)} \right) - \beta \log\left( \frac{\pi(y' \mid x)}{\pi_{\mathsf{base}}(y' \mid x)} \right) \right|.$$

By the Chernoff method, we have that with probability at least $1-\delta$,
$X \leq \log(\mathbb{E}[\exp(X)]) + \log(\delta^{-1})$

$$= \log\left( \mathbb{E}_{x \sim \mu, y, y' \sim \pi_{\mathsf{base}}(x)} \left[ \exp\left( \left| \beta \log\left( \frac{\pi(y \mid x)}{\pi_{\mathsf{base}}(y \mid x)} \right) - \beta \log\left( \frac{\pi(y' \mid x)}{\pi_{\mathsf{base}}(y' \mid x)} \right) \right| \right) \right] \right) + \log(\delta^{-1})$$

$$\leq \log\left( \mathbb{E}_{x \sim \mu, y, y' \sim \pi_{\mathsf{base}}(x)} \left[ \exp\left( \beta \log\left( \frac{\pi(y \mid x)}{\pi_{\mathsf{base}}(y \mid x)} \right) - \beta \log\left( \frac{\pi(y' \mid x)}{\pi_{\mathsf{base}}(y' \mid x)} \right) \right) \right] \right.$$

$$\left. + \mathbb{E}_{x \sim \mu, y, y' \sim \pi_{\mathsf{base}}(x)} \left[ \exp\left( \beta \log\left( \frac{\pi(y' \mid x)}{\pi_{\mathsf{base}}(y' \mid x)} \right) - \beta \log\left( \frac{\pi(y \mid x)}{\pi_{\mathsf{base}}(y \mid x)} \right) \right) \right] \right) + \log(\delta^{-1})$$

$$= \log\left( 2 \mathbb{E}_{x \sim \mu, y, y' \sim \pi_{\mathsf{base}}(x)} \left[ \exp\left( \beta \log\left( \frac{\pi(y \mid x)}{\pi_{\mathsf{base}}(y \mid x)} \right) - \beta \log\left( \frac{\pi(y' \mid x)}{\pi_{\mathsf{base}}(y' \mid x)} \right) \right) \right] \right) + \log(\delta^{-1})$$

$$= \log\left( \mathbb{E}_{x \sim \mu, y, y' \sim \pi_{\mathsf{base}}(x)} \left[ \left( \frac{\pi(y \mid x)}{\pi_{\mathsf{base}}(y \mid x)} \cdot \frac{\pi_{\mathsf{base}}(y' \mid x)}{\pi(y' \mid x)} \right)^\beta \right] \right) + \log(2\delta^{-1}).$$

As long as $\beta \leq 1$, by Jensen's inequality, we can bound

$$\mathbb{E}_{x \sim \mu, y, y' \sim \pi_{\mathsf{base}}(x)}\left[\left(\frac{\pi(y \mid x)}{\pi_{\mathsf{base}}(y \mid x)} \cdot \frac{\pi_{\mathsf{base}}(y' \mid x)}{\pi(y' \mid x)}\right)^{\beta}\right]$$

$$\leq \mathbb{E}_{x \sim \mu, y' \sim \pi_{\mathsf{base}}(x)}\left[\left(\mathbb{E}_{y \sim \pi_{\mathsf{base}}(x)}\left[\frac{\pi(y \mid x)}{\pi_{\mathsf{base}}(y \mid x)}\right] \cdot \frac{\pi_{\mathsf{base}}(y' \mid x)}{\pi(y' \mid x)}\right)^{\beta}\right]$$

$$= \mathbb{E}_{x \sim \mu, y' \sim \pi_{\mathsf{base}}(x)}\left[\left(\frac{\pi_{\mathsf{base}}(y' \mid x)}{\pi(y' \mid x)}\right)^{\beta}\right]$$

$$= \mathcal{C}_{\pi_{\mathsf{base}}/\pi;\beta},$$

which proves the result. $\qquad\qquad\square$

**Lemma J.2.** *Let $\beta \in [0, 1]$. For all models $\pi$, we have*

$$\mathbb{E}_{x \sim \mu, y, y' \sim \pi_{\mathsf{base}}(\cdot \mid x)}\left[\left|\beta \log\left(\frac{\pi(y \mid x)}{\pi_{\mathsf{base}}(y \mid x)}\right) - \beta \log\left(\frac{\pi(y' \mid x)}{\pi_{\mathsf{base}}(y' \mid x)}\right)\right|^{4}\right] \leq O(\log^4(\mathcal{C}_{\pi_{\mathsf{base}}/\pi;\beta}) + 1).$$

**Proof of Lemma J.2.** Define

$$X := \left|\beta \log\left(\frac{\pi(y \mid x)}{\pi_{\mathsf{base}}(y \mid x)}\right) - \beta \log\left(\frac{\pi(y' \mid x)}{\pi_{\mathsf{base}}(y' \mid x)}\right)\right|.$$

Set $k = \log(2\mathcal{C}_{\pi_{\mathsf{base}}/\pi;\beta})$. We can bound

$$\mathbb{E}[X^4] = \mathbb{E}\left[\int_0^{\infty} \mathbb{I}\{X^4 > t\}dt\right]$$

$$= 4\,\mathbb{E}\left[\int_0^{\infty} \mathbb{I}\{X > t\}t^3 dt\right]$$

$$= 4\int_0^{\infty} \mathbb{P}[X > t]t^3 dt$$

$$\leq k^4 + 4\int_k^{\infty} \mathbb{P}[X > t]t^3 dt$$

$$\leq k^4 + 4\int_k^{\infty} e^{k-t}t^3 dt$$

$$= k^4 + 4(k^3 + 3k^2 + 6k + 6)$$

$$= O(k^4 + 1),$$

where the third-to-last line uses Lemma J.1. $\qquad\qquad\square$

### J.1.2 PROOF OF THEOREM 4.2$'$

**Proof of Theorem 4.2$'$.** For any model $\pi \in \Pi$, define $J(\pi) := \mathbb{E}_{\pi}[\log \pi_{\mathsf{base}}(y \mid x)]$. Let $\widehat{\pi} \in \Pi$ denote the model returned by the DPO algorithm in Eq. (8). Let $\mathbb{E}_{\pi, \pi'}[\cdot]$ denote shorthand for $\mathbb{E}_{x \sim \mu, y \sim \pi(x), y' \sim \pi'(x)}[\cdot]$, and for any $r : \mathcal{X} \times \mathcal{Y} \to \mathbb{R}$ define $\Delta^r(x, y, y') := r(x, y) - r(x, y')$. Define

$$r^{\star}(x, y) := \log \pi_{\mathsf{base}}(y \mid x) = \beta \log\left(\frac{\pi_{\beta}^{\star}(y \mid x)}{\pi_{\mathsf{base}}(y \mid x)}\right) + Z(x),$$

and let $\widehat{r}(x, y) := \beta \log\left(\frac{\widehat{\pi}(y \mid x)}{\pi_{\mathsf{base}}(y \mid x)}\right)$. By a standard argument (Huang et al., 2024), we have

$$\widehat{\pi} \in \underset{\pi:\mathcal{X}\to\Delta(\mathcal{Y})}{\arg\max} \ \mathbb{E}_{\pi}[\widehat{r}(x, y)] - \beta D_{\mathsf{KL}}(\pi \,\|\, \pi_{\mathsf{base}}). \qquad\qquad (20)$$

Therefore for any comparator model $\pi^\star : \mathcal{X} \to \Delta(\mathcal{Y})$ (not necessarily in the model class $\Pi$), we have

$$
\begin{aligned}
J(\pi^\star) - J(\widehat{\pi}) &= \mathbb{E}_{\pi^\star}[r^\star(x,y)] - \mathbb{E}_{\widehat{\pi}}[r^\star(x,y)] \\
&= \mathbb{E}_{\pi^\star}[\widehat{r}(x,y)] - \beta D_{\mathsf{KL}}(\pi^\star \| \pi_{\mathsf{base}}) - \mathbb{E}_{\widehat{\pi}}[\widehat{r}(x,y)] + \beta D_{\mathsf{KL}}(\widehat{\pi} \| \pi_{\mathsf{base}}) \\
&\quad + \mathbb{E}_{\pi^\star}[r^\star(x,y) - \widehat{r}(x,y)] + \beta D_{\mathsf{KL}}(\pi^\star \| \pi_{\mathsf{base}}) + \mathbb{E}_{\widehat{\pi}}[\widehat{r}(x,y) - r^\star(x,y)] - \beta D_{\mathsf{KL}}(\widehat{\pi} \| \pi_{\mathsf{base}}) \\
&\leq \mathbb{E}_{\pi^\star}[r^\star(x,y) - \widehat{r}(x,y)] + \beta D_{\mathsf{KL}}(\pi^\star \| \pi_{\mathsf{base}}) + \mathbb{E}_{\widehat{\pi}}[\widehat{r}(x,y) - r^\star(x,y)] - \beta D_{\mathsf{KL}}(\widehat{\pi} \| \pi_{\mathsf{base}}) \\
&= \mathbb{E}_{\pi^\star, \pi_{\mathsf{base}}}\Big[\Delta^{r^\star}(x,y,y') - \Delta^{\widehat{r}}(x,y,y')\Big] + \mathbb{E}_{\widehat{\pi}, \pi_{\mathsf{base}}}\Big[\Delta^{\widehat{r}}(x,y,y') - \Delta^{r^\star}(x,y,y')\Big] \\
&\quad + \beta D_{\mathsf{KL}}(\pi^\star \| \pi_{\mathsf{base}}) - \beta D_{\mathsf{KL}}(\widehat{\pi} \| \pi_{\mathsf{base}})
\end{aligned}
\tag{21}
$$

where the inequality uses Eq. (20). To bound the right-hand side above, we will use the following lemma, which is proven in the sequel.

**Lemma J.3.** *For any model $\pi$ and any $\eta > 0$, we have that*

$$
\mathbb{E}_{\pi, \pi_{\mathsf{base}}}\Big[\Big|\Delta^{r^\star}(x,y,y') - \Delta^{\widehat{r}}(x,y,y')\Big|\Big]
$$

$$
\lesssim \mathcal{C}_\pi^{1/2} \cdot \left(\mathbb{E}_{\pi_{\mathsf{base}}, \pi_{\mathsf{base}}}\Big[\Big|\Delta^{r^\star}(x,y,y') - \Delta^{\widehat{r}}(x,y,y')\Big|^2 \mathbb{I}\Big\{\big|\Delta^{r^\star}\big| \leq \eta, \big|\Delta^{\widehat{r}}\big| \leq \eta\Big\}\Big]\right)^{1/2}
$$

$$
+ \mathcal{C}_\pi^{1/2}(\log(\mathcal{C}_{\pi_{\mathsf{base}}/\widehat{\pi};\beta}) + \log(\mathcal{C}_{\pi_{\mathsf{base}}/\pi^\star_\beta;\beta})) \cdot \left(\mathbb{P}_{\pi_{\mathsf{base}}, \pi_{\mathsf{base}}}\Big[\big|\Delta^{r^\star}\big| > \eta\Big] + \mathbb{P}_{\pi_{\mathsf{base}}, \pi_{\mathsf{base}}}\Big[\big|\Delta^{\widehat{r}}\big| > \eta\Big]\right)^{1/4}.
$$

Using Lemma J.3 to bound the first two terms of Eq. (21), and using the fact that all $\pi \in \Pi$ have $\mathcal{C}_\pi \leq C_{\mathsf{conc}}$ and $\mathcal{C}_{\pi_{\mathsf{base}}/\pi;\beta} \leq C_{\mathsf{loss}}$, we have that

$$
J(\pi^\star) - J(\widehat{\pi})
$$

$$
\lesssim (\mathcal{C}_{\pi^\star} + C_{\mathsf{conc}})^{1/2} \cdot \left(\mathbb{E}_{\pi_{\mathsf{base}}, \pi_{\mathsf{base}}}\Big[\Big|\Delta^{r^\star}(x,y,y') - \Delta^{\widehat{r}}(x,y,y')\Big|^2 \mathbb{I}\Big\{\big|\Delta^{r^\star}\big| \leq \eta, \big|\Delta^{\widehat{r}}\big| \leq \eta\Big\}\Big]\right)^{1/2}
$$

$$
+ (\mathcal{C}_{\pi^\star} + C_{\mathsf{conc}})^{1/2} \log(C_{\mathsf{loss}}) \cdot \left(\mathbb{P}_{\pi_{\mathsf{base}}, \pi_{\mathsf{base}}}\Big[\big|\Delta^{r^\star}\big| > \eta\Big] + \mathbb{P}_{\pi_{\mathsf{base}}, \pi_{\mathsf{base}}}\Big[\big|\Delta^{\widehat{r}}\big| > \eta\Big]\right)^{1/4} + \beta D_{\mathsf{KL}}(\pi^\star \| \pi_{\mathsf{base}}).
\tag{22}
$$

Let us overload notation and write $\Delta^\pi(x,y,y') = \beta \log\left(\frac{\pi(y|x)}{\pi_{\mathsf{base}}(y|x)}\right) - \beta \log\left(\frac{\pi(y'|x)}{\pi_{\mathsf{base}}(y'|x)}\right)$, so that $\Delta^{\widehat{\pi}} = \Delta^{\widehat{r}}$ and $\Delta^{\pi^\star_\beta} = \Delta^{r^\star}$. Since $\pi^\star_\beta \in \Pi$, the definition of $\widehat{\pi}$ in Eq. (4) implies that

$$
\begin{aligned}
\sum_{(x,y,y') \in \mathcal{D}_{\mathsf{pref}}} \left(\Delta^{\widehat{\pi}}(x,y,y') - \Delta^{\pi^\star_\beta}(x,y,y')\right)^2 &\leq \min_{\pi \in \Pi} \sum_{(x,y,y') \in \mathcal{D}_{\mathsf{pref}}} \left(\Delta^\pi(x,y,y') - \Delta^{\pi^\star_\beta}(x,y,y')\right)^2 \\
&\leq \sum_{(x,y,y') \in \mathcal{D}_{\mathsf{pref}}} \left(\Delta^{\pi^\star_\beta}(x,y,y') - \Delta^{\pi^\star_\beta}(x,y,y')\right)^2 \\
&= 0.
\end{aligned}
$$

Define $B_{n,\rho} := \log(2nC_{\mathsf{loss}}|\Pi|\rho^{-1})$. It is immediate that

$$
\sum_{(x,y,y') \in \mathcal{D}_{\mathsf{pref}}} \left(\Delta^{\widehat{\pi}}(x,y,y') - \Delta^{\pi^\star_\beta}(x,y,y')\right)^2 \mathbb{I}\Big\{\big|\Delta^{\widehat{\pi}}\big| \leq B_{n,\rho}, \big|\Delta^{\pi^\star_\beta}\big| \leq B_{n,\rho}\Big\} \leq 0.
$$

From here, Bernstein's inequality and a union bound implies that with probability at least $1 - \rho$,

$$
\mathbb{E}_{\pi_{\mathsf{base}}, \pi_{\mathsf{base}}}\Big[\Big|\Delta^{\widehat{\pi}}(x,y,y') - \Delta^{\pi^\star_\beta}(x,y,y')\Big|^2 \mathbb{I}\Big\{\big|\Delta^{\widehat{\pi}}\big| \leq B_{n,\rho}, \big|\Delta^{\pi^\star_\beta}\big| \leq B_{n,\rho}\Big\}\Big]
$$

$$
\lesssim \frac{B_{n,\rho}^2 \log(|\Pi|\rho^{-1})}{n} =: \varepsilon_{\mathsf{stat}}^2.
$$

In particular, if we combine this with Eq. (22) and set $\eta = B_{n,\rho}$, then Lemma J.1 implies that

$$
J(\pi^\star) - J(\widehat{\pi}) \lesssim (\mathcal{C}_{\pi^\star} + C_{\mathsf{conc}})^{1/2} \cdot \varepsilon_{\mathsf{stat}} + (\mathcal{C}_{\pi^\star} + C_{\mathsf{conc}})^{1/2} \log(C_{\mathsf{loss}}) \cdot \rho^{1/4} + \beta D_{\mathsf{KL}}(\pi^\star \| \pi_{\mathsf{base}}).
$$

Note that the above bound holds for any $\pi^\star : \mathcal{X} \to \Delta(\mathcal{Y})$. We define $\pi^\star$ by

$$\pi^\star(y \mid x) := \frac{\pi_{\mathsf{base}}(y \mid x)\mathbb{I}[y \in \boldsymbol{y}_\gamma^\star(x)]}{\pi_{\mathsf{base}}(\boldsymbol{y}_\gamma^\star(x) \mid x)},$$

which can be seen to satisfy $\mathcal{C}_{\pi^\star} \leq C_{\mathsf{cov},\gamma} \leq C_{\mathsf{conc}}$ and $D_{\mathsf{KL}}(\pi^\star \parallel \pi_{\mathsf{base}}) \leq \log(\mathcal{C}_{\pi^\star}) \leq \log(C_{\mathsf{conc}})$. With this choice, we can further bound the expression above by

$$J(\pi^\star) - J(\widehat{\pi}) \lesssim (C_{\mathsf{conc}})^{1/2} \cdot \varepsilon_{\mathsf{stat}} + (C_{\mathsf{conc}})^{1/2} \log(C_{\mathsf{loss}}) \cdot \rho^{1/4} + \beta \log(C_{\mathsf{conc}})$$

Given a desired failure probability $\rho$, applying the bound above with $\rho' := \rho \wedge (\varepsilon_{\mathsf{stat}}/\log(C_{\mathsf{loss}}))^4$ then gives

$$J(\pi^\star) - J(\widehat{\pi}) \lesssim (C_{\mathsf{conc}})^{1/2} \cdot \varepsilon_{\mathsf{stat}} + \beta \log(C_{\mathsf{conc}}).$$

Finally, we observe that for our choice of $\pi^\star$, under the margin condition with parameter $\gamma$, we have

$$\begin{aligned}
J(\pi^\star) - J(\widehat{\pi}) &= \mathbb{E}_{x\sim\mu} \mathbb{E}_{y,y'\sim\pi^\star,\widehat{\pi}} \left[\log\left(\frac{\pi_{\mathsf{base}}(y \mid x)}{\pi_{\mathsf{base}}(y' \mid x)}\right)\right] \\
&\gtrsim \gamma_{\mathsf{margin}} \cdot \mathbb{E}_{x\sim\mu} \mathbb{E}_{y'\sim\widehat{\pi}} \left[\mathbb{I}\{y' \notin \boldsymbol{y}_\gamma^\star(x)\}\right] - \gamma \\
&\gtrsim \gamma_{\mathsf{margin}}\delta \cdot \mathbb{E}_{x\sim\mu} \left[\mathbb{I}\{\widehat{\pi}(\boldsymbol{y}_\gamma^\star(x) \mid x) \leq 1 - \delta\}\right] - \gamma
\end{aligned}$$

where the first inequality uses Assumption J.2 together with the fact that $y \in \boldsymbol{y}_\gamma^\star(x)$ with probability 1 over $x \sim \mu$ and $y \sim \pi^\star(\cdot \mid x)$. This proves the result.

$\square$

**Proof of Lemma J.3.** For any $\eta > 0$, we can bound

$$\begin{aligned}
\mathbb{E}_{\pi,\pi_{\mathsf{base}}} \left[\left|\Delta^{r^\star}(x,y,y') - \Delta^{\widehat{r}}(x,y,y')\right|\right] &\leq \mathbb{E}_{\pi,\pi_{\mathsf{base}}} \left[\left|\Delta^{r^\star}(x,y,y') - \Delta^{\widehat{r}}(x,y,y')\right|\mathbb{I}\left\{|\Delta^{r^\star}| \leq \eta, |\Delta^{\widehat{r}}| \leq \eta\right\}\right] \\
&\quad + \mathbb{E}_{\pi,\pi_{\mathsf{base}}} \left[\left|\Delta^{r^\star}(x,y,y') - \Delta^{\widehat{r}}(x,y,y')\right|\mathbb{I}\left\{|\Delta^{r^\star}| > \eta \vee |\Delta^{\widehat{r}}| > \eta\right\}\right].
\end{aligned}$$

For the second term above, we can use Cauchy-Schwarz to bound

$$\begin{aligned}
&\mathbb{E}_{\pi,\pi_{\mathsf{base}}} \left[\left|\Delta^{r^\star}(x,y,y') - \Delta^{\widehat{r}}(x,y,y')\right|\mathbb{I}\left\{|\Delta^{r^\star}| > \eta \vee |\Delta^{\widehat{r}}| > \eta\right\}\right] \\
&\leq \mathcal{C}_\pi^{1/2} \cdot \left(\mathbb{E}_{\pi_{\mathsf{base}},\pi_{\mathsf{base}}} \left[\left|\Delta^{r^\star}(x,y,y') - \Delta^{\widehat{r}}(x,y,y')\right|^2 \mathbb{I}\left\{|\Delta^{r^\star}| > \eta \vee |\Delta^{\widehat{r}}| > \eta\right\}\right]\right)^{1/2} \\
&\lesssim \mathcal{C}_\pi^{1/2} \cdot \left(\mathbb{P}_{\pi_{\mathsf{base}},\pi_{\mathsf{base}}} \left[|\Delta^{r^\star}| > \eta\right] + \mathbb{P}_{\pi_{\mathsf{base}},\pi_{\mathsf{base}}} \left[|\Delta^{\widehat{r}}| > \eta\right]\right)^{1/4} \\
&\quad \cdot \left(\mathbb{E}_{\pi_{\mathsf{base}},\pi_{\mathsf{base}}} \left[\left|\Delta^{r^\star}(x,y,y')\right|^4\right] + \mathbb{E}_{\pi_{\mathsf{base}},\pi_{\mathsf{base}}} \left[\left|\Delta^{\widehat{r}}(x,y,y')\right|^4\right]\right)^{1/4} \\
&\lesssim \mathcal{C}_\pi^{1/2} \cdot \left(\mathbb{P}_{\pi_{\mathsf{base}},\pi_{\mathsf{base}}} \left[|\Delta^{r^\star}| > \eta\right] + \mathbb{P}_{\pi_{\mathsf{base}},\pi_{\mathsf{base}}} \left[|\Delta^{\widehat{r}}| > \eta\right]\right)^{1/4} \cdot (\log(\mathcal{C}_{\pi_{\mathsf{base}}/\widehat{\pi};\beta}) + \log(\mathcal{C}_{\pi_{\mathsf{base}}/\pi_\beta^\star;\beta})),
\end{aligned}$$

where the last inequality follows from Lemma J.2.

Meanwhile, for the first term, for any $\lambda > 0$ we can bound

$$\begin{aligned}
&\mathbb{E}_{\pi,\pi_{\mathsf{base}}} \left[\left|\Delta^{r^\star}(x,y,y') - \Delta^{\widehat{r}}(x,y,y')\right|\mathbb{I}\left\{|\Delta^{r^\star}| \leq \eta, |\Delta^{\widehat{r}}| \leq \eta\right\}\right] \\
&\leq \mathcal{C}_\pi^{1/2} \left(\mathbb{E}_{\pi_{\mathsf{base}},\pi_{\mathsf{base}}} \left[\left|\Delta^{r^\star}(x,y,y') - \Delta^{\widehat{r}}(x,y,y')\right|^2 \mathbb{I}\left\{|\Delta^{r^\star}| \leq \eta, |\Delta^{\widehat{r}}| \leq \eta\right\}\right]\right)^{1/2}.
\end{aligned}$$

$\square$

## J.2    PROOF OF THEOREM 4.3 AND THEOREM J.3

In this section we prove Theorem 4.3 as well as Theorem J.3, the application to linear softmax models. For the formal theorem statements, see Theorem J.2 and Theorem J.3 respectively. The section is organized as follows.

- Appendix J.2.1 gives necessary background on KL-regularized policy optimization, as well as the Sequential Extrapolation Coefficient.

- Appendix J.2.2 presents a generic guarantee for XPO under a general choice of reward function.

- Appendix J.2.3 instantiates the result above with the self-reward function $r(x, y) := \log \pi_{\mathsf{base}}(y \mid x)$ to prove Theorem 4.3.

- Finally, Appendix J.2.4 applies the preceding results to prove Theorem J.3.

### J.2.1    BACKGROUND

To begin, we give background on KL-regularized policy optimization and the Sequential Extrapolation Coefficient.

**KL-regularized policy optimization.**    Let $\beta > 0$ be given, and let $r : \mathcal{X} \times \mathcal{Y} \to [-R_{\mathsf{max}}, R_{\mathsf{max}}]$ be an unknown reward function on prompt/action pairs. Define a value function $J_\beta$ over model class $\Pi$ by:

$$J_\beta(\pi) := \mathbb{E}_\pi[r(x, y)] - \beta \cdot D_{\mathsf{KL}}(\mathbb{P}^\pi \,\|\, \mathbb{P}^{\pi_{\mathsf{base}}}).$$

We refer to this as a *KL-regularized policy optimization* objective (we use the term "policy" following the reinforcement learning literature; for our setting, policies correspond to models). Given query access to $r$, the goal is to find $\widehat{\pi} \in \Pi$ such that

$$J_\beta(\pi_\beta^\star) - J_\beta(\widehat{\pi}) \leq \epsilon$$

where $\pi_\beta^\star(y \mid x) \propto \pi_{\mathsf{base}}(y \mid x) \exp(\beta^{-1} r(x, y))$ is the model that maximizes $J_\beta$ over all models $\pi : \mathcal{X} \to \Delta(\mathcal{Y})$.

We make use of the following assumptions, as in Xie et al. (2024).

**Assumption J.3** (Realizability). *It holds that $\pi_\beta^\star \in \Pi$.*

**Assumption J.4** (Bounded density ratios). *For all $\pi \in \Pi$, $(x, y) \in \mathcal{X} \times \mathcal{Y}$, $\left|\beta \log \frac{\pi(y|x)}{\pi_{\mathsf{base}}(y|x)}\right| \leq V_{\mathsf{max}}$.*

Finally, we require two definitions.

**Definition J.1** (Sequential Extrapolation Coefficient for RLHF, (Xie et al., 2024)). *For a model class $\Pi$, reward function $r$, reference model $\pi_{\mathsf{base}}$, and parameters $T \in \mathbb{N}$ and $\beta, \lambda > 0$, the Sequential Extrapolation Coefficient is defined as*

$\mathsf{SEC}(\Pi, r, T, \beta, \lambda; \pi_{\mathsf{base}})$

$$:= \sup_{\pi^{(1)}, \dots, \pi^{(T)} \in \Pi} \left\{ \sum_{t=1}^T \frac{\mathbb{E}^{(t)}\left[\beta \log \frac{\pi^{(t)}(y|x)}{\pi_{\mathsf{base}}(y|x)} - r(x, y) - \beta \log \frac{\pi^{(t)}(y'|x)}{\pi_{\mathsf{base}}(y'|x)} + r(x, y')\right]^2}{\lambda \vee \sum_{i=1}^{t-1} \mathbb{E}^{(i)}\left[\left(\beta \log \frac{\pi^{(t)}(y|x)}{\pi_{\mathsf{base}}(y|x)} - r(x, y) - \beta \log \frac{\pi^{(t)}(y'|x)}{\pi_{\mathsf{base}}(y'|x)} + r(x, y')\right)^2\right]} \right\}$$

*where $\mathbb{E}^{(t)}$ denotes expectation over $x \sim \mu$, $y \sim \pi^{(t)}(\cdot \mid x)$, and $y' \sim \pi_{\mathsf{base}}(\cdot \mid x)$.*

**Definition J.2.** *Let $\epsilon > 0$. We say that $\Psi \subseteq \Pi$ is a $\epsilon$-net for model class $\Pi$ if for every $\pi \in \Pi$ there exists $\pi' \in \Psi$ such that*

$$\max_{x \in \mathcal{X}} \max_{y \in \mathcal{Y}} \left| \log \frac{\pi(y \mid x)}{\pi'(y \mid x)} \right| \leq \epsilon.$$

*We write $\mathcal{N}(\Pi, \epsilon)$ to denote the size of the smallest $\epsilon$-net for $\Pi$.*

---

**Algorithm 1** Reward-based variant of Exploratory Preference Optimization (Xie et al., 2024)

---

**input:** Base model $\pi_{\mathsf{base}} : \mathcal{X} \to \Delta(\mathcal{Y})$, reward function $r : \mathcal{X} \times \mathcal{Y} \to \mathbb{R}$, number of iterations $T \in \mathbb{N}$, KL regularization coefficient $\beta > 0$, optimism coefficient $\alpha > 0$.

Initialize: $\pi^{(1)} \leftarrow \pi_{\mathsf{base}}$, $\mathcal{D}^{(0)} \leftarrow \varnothing$.

**for** iteration $t = 1, \ldots, T$ **do**

    **Generate sample:** $(x^{(t)}, y^{(t)}, \widetilde{y}^{(t)})$ via $x^{(t)} \sim \mu$, $y^{(t)} \sim \pi^{(t)}(\cdot \mid x^{(t)})$, $\widetilde{y}^{(t)} \sim \pi_{\mathsf{base}}(\cdot \mid x^{(t)})$.

    **Update dataset:** $\mathcal{D}^{(t)} \leftarrow \mathcal{D}^{(t-1)} \cup \{(x^{(t)}, y^{(t)}, \widetilde{y}^{(t)})\}$.

    **Model optimization with global optimism:**

$$
\pi^{(t+1)} \leftarrow \arg\min_{\pi \in \Pi} \left\{ \alpha \sum_{(x,y,y') \in \mathcal{D}^{(t)}} \log(\pi(y' \mid x)) \right.
$$

$$
\left. - \sum_{(x,y,y') \in \mathcal{D}^{(t)}} \left( \beta \log \frac{\pi(y \mid x)}{\pi_{\mathsf{base}}(y \mid x)} - \beta \log \frac{\pi(y' \mid x)}{\pi_{\mathsf{base}}(y' \mid x)} - (r(x,y) - r(x,y')) \right)^2 \right\}.
$$

**return:** $\widehat{\pi} \leftarrow \arg\max_{t \in [T+1]} J_\beta(\pi^{(t)})$.         ▷ Can estimate $J_\beta(\pi^{(t)})$ using validation data.

---

### J.2.2 GUARANTEES FOR KL-REGULARIZED POLICY OPTIMIZATION WITH XPO

In this section, we give self-contained guarantees for the XPO algorithm (Algorithm 1). XPO was introduced in Xie et al. (2024) for KL-regularized policy optimization in the related setting where the learner only has indirect access to the reward function $r$ through *preference data* (specifically, pairs of actions labeled via a Bradley-Terry model). Standard offline algorithms for this problem, such as DPO, require bounds on concentrability of the model class (see e.g. Eq. (9)). Xie et al. (2024) show that the XPO algorithm avoids this dependence, and instead requires bounded Sequential Extrapolation Coefficient.

Algorithm 1 is a variant of the XPO algorithm which is adapted to reward-based feedback (as opposed to preference-based feedback), and Theorem J.1 shows that this algorithm enjoys guarantees similar to those of Xie et al. (2024) for this setting. Note that this is not an immediate corollary of the results in Xie et al. (2024), since the sample complexity in the preference-based setting scales with $e^{O(R_{\mathsf{max}})}$, and for our application to sharpening it is important to avoid this dependence. However, our algorithm and analysis only diverge from Xie et al. (2024) in a few places.

**Theorem J.1** (Variant of Xie et al. (2024, Theorem 3.1)). *Suppose that Assumptions J.3 and J.4 hold. For any $T \in \mathbb{N}$, $\epsilon_{\mathsf{disc}}, \rho \in (0, 1)$, by setting $\alpha := \frac{\beta}{R_{\mathsf{max}} + V_{\mathsf{max}}} \sqrt{\frac{\log(2\mathcal{N}(\Pi, \epsilon_{\mathsf{disc}})T/\rho)}{\mathsf{SEC}(\Pi)T}}$, Algorithm 1 produces a model $\widehat{\pi} \in \Pi$ such that with probability at least $1 - \rho$,*

$$
\beta D_{\mathsf{KL}}(\widehat{\pi} \| \pi_\beta^\star) = J_\beta(\pi_\beta^\star) - J_\beta(\widehat{\pi}) \lesssim (R_{\mathsf{max}} + V_{\mathsf{max}}) \sqrt{\frac{\mathsf{SEC}(\Pi) \log(2\mathcal{N}(\Pi, \epsilon_{\mathsf{disc}})T/\rho)}{T}}
$$

$$
+ \beta \epsilon_{\mathsf{disc}} \sqrt{\mathsf{SEC}(\Pi)T}
$$

*where $\mathsf{SEC}(\Pi) := \mathsf{SEC}(\Pi, r, T, \beta, V_{\mathsf{max}}^2; \pi_{\mathsf{base}})$.*

**Proof of Theorem J.1.** For compactness, we abbreviate $\mathsf{SEC}(\Pi) := \mathsf{SEC}(\Pi, r, T, \beta, V_{\mathsf{max}}^2; \pi_{\mathsf{base}})$. From Equation (37) of Xie et al. (2024), we have

$$
\frac{1}{T} \sum_{t=1}^{T} J_\beta(\pi_\beta^\star) - J_\beta(\pi^{(t)})
$$

$$
\lesssim \frac{\alpha}{\beta}(R_{\mathsf{max}} + V_{\mathsf{max}})^2 \cdot \mathsf{SEC}(\Pi) + \frac{\beta}{\alpha T} + \frac{V_{\mathsf{max}}}{T} + \frac{1}{T} \sum_{t=2}^{T} \mathbb{E}_{(x,y) \sim \pi_{\mathsf{base}}} [\beta \log \pi^{(t)}(y \mid x) - \beta \log \pi_\beta^\star(y \mid x)]
$$

$$
+ \frac{\beta}{\alpha(R_{\mathsf{max}} + V_{\mathsf{max}})^2 T} \sum_{t=2}^{T} \mathbb{E}_{\substack{x \sim \mu \\ y,y' \sim \overline{\pi}^{(t)}|x}} \left[ \left( \beta \log \frac{\pi^{(t)}(y \mid x)}{\pi_{\mathsf{base}}(y \mid x)} - r(x,y) - \beta \log \frac{\pi^{(t)}(y' \mid x)}{\pi_{\mathsf{base}}(y' \mid x)} + r(x,y') \right)^2 \right]
$$

where $\overline{\pi}^{(t)} := \frac{1}{t-1} \sum_{i<t} \pi^{(i)} \otimes \pi_{\mathsf{base}}$ denotes the model that, given $x \in \mathcal{X}$, samples $i \sim \mathsf{Unif}([t-1])$ and then samples $y \sim \pi^{(i)}(\cdot \mid x)$ and $y' \sim \pi_{\mathsf{base}}(\cdot \mid x)$. For any $2 \le t \le T$, define $L^{(t)} : \Pi \to [0, \infty)$

by

$$L^{(t)}(\pi) := \mathbb{E}_{(x,y)\sim\pi_{\text{base}}} [\beta \log \pi(y \mid x) - \beta \log \pi_{\beta}^{\star}(y \mid x)]$$

$$+ \frac{\beta}{\alpha(V_{\text{max}} + R_{\text{max}})^2} \mathbb{E}_{\substack{x\sim\mu \\ y,y'\sim\overline{\pi}^{(t)}\mid x}} \left[ \left( \beta \log \frac{\pi(y \mid x)}{\pi_{\text{base}}(y \mid x)} - r(x,y) - \beta \log \frac{\pi(y' \mid x)}{\pi_{\text{base}}(y' \mid x)} + r(x,y') \right)^2 \right].$$

Similarly, define

$$\widehat{L}^{(t)}(\pi) := \sum_{(x,y,y')\in\mathcal{D}^{(t)}} [\beta \log \pi(y' \mid x) - \beta \log \pi_{\beta}^{\star}(y' \mid x)]$$

$$+ \frac{\beta}{\alpha(V_{\text{max}} + R_{\text{max}})^2} \sum_{(x,y,y')\in\mathcal{D}^{(t)}} \left[ \left( \beta \log \frac{\pi(y \mid x)}{\pi_{\text{base}}(y \mid x)} - r(x,y) - \beta \log \frac{\pi(y' \mid x)}{\pi_{\text{base}}(y' \mid x)} + r(x,y') \right)^2 \right]$$

where $\mathcal{D}^{(t)}$ is the dataset defined in iteration $t$ of Algorithm 1. By Assumption J.3 we have $\pi_{\beta}^{\star} \in \Pi$, so $\inf_{\pi\in\Pi} \widehat{L}^{(t)}(\pi) \le 0$. Moreover by definition, $\pi^{(t)} \in \arg\min_{\pi\in\Pi} \widehat{L}^{(t)}$.

Let $\Psi$ be an $\epsilon_{\text{disc}}$-net over $\Pi$, of size $\mathcal{N}(\Pi, \epsilon_{\text{disc}})$. Fix any $\pi \in \Psi$ and $2 \le t \le T$, and define increments $X_i := \widehat{L}^{(i)}(\pi) - \widehat{L}^{(i-1)}(\pi)$ for $2 \le i \le t$, with the notation $\widehat{L}^{(1)}(\pi) := 0$ so that $\widehat{L}^{(t)}(\pi) = \sum_{i=2}^t X_i$. Let $\mathcal{F}_i$ be the filtration induced by $\mathcal{D}^{(i)}$ and define $\gamma_i := \mathbb{E}[X_i \mid \mathcal{F}_{i-1}]$. Observe that $(t-1)L^{(t)}(\pi) = \sum_{i=2}^t \gamma_i$. For any $i$, note that we can write $X_i = Y_i + Z_i$ where $Y_i \in [-V_{\text{max}}, V_{\text{max}}]$ and $Z_i \in [0, \beta/\alpha]$. By Corollary F.1, it holds with probability at least $1 - \rho/(2|\Pi|T)$

$$\sum_{i=2}^t \mathbb{E}[Z_i \mid \mathcal{F}_{i-1}] \lesssim \frac{\beta}{\alpha} \log(2|\Psi|T/\rho) + \sum_{i=2}^t Z_i.$$

By Azuma-Hoeffding, it holds with probability at least $1 - \rho/(2|\Pi|T)$ that

$$\sum_{i=2}^t \mathbb{E}[Y_i \mid \mathcal{F}_{i-1}] \lesssim V_{\text{max}}\sqrt{T \log(2|\Psi|T/\rho)} + \sum_{i=2}^t Y_i.$$

Hence, with probability at least $1 - \rho/(|\Psi|T)$ we have

$$(t-1)L^{(t)}(\pi) \lesssim \frac{\beta}{\alpha} \log(2|\Psi|T/\rho) + V_{\text{max}}\sqrt{T \log(2|\Psi|T/\rho)} + \widehat{L}^{(t)}(\pi).$$

With probability at least $1 - \rho$ this bound holds for all $\pi \in \Psi$ and $2 \le t \le T$. Henceforth condition on this event. Fix any $\pi \in \Pi$ and $2 \le t \le T$. Since $\Psi$ is an $\epsilon$-net for $\Pi$, we see by definition of $L^{(t)}$ that there is some $\pi' \in \Psi$ such that

$$|L^{(t)}(\pi) - L^{(t)}(\pi')| \lesssim \beta\epsilon_{\text{disc}} + \frac{\beta}{\alpha(V_{\text{max}} + R_{\text{max}})^2} \cdot \beta\epsilon_{\text{disc}}(V_{\text{max}} + R_{\text{max}}) \le \beta\epsilon_{\text{disc}} \left( 1 + \frac{\beta}{\alpha(V_{\text{max}} + R_{\text{max}})} \right)$$

and similarly

$$|\widehat{L}^{(t)}(\pi) - \widehat{L}^{(t)}(\pi')| \lesssim (t-1)\beta\epsilon_{\text{disc}} \left( 1 + \frac{\beta}{\alpha(V_{\text{max}} + R_{\text{max}})} \right).$$

It follows that, for all $2 \le t \le T$, since $\widehat{L}^{(t)}(\pi^{(t)}) \le 0$, we get

$$(t-1)L^{(t)}(\pi^{(t)}) \lesssim \frac{\beta}{\alpha} \log(2|\Psi|T/\rho) + V_{\text{max}}\sqrt{T \log(2|\Psi|T/\rho)} + \beta\epsilon_{\text{disc}}T \left( 1 + \frac{\beta}{\alpha(V_{\text{max}} + R_{\text{max}})} \right).$$

Hence,

$$\frac{1}{T} \sum_{t=1}^T J_{\beta}(\pi_{\beta}^{\star}) - J_{\beta}(\pi^{(t)})$$

$$\lesssim \frac{\alpha}{\beta}(R_{\text{max}} + V_{\text{max}})^2 \cdot \text{SEC}(\Pi) + \frac{\beta}{\alpha T} + \frac{V_{\text{max}}}{T} + \frac{1}{T} \sum_{t=2}^T L^{(t)}(\pi^{(t)})$$

$$\lesssim (R_{\text{max}} + V_{\text{max}})\sqrt{\frac{\text{SEC}(\Pi)\log(2|\Psi|T/\rho)}{T}} + \beta\epsilon_{\text{disc}}\sqrt{\text{SEC}(\Pi)T}$$

by taking

$$\alpha := \frac{\beta}{R_{\mathsf{max}} + V_{\mathsf{max}}} \sqrt{\frac{\log(2|\Psi|T/\rho)}{\mathsf{SEC}(\Pi)T}}.$$

Since the output $\widehat{\pi}$ of Algorithm 1 satisfies $\widehat{\pi} \in \arg\max_{t\in[T]} J_\beta(\pi^{(t)})$, the claimed bound on $J_\beta(\pi_\beta^\star) - J_\beta(\widehat{\pi})$ is immediate. Finally, observe that by definition of $\pi_\beta^\star$,

$$
\begin{aligned}
J_\beta(\pi_\beta^\star) - J_\beta(\widehat{\pi}) &= \underset{(x,y)\sim\pi_\beta^\star}{\mathbb{E}}\left[r(x,y) - \beta\log\frac{\pi_\beta^\star(y\mid x)}{\pi_{\mathsf{base}}(y\mid x)}\right] - \underset{(x,y)\sim\widehat{\pi}}{\mathbb{E}}\left[r(x,y) - \beta\log\frac{\widehat{\pi}(y\mid x)}{\pi_{\mathsf{base}}(y\mid x)}\right] \\
&= \underset{(x,y)\sim\pi_\beta^\star}{\mathbb{E}}\left[r(x,y) - \beta\log\frac{\pi_\beta^\star(y\mid x)}{\pi_{\mathsf{base}}(y\mid x)}\right] - \underset{(x,y)\sim\widehat{\pi}}{\mathbb{E}}\left[r(x,y) - \beta\log\frac{\pi_\beta^\star(y\mid x)}{\pi_{\mathsf{base}}(y\mid x)}\right] \\
&\quad + \underset{(x,y)\sim\widehat{\pi}}{\mathbb{E}}\left[\beta\log\frac{\widehat{\pi}(y\mid x)}{\pi_\beta^\star(y\mid x)}\right] \\
&= \beta\log\underset{(x,y)\sim\pi_{\mathsf{base}}}{\mathbb{E}}[\exp(r(x,y))] - \beta\log\underset{(x,y)\sim\pi_{\mathsf{base}}}{\mathbb{E}}[\exp(r(x,y))] + \beta D_{\mathsf{KL}}\big(\widehat{\pi}\,\|\,\pi_\beta^\star\big) \\
&= \beta D_{\mathsf{KL}}\big(\widehat{\pi}\,\|\,\pi_\beta^\star\big).
\end{aligned}
$$

This completes the proof. $\qquad\square$

### J.2.3 APPLYING XPO TO MAXIMUM-LIKELIHOOD SHARPENING

We now prove Theorem J.2, the formal statement of Theorem 4.3, which applies XPO to maximum-likelihood sharpening. This result is a straightforward corollary of Theorem J.1 with the reward function $r_{\mathsf{self}}(x,y) := \log\pi_{\mathsf{base}}(y\mid x)$, together with the observation that low KL-regularized regret implies sharpness (under Assumption 4.2).

**Theorem J.2** (Sharpening via active exploration). *There are absolute constants $c_{\mathrm{J.2}}, C_{\mathrm{J.2}} > 0$ so that the following holds. Let $\epsilon, \delta, \gamma_{\mathsf{margin}}, \rho, \beta \in (0,1)$ and $T \in \mathbb{N}$ be given. For base model $\pi_{\mathsf{base}}$, define reward function $r(x,y) := \log\pi_{\mathsf{base}}(y\mid x)$. Let $R_{\mathsf{max}} \geq 1 + \max_{x,y}\log\frac{1}{\pi_{\mathsf{base}}(y\mid x)}$. Suppose that $\pi_{\mathsf{base}}$ satisfies Assumption 4.2 with parameter $\gamma_{\mathsf{margin}}$, that $\beta^{-1} \geq 2\gamma_{\mathsf{margin}}^{-1}\log(2|\mathcal{Y}|/\delta)$, and that there is $\epsilon_{\mathsf{disc}} \in (0,1)$ so that*

$$T \geq C_{\mathrm{J.2}} \frac{R_{\mathsf{max}}^2 \mathsf{SEC}(\Pi)\log(2\mathcal{N}(\Pi,\epsilon_{\mathsf{disc}})T/\rho)}{\epsilon^2\delta^2\beta^2}$$

*and*

$$\epsilon_{\mathsf{disc}} \leq c_{\mathrm{J.2}} \frac{\epsilon\delta}{\sqrt{\mathsf{SEC}(\Pi)T}}$$

*where $\mathsf{SEC}(\Pi) := \mathsf{SEC}(\Pi, r, T, \beta, R_{\mathsf{max}}^2; \pi_{\mathsf{base}})$. Also suppose that $\pi_\beta^\star \in \Pi$ where $\pi_\beta^\star(y\mid x) \propto \pi_{\mathsf{base}}^{1+\beta^{-1}}(y\mid x)$.*

*Then applying Algorithm 1 with base model $\pi_{\mathsf{base}}$, reward function $r$, iteration count $T$, regularization $\beta$, and optimism parameter $\alpha := \frac{\beta}{R_{\mathsf{max}}}\sqrt{\frac{\log(2\mathcal{N}(\Pi,\epsilon_{\mathsf{disc}})T/\delta)}{\mathsf{SEC}(\Pi)T}}$ yields a model $\widehat{\pi} \in \Pi$ such that with probability at least $1 - \rho$,*

$$\mathbb{P}_{x\sim\mu}[\widehat{\pi}(\boldsymbol{y}^\star(x)\mid x) < 1 - \delta] \leq \epsilon.$$

*The total sample complexity is*

$$m = \widetilde{O}\left(\frac{R_{\mathsf{max}}^2\mathsf{SEC}(\Pi)\log(\mathcal{N}(\Pi,\epsilon_{\mathsf{disc}})/\rho)\log^2(|\mathcal{Y}|\delta^{-1})}{\gamma_{\mathsf{margin}}^2\epsilon^2\delta^2}\right).$$

**Proof of Theorem J.2.** By definition of $r$, we have $|r(x,y)| \leq R_{\mathsf{max}}$ for all $x,y$. By assumption, Assumption J.3 is satisfied, and by definition of $R_{\mathsf{max}}$, Assumption 4.5 is satisfied with parameter

$V_{\max} := \beta R_{\max} \leq R_{\max}$. It follows from Theorem J.1 that with probability at least $1 - \rho$, the output $\widehat{\pi}$ of Algorithm 1 satisfies

$$\beta D_{\mathsf{KL}}\left(\widehat{\pi} \,\|\, \pi_\beta^\star\right) \lesssim (R_{\max} + V_{\max})\sqrt{\frac{\mathsf{SEC}(\Pi)\log(2\mathcal{N}(\Pi, \epsilon_{\mathsf{disc}})T/\rho)}{T}}$$
$$+ \beta\epsilon_{\mathsf{disc}}\sqrt{\mathsf{SEC}(\Pi)T}.$$

By choice of $T$ and $\epsilon_{\mathsf{disc}}$, so long as $C_{\mathsf{J}.2} > 0$ is chosen to be a sufficiently large constant and $c_{\mathsf{J}.2} > 0$ is chosen to be a sufficiently small constant, we have $\beta D_{\mathsf{KL}}\left(\widehat{\pi} \,\|\, \pi_\beta^\star\right) \leq \frac{1}{12}\beta\epsilon\delta$, so by e.g. Equation (16) of Sason & Verdú (2016), $D_{\mathsf{H}}^2\left(\widehat{\pi}, \pi_\beta^\star\right) \leq \epsilon\delta/(12)$.

For any $x \in \mathcal{X}$ and $y' \in \mathcal{Y} \setminus \boldsymbol{y}^\star(x)$, by Assumption 4.2 and definition of $\pi_\beta^\star$ we have

$$\frac{1}{\pi_\beta^\star(y' \mid x)} \geq \frac{\max_{y \in \mathcal{Y}} \pi_\beta^\star(y \mid x)}{\pi_\beta^\star(y' \mid x)} = \left(\frac{\max_{y \in \mathcal{Y}} \pi_{\mathsf{base}}(y \mid x)}{\pi_{\mathsf{base}}(y' \mid x)}\right)^{1+\beta^{-1}}$$
$$\geq (1 + \gamma_{\mathsf{margin}})^{1+\beta^{-1}} \geq e^{\gamma_{\mathsf{margin}}/(2\beta)} \geq \frac{2|\mathcal{Y}|}{\delta}$$

where the final inequality is by the assumption on $\beta$ in the theorem statement. Therefore

$$\pi_\beta^\star(\boldsymbol{y}^\star(x) \mid x) \geq 1 - \sum_{y' \in \mathcal{Y} \setminus \boldsymbol{y}^\star(x)} \pi_\beta^\star(y' \mid x) \geq 1 - \frac{\delta}{2}.$$

Now for any $x$, we can lower bound

$$D_{\mathsf{H}}^2\left(\widehat{\pi}(\cdot \mid x), \pi_\beta^\star(\cdot \mid x)\right) \geq \left(\sqrt{1 - \widehat{\pi}(\boldsymbol{y}^\star(x) \mid x)} - \sqrt{1 - \pi_\beta^\star(\boldsymbol{y}^\star(x) \mid x)}\right)^2$$
$$\geq \frac{\delta}{12} \cdot \mathbb{I}\{\widehat{\pi}(\boldsymbol{y}^\star(x) \mid x) \leq 1 - \delta\}.$$

Hence,

$$\mathbb{P}_{x \sim \mu}[\widehat{\pi}(\boldsymbol{y}^\star(x) \mid x) < 1 - \delta] \leq \frac{12}{\delta}\mathbb{E}_{x \sim \mu}D_{\mathsf{H}}^2\left(\widehat{\pi}(\cdot \mid x), \pi_\beta^\star(\cdot \mid x)\right)$$
$$= \frac{12}{\delta}D_{\mathsf{H}}^2\left(\widehat{\pi}, \pi_\beta^\star\right)$$
$$\leq \epsilon.$$

as claimed. $\qquad\qquad\qquad\qquad\qquad\qquad\qquad\qquad\qquad\qquad\qquad\qquad\qquad\qquad\qquad\qquad\square$

### J.2.4 APPLICATION: LINEAR SOFTMAX MODELS

In this section we apply Theorem 4.3 to the class of linear softmax models, proving Theorem J.3. This demonstrates that Algorithm 1 can achieve an exponential improvement in sample complexity compared to SFT-Sharpening.

**Definition J.3** (Linear softmax model). *Let $d \in \mathbb{N}$ be given, and let $\phi : \mathcal{X} \times \mathcal{Y} \to \mathbb{R}^d$ be a feature map with $\|\phi(x, y)\|_2 \leq 1$ for all $x, y$. Let $\pi_{\mathsf{zero}} : \mathcal{X} \to \Delta(\mathcal{Y})$ be the uniform model $\pi_{\mathsf{zero}}(y \mid x) := \frac{1}{|\mathcal{Y}|}$, and let $B \geq 1$.[15] We consider the linear softmax model class $\Pi_{\phi,B} := \{\pi_\theta : \theta \in \mathbb{R}^d, \|\theta\|_2 \leq B\}$ where $\pi_\theta : \mathcal{X} \to \Delta(\mathcal{Y})$ is defined by*

$$\pi_\theta(y \mid x) \propto \pi_{\mathsf{zero}}(y \mid x)\exp(\langle\phi(x, y), \theta\rangle).$$

**Theorem J.3.** *Let $\epsilon, \delta, \gamma_{\mathsf{margin}}, \rho \in (0, 1)$ be given. Suppose that $\pi_{\mathsf{base}} = \pi_{\theta^\star} \in \Pi_{\phi,B}$ for some $\theta^\star \in \mathbb{R}^d$ with $\|\theta^\star\|_2 \leq \frac{\gamma_{\mathsf{margin}}B}{3\log(2|\mathcal{Y}|/\delta)}$. Also, suppose that $\pi_{\mathsf{base}}$ satisfies Assumption 4.2 with parameter $\gamma_{\mathsf{margin}}$. Then Algorithm 1 with base model $\pi_{\mathsf{base}}$, reward function $r(x, y) := \log \pi_{\mathsf{base}}(x, y)$, regularization parameter $\beta := \gamma_{\mathsf{margin}}/(2\log(2|\mathcal{Y}|/\delta))$, and optimism parameter $\alpha(T) \propto \frac{\beta}{B+\log(|\mathcal{Y}|)}\sqrt{\frac{d\log(BdT/(\epsilon\delta))+\log(T/\rho)}{dT\log(T)}}$ returns an $(\epsilon, \delta)$-sharpened model with probability at least $1 - \rho$, and has sample complexity*

$$m = \mathrm{poly}(\epsilon^{-1}, \delta^{-1}, \gamma_{\mathsf{margin}}^{-1}, d, B, \log(|\mathcal{Y}|/\rho)).$$

---

[15] We use the notation $\pi_{\mathsf{zero}}$ to highlight the fact that $\pi_{\mathsf{zero}} = \pi_\theta$ for $\theta = 0$.

Before proving the result, we unpack the conditions. Theorem J.3 requires the base model $\pi_{\text{base}}$ to lie in the model class and also satisfy the margin condition (Assumption 4.2). For any constant $\epsilon, \delta > 0$, the sharpening algorithm then succeeds with sample complexity $\text{poly}(d, \gamma_{\text{margin}}^{-1}, B, \log(|\mathcal{Y}|))$. These conditions are non-vacuous; in fact, there are fairly natural examples for which non-exploratory algorithm such as SFT-Sharpening require sample complexity $\exp(\Omega(d))$, whereas all of the above parameters are $\text{poly}(d)$. The following is one such example.

**Example J.1** (Separation between RLHF-Sharpening and SFT-Sharpening). Set $\mathcal{X} = \{x\}$ and let $\mathcal{Y} \subset \mathbb{R}^d$ be a $1/4$-packing of the unit sphere in $\mathbb{R}^d$ of cardinality $\exp(\Theta(d))$. Define $\phi : \mathcal{X} \times \mathcal{Y} \to \mathbb{R}^d$ by $\phi(x, y) := y$, and let $B = Cd \log d$ for an absolute constant $C > 0$. Fix any $y^\star \in \mathcal{Y}$ and define $\pi_{\text{base}} := \pi_{\theta^\star} \in \Pi_{\phi, B}$ by $\theta^\star := y^\star$. Then for any $y \neq y^\star$, we have $\langle y, y^\star \rangle \leq 1 - \Omega(1)$, so

$$\frac{\pi_{\text{base}}(y^\star \mid x)}{\pi_{\text{base}}(y \mid x)} = \exp(\langle y^\star - y, y^\star \rangle) = \exp(\Omega(1)) = 1 + \Omega(1).$$

Thus, $\pi_{\text{base}}$ satisfies Assumption 4.2 with $\gamma_{\text{margin}} = \Omega(1)$. Moreover, $\|\theta^\star\|_2 = 1 \leq \frac{\gamma_{\text{margin}} B}{3 \log(2|\mathcal{Y}|/\delta)}$ for any $\delta = 1/\text{poly}(d)$, so long as $C$ is a sufficiently large constant. It follows from Theorem J.3 that Algorithm 1 computes an $(\epsilon, \delta)$-sharpened model with sample complexity $\text{poly}(\epsilon^{-1}, \delta^{-1}, d)$. However, since $\pi_{\text{base}}(y^\star \mid x) \leq \pi_{\text{base}}(y \mid x) \cdot \exp(2)$ for all $y \in \mathcal{Y}$, it is clear that

$$C_{\text{cov}} = \mathbb{E}\left[\frac{1}{\pi_{\text{base}}(\boldsymbol{y}^\star(x) \mid x)}\right] = \frac{1}{\pi_{\text{base}}(y^\star \mid x)} = \Omega(|\mathcal{Y}|) = \exp(\Omega(d)).$$

Thus, the sample complexity guarantee for SFT-Sharpening in Theorem 4.1 will incur *exponential* dependence on $d$ in the sample complexity. It is straightforward to check that this dependence is real for SFT-Sharpening, and not just an artifact of the analysis, since the model that SFT-Sharpening is trying to learn (via MLE) will itself not be sharp in this example, unless $\exp(\Omega(d))$ samples are drawn per prompt. ◁

We now proceed to the proof of Theorem J.3, which requires the following bounds on the covering number and the Sequential Extrapolation Coefficient of $\Pi_{\phi, B}$.

**Lemma J.4.** Let $\epsilon_{\text{disc}} > 0$. Then $\Pi_{\phi, B}$ has an $\epsilon_{\text{disc}}$-net of size $(6B/\epsilon_{\text{disc}})^d$.

**Proof of Lemma J.4.** By a standard packing argument, there is a set $\{\theta_1, \ldots, \theta_N\}$ of size $(6B/\epsilon_{\text{disc}})^d$ such that for every $\theta \in \mathbb{R}^d$ with $\|\theta\|_2 \leq B$ there is some $i \in [N]$ with $\|\theta_i - \theta\|_2 \leq \epsilon_{\text{disc}}/2$. Now for any $x \in \mathcal{X}$ and $y \in \mathcal{Y}$,

$$\log \frac{\pi_\theta(y \mid x)}{\pi_{\theta_i}(y \mid x)} = \log \frac{\exp(\langle \phi(x, y), \theta \rangle)}{\exp(\langle \phi(x, y), \theta_i \rangle)} + \log \frac{\mathbb{E}_{(x', y') \sim \pi_{\text{zero}}} \exp(\langle \phi(x', y'), \theta_i \rangle)}{\mathbb{E}_{(x', y') \sim \pi_{\text{zero}}} \exp(\langle \phi(x', y'), \theta \rangle)}$$

$$= \langle \phi(x, y), \theta - \theta_i \rangle + \log \frac{\mathbb{E}_{(x', y') \sim \pi_{\text{zero}}} \left[ \exp(\langle \phi(x', y'), \theta \rangle) \exp(\langle \phi(x', y'), \theta_i - \theta \rangle) \right]}{\mathbb{E}_{(x', y') \sim \pi_{\text{zero}}} \exp(\langle \phi(x', y'), \theta \rangle)}.$$

The first term is bounded by $\epsilon_{\text{disc}}/2$ in magnitude. In the second term, we have $\exp(\langle \phi(x', y'), \theta_i - \theta \rangle) \in [\exp(-\epsilon_{\text{disc}}/2), \exp(\epsilon_{\text{disc}}/2)]$, so the ratio of expectations lies in $[\exp(-\epsilon_{\text{disc}}/2), \exp(\epsilon_{\text{disc}}/2)]$ as well, and so the log-ratio lies in $[-\epsilon_{\text{disc}}/2, \epsilon_{\text{disc}}/2]$. In all, we get $\left| \log \frac{\pi_\theta(y|x)}{\pi_{\theta_i}(y|x)} \right| \leq \epsilon_{\text{disc}}$. Thus, $\{\pi_{\theta_1}, \ldots, \pi_{\theta_N}\}$ is an $\epsilon_{\text{disc}}$-net for $\Pi$. □

**Lemma J.5.** Let $r : \mathcal{X} \times \mathcal{Y} \to [-R_{\max}, R_{\max}]$ be a reward function and let $T \in \mathbb{N}$ and $\beta > 0$. If $\lambda \geq 4\beta^2 B^2 + R_{\max}^2$ then for any $\pi^\star \in \Pi_{\phi, B}$,

$$\text{SEC}(\Pi_{\phi, B}, r, T, \beta, \lambda; \pi^\star) \lesssim d \log(T + 1).$$

**Proof of Lemma J.5.** Fix $\pi^{(1)}, \ldots, \pi^{(T)} \in \Pi_{\phi, B}$. By definition, there are some $\theta^{(1)}, \ldots, \theta^{(T)} \in \mathbb{R}^d$ with $\|\theta^{(t)}\|_2 \leq B$ and

$$\pi^{(t)}(y \mid x) \propto \pi_{\text{zero}}(y \mid x) \exp(\langle \phi(x, y), \theta^{(t)} \rangle)$$

for all $t \in [T]$ and $(x, y) \in \mathcal{X} \times \mathcal{Y}$. Similarly, there is some $\theta^\star \in \mathbb{R}^d$ with $\|\theta^\star\|_2 \leq B$ and $\pi^\star(y \mid x) \propto \pi_{\text{zero}}(y \mid x) \exp(\langle \phi(x, y), \theta^\star \rangle)$.

Define $\widetilde{\phi} : \mathcal{X} \times \mathcal{Y} \to \mathbb{R}^{d+1}$ by $\widetilde{\phi}(x,y) := [\phi(x,y), \frac{r(x,y)}{R_{\max}}]$ and define $\widetilde{\theta}^{(t)} := [\beta(\theta^{(t)} - \theta^\star), -R_{\max}]$. Then for any $t \in [T]$ we have

$$\frac{\mathbb{E}^{(t)} \left[ \beta \log \frac{\pi^{(t)}(y|x)}{\pi^\star(y|x)} - r(x,y) - \beta \log \frac{\pi^{(t)}(y'|x)}{\pi^\star(y'|x)} + r(x,y') \right]^2}{\lambda \vee \sum_{i=1}^{t-1} \mathbb{E}^{(i)} \left[ \left( \beta \log \frac{\pi^{(t)}(y|x)}{\pi^\star(y|x)} - r(x,y) - \beta \log \frac{\pi^{(t)}(y'|x)}{\pi^\star(y'|x)} + r(x,y') \right)^2 \right]}$$

$$= \frac{\mathbb{E}^{(t)} \left[ \langle \widetilde{\phi}(x,y) - \widetilde{\phi}(x,y'), \widetilde{\theta}^{(t)} \rangle \right]^2}{\lambda \vee \sum_{i=1}^{t-1} \mathbb{E}^{(i)} \left[ \left( \langle \widetilde{\phi}(x,y) - \widetilde{\phi}(x,y'), \widetilde{\theta}^{(t)} \rangle \right)^2 \right]}$$

$$\leq \frac{(\widetilde{\theta}^{(t)})^\top \Sigma^{(t)} \widetilde{\theta}^{(t)}}{\lambda \vee \sum_{i=1}^{t-1} (\widetilde{\theta}^{(t)})^\top \Sigma^{(i)} \widetilde{\theta}^{(t)}}$$

where for each $i \in [T]$ we have defined $\Sigma^{(i)} := \mathbb{E}^{(i)} \left[ (\widetilde{\phi}(x,y) - \widetilde{\phi}(x,y'))(\widetilde{\phi}(x,y) - \widetilde{\phi}(x,y'))^\top \right]$. Observe that $\|\widetilde{\theta}^{(t)}\|_2^2 \leq 4\beta^2 B^2 + R_{\max}^2 \leq \lambda$ by assumption on $\lambda$. Therefore,

$$\frac{(\widetilde{\theta}^{(t)})^\top \Sigma^{(t)} \widetilde{\theta}^{(t)}}{\lambda \vee \sum_{i=1}^{t-1} (\widetilde{\theta}^{(t)})^\top \Sigma^{(i)} \widetilde{\theta}^{(t)}} \lesssim \frac{(\widetilde{\theta}^{(t)})^\top \Sigma^{(t)} \widetilde{\theta}^{(t)}}{\lambda + \sum_{i=1}^{t-1} (\widetilde{\theta}^{(t)})^\top \Sigma^{(i)} \widetilde{\theta}^{(t)}}$$

$$\leq \frac{(\widetilde{\theta}^{(t)})^\top \Sigma^{(t)} \widetilde{\theta}^{(t)}}{(\widetilde{\theta}^{(t)})^\top \left( I_d + \sum_{i=1}^{t-1} \Sigma^{(i)} \right) \widetilde{\theta}^{(t)}}$$

$$\leq \lambda_{\max} \left( \left( I_d + \sum_{i=1}^{t-1} \Sigma^{(i)} \right)^{-1/2} \Sigma^{(t)} \left( I_d + \sum_{i=1}^{t-1} \Sigma^{(i)} \right)^{-1/2} \right)$$

$$\leq \mathrm{Tr} \left( \left( I_d + \sum_{i=1}^{t-1} \Sigma^{(i)} \right)^{-1/2} \Sigma^{(t)} \left( I_d + \sum_{i=1}^{t-1} \Sigma^{(i)} \right)^{-1/2} \right)$$

$$= \mathrm{Tr} \left( \left( I_d + \sum_{i=1}^{t-1} \Sigma^{(i)} \right)^{-1} \Sigma^{(t)} \right).$$

Observe that $\mathrm{Tr}(\Sigma^{(t)}) \leq \max_{x,y} \|\widetilde{\phi}(x,y)\|_2^2 \lesssim 1$. Hence by Lemma F.2, we have

$$\sum_{t=1}^{T} \frac{\mathbb{E}^{(t)} \left[ \beta \log \frac{\pi^{(t)}(y|x)}{\pi^\star(y|x)} - r(x,y) - \beta \log \frac{\pi^{(t)}(y'|x)}{\pi^\star(y'|x)} + r(x,y') \right]^2}{\lambda \vee \sum_{i=1}^{t-1} \mathbb{E}^{(i)} \left[ \left( \beta \log \frac{\pi^{(t)}(y|x)}{\pi^\star(y|x)} - r(x,y) - \beta \log \frac{\pi^{(t)}(y'|x)}{\pi^\star(y'|x)} + r(x,y') \right)^2 \right]}$$

$$\lesssim \sum_{t=1}^{T} \mathrm{Tr} \left( \left( I_d + \sum_{i=1}^{t-1} \Sigma^{(i)} \right)^{-1} \Sigma^{(t)} \right)$$

$$\lesssim d \log(T+1).$$

Since $\pi^{(1)}, \ldots, \pi^{(T)} \in \Pi$ were arbitrary, this completes the proof. $\qquad \square$

The proof is now immediate from Theorem J.2 and the above lemmas.

**Proof of Theorem J.3.** By the assumption on $\theta^\star$ and choice of $\beta$, the model $\pi_\beta^\star$ defined by $\pi_\beta^\star(y \mid x) \propto \pi_{\mathsf{base}}(y \mid x)^{1+\beta^{-1}}$ satisfies $\pi_\beta^\star = \pi_{(1+\beta^{-1})\theta^\star} \in \Pi_{\phi,B}$. By Lemma J.4, we have $\mathcal{N}(\Pi_{\phi,B}, \epsilon_{\mathsf{disc}}) \leq (6B/\epsilon_{\mathsf{disc}})^d$. Take $R_{\max} := \sqrt{4\beta^2 B^2 + (2B + \log|\mathcal{Y}|)^2}$. We know that $r(x,y) := \log \pi_{\mathsf{base}}(y \mid x)$ satisfies $|r(x,y)| \leq 2B + \log|\mathcal{Y}|$ for all $x, y$. By Lemma J.5, we therefore get that $\mathsf{SEC}(\Pi_{\phi,B}, r, T, \beta, R_{\max}^2; \pi_{\mathsf{base}}) \lesssim d\log(T+1)$. Substituting these bounds into Theorem J.2 yields the claimed result. $\qquad \square$

