# OpenReview forum: "Self-Improvement in Language Models: The Sharpening Mechanism"
_ICLR.cc/2025/Conference — ICLR 2025 Oral_

### Official Review · Reviewer_ABNY · 2024-11-06

**Soundness:** 3
**Presentation:** 3
**Contribution:** 3
**Rating:** 8
**Confidence:** 3

**Summary:**

This paper proposes a theoretical framework to understand self-improvement. The authors define the sharpening as the best sequence according to the model likelihood which is different than greedy decoding or temperature sampling (the theory implies greedy decoding only covers the best sequence under some constraints). They further define self-improvement via SFT and RLHF using BoN and DPO, respectively. The authors show sample complexity bounds for SFT and RLHF using a model and data dependent coverage and concentrability coefficients. For example, sample complexity for SFT is proportional to coverage coefficient and logarithmically scales with size of the policy class. Finally, by adding exploration, the dependence on the coverage coefficient is replaced by sequential exploration coefficient.

**Strengths:**

Update: Authors' rebuttal addressed my concerns. I increased my score accordingly.

The proposed theoretical framework captures a very significant class of post-training improvements, ranging from SFT to RLHF. In particular, it covers BoN and DPO that are two very popular inference-time and post-training methods. The paper is also easy to follow.

**Weaknesses:**

My main concern is the lack of implications of the theoretical results for practical use.

1. While the theory covers many popular methods, such as BoN and DPO, there is not much connection to what this theory implies in practice.

A. For example, what predictions does it make about BoN, can we choose “N” based on your theory?

B. What is the minimal experimental setup to cover both SFT and RLHF?

C. How should we interpret Figure-1 based on your theory? Such as, would convergence rate of BoN be explained by your method?

2. While I understand that using the same context “x” for both policy and reward function is meaningful, in practice it is generally different. Can you discuss if an extension of your theory explains the sample complexity when using different contexts that are related by a function? In euclidean space, this could simply be defined as a bound on the distance between two context.

**Questions:**

Please see above for specific questions.

---

> ### Author Response · Authors · 2024-11-19
> **Response**
>
> We thank the reviewer for their careful attention to our work.
>
> > My main concern is the lack of implications of the theoretical results for practical use.  While the theory covers many popular methods, such as BoN and DPO, there is not much connection to what this theory implies in practice.  For example, what predictions does it make about BoN, can we choose “N” based on your theory?
>
> The main role of our theory is to propose sharpening as a simple, yet plausible mechanism for self-improvement in the absence of external feedback; while we analyze specific algorithms, our focus is not to prescribe a specific way to perform sharpening, and we view the algorithms and analyses themselves as supporting results.
>
> Nonetheless, in the rebuttal, we provide additional experiments that show that sharpening can indeed be amortized via SFT, which we hope resolves the disconnect between the presented theory and what is feasible in practice. Please refer to the discussion of additional experiments in the joint response to all reviewers above for more details.
>
>
> > What is the minimal experimental setup to cover both SFT and RLHF?
>
> Please refer to the discussion of additional experiments in the joint response to all reviewers above. Briefly, our new experiments show that it is indeed possible to amortize sharpening via SFT on tasks for which we report improvements from inference-time sharpening.
>
> > How should we interpret Figure-1 based on your theory? Such as, would convergence rate of BoN be explained by your method?
>
> We view Figure 1 primarily as a validation of the claim that maximum likelihood sharpening is a reasonable desideratum. Figure 1 demonstrates that if we are given sufficient inference-time compute, we can improve our performance on a number of benchmarks using inference-time BoN-Sharpening for a sufficiently large $N$. This is somewhat complementary to our main theoretical results in Section 4, which attempt to *amortize inference time compute into training* but do not say anything about the quality of the generations given sufficient inference-time compute (with BoN being a particular example of how one might use additional inference-time computation). In this regard, Figure 1 can be viewed as a sort of “skyline” on the performance improvement we may expect by running our proposed training-time sharpening algorithms.
>
> Let us mention in passing that we do give a theoretical guarantee for inference-time BoN-Sharpening in Proposition B.1, which shows that this algorithm succeeds whenever $N$ is chosen to be sufficiently large as a function of the coverage $\pi_{base}(y^{*}(x) | x)$ for the base model. Our results in Figure 1 suggest that the tasks we consider do indeed exhibit favorable coverage.
>
> (*In the above, we refer to the old Figure 1; the updated figure includes many more models and datasets.*)
>
> > While I understand that using the same context “x” for both policy and reward function is meaningful, in practice it is generally different. Can you discuss if an extension of your theory explains the sample complexity when using different contexts that are related by a function? In euclidean space, this could simply be defined as a bound on the distance between two context.
>
> We would appreciate it if the reviewer can expand upon this question with a bit more context, as we are a little confused about what the question is asking.  We can interpret the question in two distinct ways and address each below, but are happy to elaborate if we have misunderstood.
>
> Interpretation #1: The reviewer is asking whether our theory covers the setting where there are possibly different prompts (contexts) at training time and inference time. If the training-time prompts and inference-time prompts are different, but come from the same distribution, then this is precisely what our theory covers.  In greater detail, our theory considers the setting where prompts/contexts are drawn iid from some distribution and our performance is measured on new prompts drawn from the same distribution. An interesting direction for future research would be to provide guarantees for a more general setting in which the prompts/context at training time come from a different distribution than at test time; note however that this problem is not yet well understood even for simpler supervised learning settings, in spite of extensive research.
>
> Interpretation #2: The reviewer is considering a scenario where the reward is obtained by prompting a language model (i.e., LLM-as-a-judge). Further, the prompt consists of a “problem statement” as well as “other text” providing instructions, etc, and where the “other text” for the policy and the reward function are different. This setting is precisely captured by our framework: we should view “x” as just the “problem statement” while the “other text” is abstracted into the policy and reward.

---

> > ### Author Response · Authors · 2024-11-30
> > **Following up re: experiments and implications for practical use**
> >
> > Dear reviewer ABNY,
> >
> > We wanted to briefly follow up to see if you had a chance to look at the new experiments we have added (see the general response at the top of the page for an overview of the specific results). Please let us know if these results address any of your concerns regarding connections between theory and practice and takeaways for practical use.
> >
> > Thank you,
> > Authors

---

### Official Review · Reviewer_JzdL · 2024-11-10

**Soundness:** 3
**Presentation:** 3
**Contribution:** 3
**Rating:** 8
**Confidence:** 2

**Summary:**

This paper analyses self-improvement in language models (LLMs) through a mechanism called "sharpening." The authors focus on self-improvement where log-probabilities (of sequences of tokens) are used as the reward signal. They analyze two algorithms: SFT-Sharpening, which is optimal under sufficient model coverage, and RLHF-Sharpening, which can surpass SFT by utilizing online exploration to overcome coverage limitations.

**Strengths:**

This work develops a theoretical framework for studying and understanding self-improvement.
They compare two approaches, SFT-Sharpening and RLHF-Sharpening, and prove their convergence.

**Weaknesses:**

1. While the authors empirically show there is useful signal in Best-of-N Sharpening (by sampling at inference), they do not empirically verify that SFT or RLHF training with this signal works.
2. The authors focus their analysis on the maximum likelihood self-reward. While this objective is simple, it has not been empirically explored a lot in self-improvement literature (as the authors themselves note).
3. The sentence in the abstract "*Motivated by the observation that language models are often better at verifying response quality than they are at generating correct responses*" is misleading in the context of this paper. Here, the reward is not assigned by asking the model to verify or critique its own solutions but by having access the log-probabilities of generated solutions. In general (relevant for Introduction and Related Work sections), claims of self-improvement in LLMs should be handled with more nuance.  While there are papers that LLMs can self-improve without external feedback, there is also recent evidence of the contrary:
    -  Ryo Kamoi, Yusen Zhang, Nan Zhang, Jiawei Han, and Rui Zhang. 2024. *When can LLMs actually correct their own mistakes?
A critical survey of self-correction of LLMs*
    -  Kaya Stechly, Karthik Valmeekam, and Subbarao Kambhampati. 2024. *On the self-verification limitations of large language models on reasoning and planning tasks*
    -  Jie Huang, Xinyun Chen, Swaroop Mishra, Huaixiu Steven Zheng, Adams Wei Yu, Xinying Song, and Denny Zhou. 2023. *Large language models cannot self-correct reasoning yet.*
    -  Kaya Stechly, Matthew Marquez, and Subbarao Kambhampati. 2023. *Gpt-4 doesn’t know it’s wrong: An analysis of iterative prompting for reasoning problems.*
    -  Karthik Valmeekam, Matthew Marquez, and Subbarao Kambhampati. 2023. *Can large language models really improve by self-critiquing their own plans?*

**Questions:**

While theoretically sound, can your work provide any guidelines for the practical deployment of sharpening techniques?

---

> ### Author Response · Authors · 2024-11-19
>
> We thank the reviewer for their careful attention to our work.
>
> > While the authors empirically show there is useful signal in Best-of-N Sharpening (by sampling at inference), they do not empirically verify that SFT or RLHF training with this signal works.
>
> Please refer to the discussion of additional experiments in the joint response to all reviewers above. Briefly, our new experiments show that it is indeed possible to amortize sharpening via SFT on tasks for which we report improvements from inference-time sharpening.
>
>
> > The authors focus their analysis on the maximum likelihood self-reward. While this objective is simple, it has not been empirically explored a lot in self-improvement literature (as the authors themselves note).
>
> We do not view this as a weakness, but rather a contribution/novelty. In particular, the sharpening (maximum likelihood self-reward) framework permits strong theoretical guarantees, and our inference-time results show that empirically, it is a powerful self-improvement signal. Unlike reward-based methods, this self-reward comes “for free”, as it does not require external feedback or preference data. For future work, we hope to build on these theoretical techniques to analyze more complex self-reward heuristics such as self-judging, with the present work serving as a starting point.
>
> > The sentence in the abstract "Motivated by the observation that language models are often better at verifying response quality than they are at generating correct responses" is misleading in the context of this paper. Here, the reward is not assigned by asking the model to verify or critique its own solutions but by having access the log-probabilities of generated solutions.
>
> We view the log-probabilities as a stylized self-reward function, which offers perhaps the simplest objective for self-improvement in the absence of external feedback, and which permits strong theoretical guarantees; other self-rewards derived from asking the model to verify or critique its own responses also fall into our general formulation in Eq. (1), but may not admit such strong guarantees in general (though this is an interesting topic to explore in future research). In particular, not all models, e.g., smaller or less general ones, are accurate at judging themselves, and this is where the simplicity of our framework is beneficial.
>
> > Recent evidence contradicting that LLMs can self-improve without external feedback
>
> We are not claiming that LLMs can always self-improve; we identify specific conditions (boundedness of the coverage coefficient in Eq. (6), and expressivity of the model class (Assumption 4.1 or 4.3)) under which the sharpening methods we consider are guaranteed to do so in theory, but make no claims outside of the regime where these assumptions hold.
> That said, we agree it would be helpful to cite these works and include the above discussion of how they pertain to our results.
>
> > Can your work provide any guidelines for the practical deployment of sharpening techniques?
>
> Yes, we show empirically that inference-time maximum-likelihood sharpening can lead to improvement across a variety of standard benchmarks and models; note that this improvement comes “for free”, in the sense that it only uses the base model, and does not require a reward model or external feedback. We also show that it is indeed possible to amortize this form of sharpening via SFT.
>
> Please refer to the joint response to all reviewers above for further discussion.

---

> ### Comment · Reviewer_JzdL · 2024-11-27
>
> Thank you for the responses to my questions. Additional experiments strengthen the contribution of this paper, so I am increasing my score.

---

### Official Review · Reviewer_6usC · 2024-11-11

**Soundness:** 3
**Presentation:** 4
**Contribution:** 3
**Rating:** 8
**Confidence:** 3

**Summary:**

The paper provides an answer to why self-improvement methods can improve model performance even if they do not increase the information the mode has access to.
It defines sharpening as the greedification of a model, i.e. outputting its most likely answer, which is typically hard to compute.
Sharpening can be applied at inference time (e.g. with Best of N) or at training time, and the paper identifies self-improvement as a training-time sharpening method which allows to replace or amortize inference-time sharpening.
It defines a framework for theoretically evaluating the sample complexity of sharpening methods and provides results for the sample complexity of SFT and RLHF sharpening/self-improvement (using the model's likelihood as a filter or a reward), giving sample complexity bounds for each method to achieve sharpening at training time.

**Strengths:**

### Contribution
- The identification of self-improvement methods as sharpening methods that trade training time for inference-time search is novel and important in the discussion of self-improvement methods.
- The sample complexity results provided for SFT-Sharpening and RLHF-Sharpening are novel and relevant to the community. The results on adding exploration to RLHF methods are relevant to the online vs offline discussion of post-training.
- The additional results in the Appendix, such as adaptive sampling for SFT-Sharpening to improve its sampling complexity, are appreciated and can drive the design of novel practical algorithms.

### Soundness
- The sharpening objective and definition and the sample-and-evaluate framework are well-motivated and provide results that are relevant to practical settings.
- The assumptions made in the paper seem reasonable.

### Presentation
- The paper is very well written and easy to follow. The claims and their associated evidence/results are easy to identify.
- The remarks discussing the assumptions in depth, such as Remark 4.1, are well appreciated.

**Weaknesses:**

### Contribution:
- Line 256: It's appreciated and essential that the paragraph "Empirical validation of maximum-likelihood sharpening" verifies that sharpening provides downstream task improvement; however, in my view, this is a fairly known fact which could be presented in a more concise way, or acknowledged as a general fact.
- Definition 3.2: the generation-verification operation seems a bit restricted, but in a self-improvement context, this seems enough to me.
- The results seem limited to the specific algorithms chosen IN RLHF-sharpening mainly REBEL and XPO.

### Presentation:
- It can be misunderstood from the paper that self-improvement methods should exclusively be seen as sharpening methods, from the strong questions raised in the abstract and the introduction. I suggest rephrasing those as motivation to see self-improvement as sharpening.
- Line 194 typo: the expectation's long definition should be over $\pi$ instead of $\pi_{base}$.
- Proposition 3.1 introduces an important result, which is then not discussed. The transition to the next section is too abrupt.

**Questions:**

- Could the authors elaborate on their choice of REBEL and XPO as the algorithms studied?
- To what extent are the results in the paper limited to these algorithms?

---

> ### Author Response · Authors · 2024-11-19
> **Response**
>
> We thank the reviewer for their careful attention to our work and for catching the typo in line 194.
>
> > Line 256: It's appreciated and essential that the paragraph "Empirical validation of maximum-likelihood sharpening" verifies that sharpening provides downstream task improvement; however, in my view, this is a fairly known fact which could be presented in a more concise way, or acknowledged as a general fact.
>
> We are not familiar with prior works that explicitly validate the utility of log-probabilities as a reward (rather than some external reward or more complex self-reward); an exception is that various decoding strategies like beam search (Meister et al., 2020) or chain-of-thought decoding (Wang & Zhou, 2024) admit an interpretation as sharpening w.r.t. log-probabilities, which we discuss in Appendix A. However, if the reviewer has specific references in mind we are more than happy to add them.
>
>
> > The results seem limited to the specific algorithms chosen IN RLHF-sharpening mainly REBEL and XPO.
>
> We emphasize that our general framework is not restricted to these algorithms and we could likely extend our results to most post-training algorithms that attempt to optimize some notion of reward. While we choose several (of the most ubiquitous) algorithms for concreteness, guarantees similar to Theorem 4.2 can likely be proven for most common RLHF algorithms. Theorem 4.3 is tailored to XPO because it makes use of *exploration*, and to our knowledge, XPO is currently the most well-known/realistic RLHF algorithm that provides provable exploration. We also remark that Theorem 4.1 is about an SFT-style algorithm (SFT-Sharpening), not REBEL/XPO.
>
>
> > It can be misunderstood from the paper that self-improvement methods should exclusively be seen as sharpening methods, from the strong questions raised in the abstract and the introduction. I suggest rephrasing those as motivation to see self-improvement as sharpening.
>
> Thank you for the suggestion, we will be sure to emphasize in the revision that sharpening is a *type* of self-improvement.
>
> > Proposition 3.1 introduces an important result, which is then not discussed. The transition to the next section is too abrupt.
>
> Thank you for pointing this out; we will include a more extensive discussion in the revision.  The main finding here is that for autoregressive models, Proposition 3.1 shows that one only needs to sharpen to constant precision $\delta$ if the arg-max sequence is unique. That is, it suffices to take $\delta < 1 / 2$ for greedy decoding on the sharpened model to return the arg-max sequence. This is an important takeaway, as the complexity guarantees for SFT-Sharpening and RLHF-Sharpening depend polynomially on $1/\delta$, so setting delta to be constant can improve sample complexity.

---

> > ### Comment · Reviewer_6usC · 2024-11-20
> >
> > I thank the authors for the clarifications. My comment about line 210 (previously 194) still holds.

---

> > > ### Author Response · Authors · 2024-11-20
> > >
> > > Thank you, that typo is now fixed in the revised draft.

---

### Official Review · Reviewer_6RKv · 2024-11-12

**Soundness:** 3
**Presentation:** 4
**Contribution:** 4
**Rating:** 8
**Confidence:** 3

**Summary:**

This paper introduces a theoretical framework for analyzing language model self-improvement through the lens of "sharpening"—defined as the process of tilting a model's probability mass toward sequences with high self-reward (which crucially does not rely on any external information). The paper focuses mostly on the case where self-reward = log π(y|x), which has been used in previous self-improvement literature. It also focuses mostly on sharpening via self-training (as opposed to sharpening via only inference-time computation), which can be viewed as amortizing expensive inference-time computation/sharpening. With this in mind, the authors present two families of algorithms for sharpening via self-training: SFT-sharpening and RLHF-sharpening. SFT-sharpening uses Best-of-N with self-reward to generate a new SFT dataset to fine-tune the model, while RLHF-sharpening uses RLHF with self-reward as the reward function for preference scoring (the paper looks at DPO-style algorithms specifically). These families of algorithms come from previous self-improvement literature.

To analyze the sample complexity of these algorithm families, the authors introduce a "sample-and-evaluate" framework where sample complexity is measured in terms of m = n * N sample-and-evaluate queries (n sampled prompts with N generations sampled per prompt). They prove a lower bound on the sample complexity for any sharpening algorithm. Then, they prove that both SFT- and RLHF-sharpening can learn sharpened models under the maximum likelihood objective (ie, maximizing self-reward), and they prove the sample complexity of each (with certain assumptions about model coverage). Finally, they demonstrate that adding online exploration to RLHF-sharpening can replace the dependence of sample complexity on model coverage to a dependency on the complexity of exploration.

**Strengths:**

- The paper is well written and easy to follow, with a detailed appendix containing proofs and additional results like guarantees for purely inference-time sharpening.
- The work is relevant to the scope of ICLR and introduces a novel theoretical perspective and statistical framework to analyze language model self-improvement algorithms. It provides a much needed theoretical understanding of recently popularized self-improvement methods for language models.
- The SFT-sharpening and RLHF-sharpening algorithm families are widely used in self-improvement literature, making the theoretical results immediately relevant and valuable as a foundation for future work.
- The result showing that using online exploration with RLHF-sharpening shifts dependency from model coverage to complexity of exploration is a particularly interesting and significant result. Future works can analyze how different online exploration strategies affect sample complexity.
- The choice to focus on log π(y|x) as self-reward is simple yet well-motivated given its use in prior self-improvement literature. The framework established by the authors provides a foundation for future work to analyze the effect of using other self-reward functions on sharpening guarantees and sample complexity.

**Weaknesses:**

- On lines 187 and 199, the paper states that the corresponding algorithm scheme converges to a sharpened model / sharpening objective in the limit. However, it does not seem to provide a justification or proof for these two statements (please let me know if I have missed it).
- Although focusing on log-probability as self-reward is justified, the paper would benefit from including a survey of other self-reward functions used in prior self-improvement works. This context would provide valuable guidance for others to build on the sharpening perspective by looking at alternative self-rewards.
- While the paper does mention the prior literature on self-distillation, it would benefit from a more complete summary of this prior line of work, including key findings and making it very clear how this work is different/novel.
- While emperical results are shown for inference-time sharpening, there are no experiments for self-training methods (SFT-sharpening, RLHF-sharpening) which are the main focus of the theoretical analysis. The paper would greatly benefit from including experimental results for SFT-sharpening and/or RLHF-sharpening, **with a focus on whether the theoretical results can be used to correctly predict something about the emperical results.** For example, can the theoretical results be used to predict the difference between running an RLHF-sharpening setup with and without online exploration? Or between SFT- and RLHF-sharpening? I would be curious to hear the authors' views on this.
- The conclusion focuses on future work but lacks a summary of the paper's key contributions and findings. The final manuscript would benefit from a summary of key takeaways in the conclusion. (No need to include it now in the rebuttal.)

**Questions:**

- What should experimentalists take away from this work? For example, researchers working on inference-time computation scaling, or those doing self-training. Can this work help them in some way?
- Do the authors see this work as being helpful for investigations into scaling laws for self-improvement? If so, how?
- In the RLHF-sharpening formulation, my understanding is that the KL-term ensures that the finetuned policy stays close to the base-policy, which is required since self-reward is defined using the base-policy (i.e., to keep generations in-distribution for the reward function). Is this correct? Furthermore, I am wondering whether the self-reward can be defined using the non-stationary policy that is being fine-tuned. What would be the effect of this change?
- Can you provide a more detailed explanation on the meaning of the coverage coefficient and how it can be problem dependent?
- [The questions mentioned in "Weaknesses"]

---

> ### Author Response · Authors · 2024-11-19
> **Response 1 of 2**
>
> We thank the reviewer for their careful attention to our work.
>
> > What should experimentalists take away from this work? For example, researchers working on inference-time computation scaling, or those doing self-training. Can this work help them in some way?
>
> Please refer to the discussion of additional experiments in the joint response to all reviewers above. Briefly, some key takeaways include:
> - We show empirically that inference-time maximum-likelihood sharpening can lead to improvement across a variety of standard benchmarks and models; note that this improvement comes “for free”, in the sense that only uses the base model, and does not require a reward model or external feedback.
> - We show that it is indeed possible to amortize this form of sharpening via SFT.
>
>
> > Do the authors see this work as being helpful for investigations into scaling laws for self-improvement? If so, how?
>
> If sharpening in the sense of Eq. (2) is the desideratum, then our work suggests that one should observe favorable scaling laws for sharpening-based self-improvement, both with respect to inference-time computation (e.g., through inference-time Best-of-N Sharpening) or through training-time computation (SFT-Sharpening or RLHF-Sharpening). However, we acknowledge that this is somewhat speculative, as sharpening is almost certainly not the *only* objective one might care about for self-training, and may not always correlate with performance at the task of interest. We hope to investigate this more deeply in future work.
>
> In addition, we have not deeply investigated the effect of model capacity/architecture etc on the ability to sharpen. In this direction, we noticed a concurrent paper, also in submission to ICLR, that considers scaling laws for self-improvement but does not use our sharpening setup (https://openreview.net/forum?id=mtJSMcF3ek). Developing analogous scaling laws for sharpening is a great direction for future work.
>
> > In the RLHF-sharpening formulation, my understanding is that the KL-term ensures that the finetuned policy stays close to the base-policy, which is required since self-reward is defined using the base-policy (i.e., to keep generations in-distribution for the reward function). Is this correct?
>
> Yes, this is correct; note however that the regularization parameter \beta is chosen to be relatively small for our theoretical guarantees, which ensures that the fine-tuned policy can become sufficiently sharpened.
>
> > Furthermore, I am wondering whether the self-reward can be defined using the non-stationary policy that is being fine-tuned. What would be the effect of this change?
>
> This is a very natural question! In fact, we can show that an iterative version of the RLHF-Sharpening algorithm which repeatedly updates the self-reward function based on policy being fine-tuned does indeed converge, and enjoys similar guarantees to Theorem 4.2. We do not currently know whether this approach enjoys provable *benefits* over Theorem 4.2 (one interesting difference is that it does permit a larger setting for the regularization parameter \beta discussed above), but we are very keen to explore this in future work.
>
> > Can you provide a more detailed explanation on the meaning of the coverage coefficient and how it can be problem dependent?
>
> Briefly, the coverage coefficient captures the probability mass that the base policy \pi_base places on the optimal sequence-level response $\pi*$. If the coverage coefficient is large, it means that we are very unlikely to observe the sequence-level response by sampling y ~ pi_base, which means that algorithms like SFT-Sharpening and RLHF-Sharpening will be slow to converge. This quantity is *problem-dependent* in the sense that it depends on both $\pi_base$ itself and $y*(x)$.
>
>
> > On lines 187 and 199, the paper states that the corresponding algorithm scheme converges to a sharpened model / sharpening objective in the limit. However, it does not seem to provide a justification or proof for these two statements (please let me know if I have missed it).
>
> This is proven in our main theoretical guarantees, Theorem 4.1 (SFT-Sharpening) and Theorem 4.2 (RLHF-Sharpening), provided the regularity assumptions (e.g., realizability for \Pi) required for the theorems to hold. We are happy to clarify this in the final version of the paper.
>
> > Although focusing on log-probability as self-reward is justified, the paper would benefit from including a survey of other self-reward functions used in prior self-improvement works. This context would provide valuable guidance for others to build on the sharpening perspective by looking at alternative self-rewards.
>
> Thank you for the helpful suggestion! We are happy to include more extensive discussion of the specific self-rewards used in the prior work we cite in the final version of the paper. If you are aware of any references we missed, please let us know and we’d be happy to include them as well!

---

> ### Author Response · Authors · 2024-11-19
> **Response 2 of 2**
>
> > While the paper does mention the prior literature on self-distillation, it would benefit from a more complete summary of this prior line of work, including key findings and making it very clear how this work is different/novel.
>
> Thank you for the natural suggestion! Most theoretical work on self-distillation (c.f Appendix A) has focused on supervised learning settings (e.g., regression or binary classification) instead of language modeling. For these settings, we typically have explicit supervision and a clear choice of metric (e.g., MSE or classification accuracy) through which to evaluate the algorithm. This is somewhat complementary to the question we consider, which asks *how* to evaluate self-training algorithms for generative language modeling in the absence of such supervision. Whether some form of self-distillation can be analyzed using the sharpening objective we consider is a very natural question, and we would be curious to explore this in future work.
>
> > While empirical results are shown for inference-time sharpening, there are no experiments for self-training methods (SFT-sharpening, RLHF-sharpening) which are the main focus of the theoretical analysis. The paper would greatly benefit from including experimental results for SFT-sharpening and/or RLHF-sharpening, with a focus on whether the theoretical results can be used to correctly predict something about the empirical results. For example, can the theoretical results be used to predict the difference between running an RLHF-sharpening setup with and without online exploration? Or between SFT- and RLHF-sharpening? I would be curious to hear the authors' views on this.
> Please refer to the discussion of additional experiments in the joint response to all reviewers above. Briefly, we show empirically that it is indeed possible to amortize sharpening via SFT, as predicted by our theoretical results.
>
> On the subject of whether the theory can predict more-detailed empirical phenomena like tradeoffs between algorithms, we agree that this is a good question. However, due to the scope and depth of an investigation into the empirical benefits and drawbacks, we leave this fascinating question open to further inquiry.

---

> > ### Comment · Reviewer_6RKv · 2024-11-23
> >
> > Thanks to the authors for their detailed reply to my questions, and for the additional experiments. I look forward to seeing future research build on this work.

---

### Author Response · Authors · 2024-11-19
**General Response on Experiments (See Appendix E of the new version)**

Several reviewers have asked about further experimental results involving both (i) alternative self-improvement self-reward function and (ii) implementing the amortization algorithms described by our theory.  In the revision, we have included a number of new experimental results both involving inference-time sharpening and amortization.  In particular, we have since conducted the following experiments:

1. **Inference-time experiments**: We have added a number of model-task pairs to our inference-time experiments, with considered models including Phi-3-mini, Phi-3.5-mini, Phi-3-small, Phi-3-Medium, Mistral-7B, Llama-3.2-3B, and GPT-3.5-Turbo.  We have also expanded the set of tasks we consider, with the set of tasks now being: GSM8K, MATH, ProntoQA, MMLU Biology, MMLU Chemistry, and MMLU Physics.  We have also implemented an inference-time BoN evaluation with a finetune of Llama on the Game24 task.

2. **Alternative self-reward functions**: We have explored alternative scoring metrics at inference time, with the following self-rewards considered: (i) Log probability (this is what we already had, and what our theory focuses on), (ii) Length-Normalized Log probability, and (iii) Majority voting on generations’ answers.  In addition, we have empirically validated the question of *coverage*, wherein we have explored the extent to which models produce at least one correct generation given $N$ generations (in the sense of Brown et al. (2024)).

3. **Experiments with training-time sharpening/amortization**: For training-time sharpening (amortization), we have implemented SFT-sharpening on a subset of the model-task pairs described above (Phi-3.5-mini with GSM8K, Math, and ProntoQA and Mistral7B with Math).

Our expanded inference-time experiments are consistent with the validation experiments in the original submission, but substantially expand their scope. In particular, we find that BoN sampling using log-probability as the self-reward function often leads to improved performance over greedy decoding (though not for every single model/task pair). The other self-reward functions we evaluate (length-normalized log-probability and majority) further improve performance; notably, majority improves over greedy decoding on all but 2 out of 42 (model,task) pairs.

Our experiments with SFT-Sharpening primarily validate the theoretical claim that this method can amortize inference-time computation. In particular, after sharpening/fine-tuning on a training set of prompts (with responses selected by BoN w.r.t. log-likelihood), the fine-tuned model outperforms greedy decoding on a held-out test set of prompts.

**See Appendix E of the new version of the paper that we uploaded for more details on our experiments**

---

> ### Comment · Reviewer_6usC · 2024-11-20
>
> I thank the authors for the complementary experiments. Can they also please add references to the figures that show the results of these experiments in this general response?

---

> > ### Author Response · Authors · 2024-11-20
> > **Figure References**
> >
> > We would be happy to give a guide to the new figures added.
> >
> > 1. **Inference Time Experiments:** We summarize our inference time experiments in  **Figure 1** by showing (a) the % lift in accuracy relative to greedy decoding of Best of N sampling with N=50 for all our model-dataset pairs except for Game24, which we display in Figure 5.  in (b) we show the effect that increasing N from 1 to 50 has on accuracy for all models on the MATH dataset and in (c) we display the empirical distribution of the log-likelihoods under the reference model (Phi-3.5-Mini in this case) of the reference model's sampled generations, conditioned on whether or not the generations are correct.  We display analogous figures to (b) above for all tasks in **Figure 3** and in **Figure 4** we report the effect that increasing N has on the log-likelihood of the response under the reference model.  Finally, in **Figure 6** we display the empirical distributions of log-likelihoods conditioned on correctness for additional model-dataset pairs.
> >
> > 2. **Alternative self-reward functions:** We display the results of Best of N sampling with alternative self-reward functions in **Figure 2** by reporting the % lift over greedy decoding that each inference-time sampling strategy achieves for all of our model-dataset tasks on (a) length-normalized sequence-level log-likelihood and (b) majority voting.  For both schemes we use $N = 50$.  In (c) of that figure, we report the result of our investigation of coverage, displaying the Pass@50 Accuracy (i.e., the percentage of prompts such that the reference model sampled with temperature 1 returns at least one correct response out of 50) for all of our model-dataset pairs.  Finally, in (d) we report the accuracy of greedily decoding the reference model for each model-dataset pair, which forms our baseline.
> >
> > 3. **Experiments with training-time sharpening:** We display the results of our experiment with SFT-Sharpening for a strict subset of the model-dataset pairs in **Table 1**, where we report both the % lift in accuracy over greedy decoding as well as the lift in sequence-level log-likelihood over the same for each studied model-dataset pair.  In **Figures 7 and 8** we display the evolution of accuracy and log-likelihood (according to the reference model) throughout training for each of these experiments and in **Figure 9** we report the effect that changing N has on SFT-Sharpening for a particular task.

---

> > > ### Author Response · Authors · 2024-11-20
> > > **General response regarding takeaways for experimentalists**
> > >
> > > Reviewers 6RKv and ABNY both had questions regarding takeaways for experimentalists. Briefly, some key points that empiricists can take away from our work are as follows:
> > > 1. **Validation of inference-time sharpening.** We show empirically that inference-time maximum-likelihood sharpening can lead to improvement across a variety of standard benchmarks and models; note that this improvement comes “for free”, in the sense that *no external feedback is required*. This is consistent with our theory, which shows that inference-time sharpening succeeds whenever the base model has sufficient coverage.
> > >
> > > 2. **Amortization.** Our experiments show that it is indeed possible to amortize sharpening via SFT; this is consistent with our theory, which shows that this is possible whenever the base model has sufficient coverage and the model class is sufficiently expressive.
> > >
> > > 3. **The role of coverage.** Our theoretical results highlight the importance of coverage (E. (6); i.e., the probability mass that the base policy \pi_base places on the optimal sequence-level response pi*), showing that this is necessary and sufficient to learn a sharpened policy. This is an important finding, as recent work (e.g, Brown et al. (2024)) shows that pre-trained language models often exhibit favorable coverage on standard tasks.

---

### Meta-Review · Area_Chair_nx2w · 2024-12-19

**Metareview:**

### Summary

This paper studies how language models can refine their outputs without external feedback, a process the authors call “sharpening.” The authors observe that models are often better at evaluating responses than generating them. This motivates the development of self-improvement algorithms that focus on shifting probability mass toward high-quality outputs. They propose a framework to analyze this process and examine two algorithms: one based on supervised fine-tuning (SFT) and another using reinforcement learning with human feedback (RLHF). The SFT approach generates fine-tuning data by selecting high-reward outputs, while the RLHF method uses a reward function derived from the model to refine preferences. The authors analyze these methods under a statistical framework and establish theoretical bounds on their efficiency, particularly regarding sample requirements. While SFT is optimal under certain conditions, RLHF can outperform it by leveraging exploration during training. This work provides a foundation for understanding and improving self-improvement strategies in language models.

### Decision

The paper is very well-written and has a very clear presentation. The research question the paper is studying is essential and relevant to the community. The insights provided in the paper are novel, and the theory is interesting. Thus, I recommend the paper for acceptance and spotlight.

**Additional Comments On Reviewer Discussion:**

Initially, the reviewers had some concerns about the paper and the findings presented in the paper. However, during the rebuttal phase, the authors have done an excellent job addressing that feedback. As a result, all the reviewers increased their scores to "8: accept, good paper". Thus, I recommend the paper for acceptance and spotlight presentation during the conference.

---

### Decision · Program_Chairs · 2025-01-22

Accept (Oral)